# Vision Transformer Finetuning Benefits from Non-Smooth Components

**Ambroise Odonnat** [1 2 3]  **Laetitia Chapel** [2 4]  **Romain Tavenard** [2 3]  **Ievgen Redko** [1]

## Abstract

The smoothness of the transformer architecture has been extensively studied in the context of generalization, training stability, and adversarial robustness. However, its role in transfer learning remains poorly understood. In this paper, we analyze the ability of vision transformer components to adapt their outputs to changes in inputs, or, in other words, their *plasticity*. Defined as an average rate of change, it captures the sensitivity to input perturbation; in particular, a high plasticity implies a low smoothness. Our theoretical analysis and extensive experiments – over $1,000$ finetuning runs on large-scale vision transformers – showcase that this perspective provides principled guidance in choosing the components to prioritize during adaptation. A key takeaway for practitioners is that the high plasticity of the attention modules and feedforward layers consistently leads to better finetuning performance. Our findings depart from the prevailing assumption that smoothness is desirable, offering a novel perspective on transformers' functional properties.

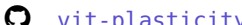 vit-plasticity

## 1. Introduction

Transformers (Vaswani et al., 2017) have become the default backbone of state-of-the-art models in a wide range of domains, including natural language processing (Brown et al., 2020; Touvron et al., 2023), computer vision (Caron et al., 2021; Dosovitskiy et al., 2021), time series forecasting (Ilbert et al., 2024; Nie et al., 2023), and mathematical reasoning (Comanici et al., 2025; Guo et al., 2025). These foundation models are typically pretrained on large amounts of diverse data and then adapted to more specific

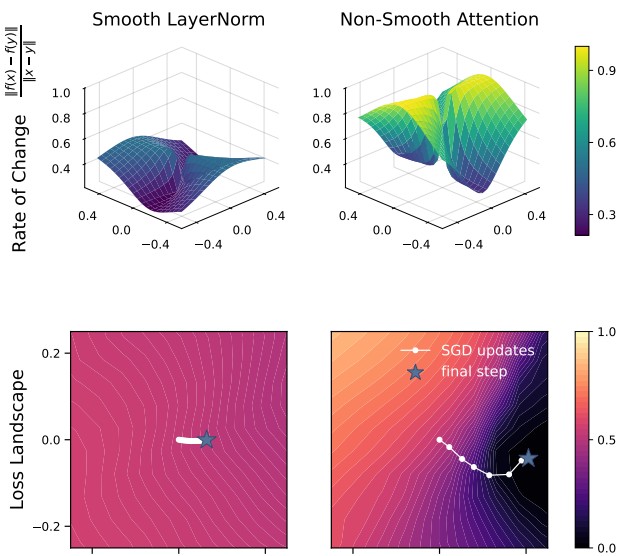

*Figure 1.* **Non-smooth components facilitate finetuning.** We illustrate the benefits of a high plasticity during the finetuning of ViT-Base on Cifar10 (values normalized to $[0, 1]$). Smooth modules like LayerNorm (**top left**) have low and steady rates of change, resulting in low plasticity (see Definition 1). This constrains the gradient norms during the optimization, leading to a slow descent on the loss landscape (**bottom left**). In contrast, the rates of change of non-smooth components, such as multi-head attention (**top right**), are large and vary a lot, resulting in high plasticity and gradients of high magnitude. This allows the exploration of the loss landscape and a faster descent towards (local) minima (**bottom right**).

domains (Shukor et al., 2025). In practice, the discrepancy between the training and downstream data can hurt performance (Quionero-Candela et al., 2009) and requires updating the model weights to adapt to the distribution shift.

**Finetuning foundation models.** The cost of adaptation has drastically increased, with models growing larger and larger as a byproduct of the scaling hypothesis (Hoffmann et al., 2022; Kaplan et al., 2020). This has led to considerable research effort toward parameter-efficient finetuning methods (PEFT, Han et al., 2024; Houlsby et al., 2019; Liu et al., 2022). It allows finetuning foundation models at a fraction of the cost required for full adaptation and quickly became a standard practice in research and industry (Mangrulkar et al., 2022). We focus on the popular family of selective approaches, where only a subset of parameters

[1]Noah's Ark Lab [2]IRISA [3]Univ. Rennes 2, Inria [4]L'Institut Agro Rennes-Angers. Correspondence to: Ambroise Odonnat <ambroise.odonnat@gmail.com>.

*Proceedings of the $43^{rd}$ International Conference on Machine Learning*, Seoul, South Korea. PMLR 306, 2026. Copyright 2026 by the author(s).

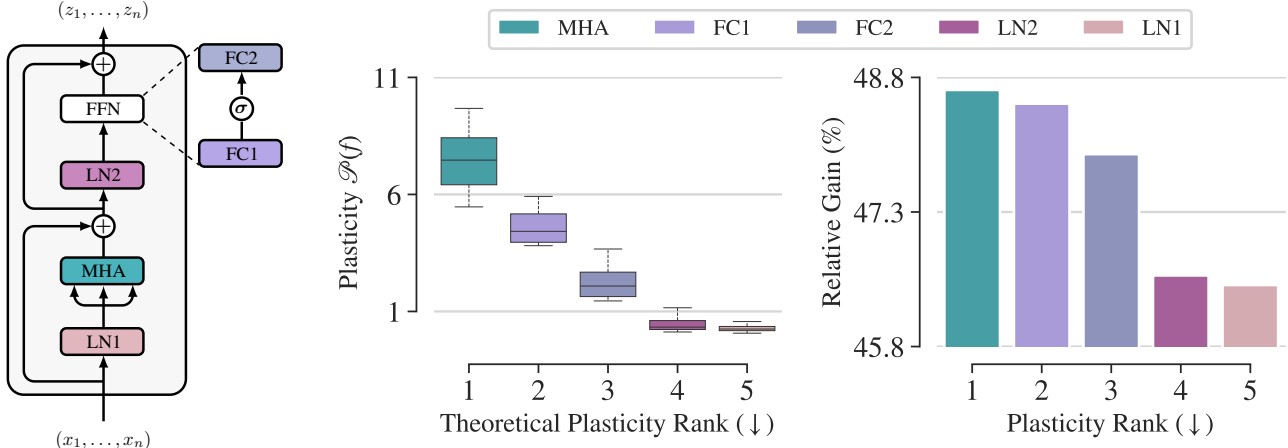

*Figure 2.* **Overview of our contributions.** We conduct a comprehensive analysis of vision transformer components (**left**) through the perspective of their plasticity (Definition 1). Our theoretical analysis allows us to rank modules in terms of their plasticity (Section 4). Experiments on large-scale ViTs support our theoretical insights (Section 5.1), as shown by the distribution of plasticity over all benchmarks (**middle**). Through large-scale finetuning runs on an 86M-parameter ViT (Section 5.2), we demonstrate the real-world benefits of plasticity. As showcased by the average relative gain, i.e, improvement over the linear probing accuracy, on a diverse set of 11 classification benchmarks (**right**), a higher plasticity yields greater finetuning benefits.

is updated during finetuning (Guo et al., 2019; Lee et al., 2019, 2023). Recent works *empirically* studied the benefits of adapting one type of transformer component across the whole network: the normalization layers (Zhao et al., 2024), the attention module (Touvron et al., 2022), or the feed-forward layers (Ye et al., 2023). However, little is known from a theoretical perspective about the adaptability of those modules[1]. This motivates us to ask:

> *Which transformer components should be*
> *prioritized during finetuning and why?*

**Our approach.** We focus on vision transformers (ViT, Dosovitskiy et al., 2021) and aim to reconcile the intrinsic functional properties of the individual components with the empirical performance observed when adapting them. To avoid confounders and since considering all possible combinations of the modules is computationally prohibitive, we conduct a systematic component-wise study where each type of module is finetuned in isolation. We build upon the intuition that promoting the smoothness of a neural network, e.g., by regularizing its Lipschitz constant (Newhouse et al., 2025), reduces its sensitivity to input perturbations (Rosca et al., 2020). While this is desirable for generalization (Krogh & Hertz, 1991; Neyshabur et al., 2017; Rosca et al., 2020), training stability (Zhai et al., 2023), or adversarial robustness (Miyato et al., 2018), it limits the degree of freedom a given component has to adapt its outputs to changes in the inputs, and thus its *plasticity*. As a result, it hinders the adaptation to downstream data during finetuning.

This motivates us to quantify the plasticity of transformer components as an average rate of change, where high values would indicate low smoothness (the formal definition shall come in Section 3).

**Our contributions.** We provide a theoretical ranking of vision transformer components in terms of plasticity, supported by empirical evidence. We demonstrate through comprehensive experiments that high plasticity consistently leads to better finetuning performance. Our main contributions, illustrated in Fig. 2, are:

1. **Intuitive measure:** We formalize the plasticity of a module as its average rate of change, which captures how it responds to variations in its input (Section 3).

2. **Theoretical analysis:** We establish a theoretical ranking among transformer components by deriving upper bounds on their plasticity (Section 4).

3. **Plasticity ranking:** We validate our theoretical insights on pretrained ViTs, showing that the attention module consistently has the highest plasticity, followed by the first and second feedforward layers, the LayerNorm preceding the feedforward, and finally the LayerNorm preceding the attention module. This ranking is not limited to ViTs, and holds for DINOv3 and GPT2 (Section 5.1).

4. **Finetuning benefits:** We conduct exhaustive finetuning runs of large-scale ViTs on a diverse set of classification benchmarks with both SGD and Adam optimizers. Our findings showcase that adapting modules with high plasticity, namely attention modules and the feedforward layers, results in higher and more stable performance across initialization and learning rates (Section 5.2).

---

[1]In what follows, we use the terms module and component interchangeably, always referring to normalization layers, multi-head attention modules, and feedforward layers.

Our work, supported both theoretically and empirically, provides a novel perspective on the role of smoothness in fine-tuning. We believe that the highlighted link between plasticity and gradient norms, illustrated in Fig. 1, will help guide the design of more efficient adaptation methods.

## 2. Background

Throughout the paper, we use the notation $[n]$ to represent the set $\{1, \ldots, n\}$. The Euclidean norm of $\mathbb{R}^n$ is denoted by $\|\cdot\|$ and its $\ell_\infty$-norm is denoted by $\|\cdot\|_\infty$. A sequence of tokens $x = (x_1, \ldots, x_n) \in (\mathbb{R}^d)^n$ can be seen as a matrix[2] in $\mathbb{R}^{d \times n}$ with Frobenius norm $\|x\|_F = (\sum_i^n \|x_i\|^2)^{1/2}$ and spectral norm $\|x\|_2 = \sigma_{\max}(x)$, with $\sigma_{\max}(x)$ the largest singular value of $x$. We denote by $B_r \subset \mathbb{R}^d$ the closed ball centered at 0 with radius $r > 0$.

**Neural network smoothness.** Formally, the smoothness of a function is related to the number of continuous derivatives it has on its domain. In deep learning, it can refer to several related concepts, such as differentiability, Lipschitz continuity, or robustness to input perturbations (Rosca et al., 2020). A common way to quantify smoothness is through the notion of Lipschitz continuity. A function $f \colon (\mathbb{R}^d)^n \to (\mathbb{R}^d)^n$ is said to be Lipschitz continuous if there exists a constant $K \leq 0$ such that for any pair of inputs $x, y \in (\mathbb{R}^d)^n$, we have $\|f(x) - f(y)\|_F \leq K\|x - y\|_F$. The smallest constant $K$ is called the Lipschitz constant of $f$, denoted by $\mathrm{Lip}(f)$, and writes $\mathrm{Lip}(f) = \sup_{x \neq y \in (\mathbb{R}^d)^n} \frac{\|f(x)-f(y)\|_F}{\|x-y\|_F}$.

**Vision transformers.** A ViT takes as input 2D images, embedded into sequences of tokens by splitting them into patches of size $P$, which are then flattened and linearly projected in $\mathbb{R}^d$. The architecture consists of a succession of transformer encoders. Akin to BERT (Devlin et al., 2019), a classification token CLS is prepended to the sequence of tokens to perform classification. A transformer encoder is illustrated in Fig. 2 (left), where the LayerNorms are denoted by LN1 and LN2, the attention module is denoted by MHA, and the feedforward linear layers are denoted by FC1 and FC2. After the last layer, the embedding of the CLS token is pooled to perform classification. Implementation details are given in Appendix D.1.

**Transformer components.** We recall below how each module operates on a sequence of tokens $x \in (\mathbb{R}^d)^n$.

- **LayerNorms**: A LayerNorm with weights $\gamma, \beta \in \mathbb{R}^d$ acts on each input token individually with the formula

$$f(x) = \left(\gamma \odot \frac{x_j - \mu(x_j)}{\sigma(x_j)} + \beta\right)_{1 \leq j \leq n} \in (\mathbb{R}^d)^n,$$

where $\odot$ is the element-wise product and $\mu(x_j), \sigma(x_j)$ are the mean and standard deviation of the token $x_j$.

- **Multi-head self-attention**: Let $H \in \mathbb{N}$ such that $k = \frac{H}{d}$ is an integer. Let $Q^h, K^h, V^h$ be matrices in $\mathbb{R}^{k \times d}$ and $O^h \in \mathbb{R}^{d \times k}$. A multi-head self-attention module with weights $(O^h, Q^h, K^h, V^h)_{1 \leq h \leq H}$ outputs

$$f(x) = \sum_{h=1}^H O^h f_{\mathrm{att}}^h(x) \in (\mathbb{R}^d)^n,$$

where the single-head self-attention $f_{\mathrm{att}}^h$ has weights $Q^h, K^h, V^h$ and writes

$$f_{\mathrm{att}}^h(x) = (V^h x) \cdot \mathrm{softmax}\left(\frac{(Q^h x)^\top K^h x}{\sqrt{k}}\right) \in (\mathbb{R}^k)^n,$$

with the softmax applied row-wise.

- **Feedforward linear layers**: A feedforward module with weights $W_1 \in \mathbb{R}^{d \times 4d}, W_2 \in \mathbb{R}^{4d \times d}$ combines two linear layers $x \mapsto W_1 x$ and $x \mapsto W_2 x$ with a GeLU to output

$$f(x) = W_2 \mathrm{GeLU}(W_1 x) \in (\mathbb{R}^d)^n.$$

## 3. Vision transformer plasticity

Regularizing the Lipschitz constant of a model is a common approach to encourage smoothness (Miyato et al., 2016; Newhouse et al., 2025). While it serves as a useful inductive bias in generalization (Bartlett et al., 2017; Sokolić et al., 2017), training stability (Miyato et al., 2018; Zhai et al., 2023), and adversarial robustness (Jia et al., 2024; Tsuzuku et al., 2018), too much smoothness can constrain the model's capacity and its adaptability to new tasks, as shown in Rosca et al. (2020). This motivates us to identify the components whose Lipschitz constants might be too small (see Rosca et al., 2020, Section 5), which could impact the adaptation during finetuning. This can be done by analyzing the rates of change of the components $f$ since they lower bound the Lipschitz constant via $\frac{\|f(x)-f(y)\|_F}{\|x-y\|_F} \leq \mathrm{Lip}(f)$.

**Plasticity measure.** Building upon this intuition, we formalize below the *plasticity*[3] of a module, i.e., its ability to adapt its output in response to changes in the inputs:

**Definition 1** (Plasticity). *Let $\nu$ be the uniform distribution over the set of distinct pairs of sequences of tokens in $(\mathbb{R}^d)^n$. We define the plasticity of a transformer*

---

[2]In the PyTorch implementation, all matrices are transposed because the input data is viewed as matrices in $\mathbb{R}^{n \times d}$ instead of the common $\mathbb{R}^{d \times n}$ we use.

[3]Akin to the neuroplasticity of the brain defined as its ability to change its activity "in response to intrinsic or extrinsic stimuli" (Puderbaugh & Emmady, 2023). The loss of plasticity at the network level has been studied in deep reinforcement learning (Lyle et al., 2023) or continual learning (Dohare et al., 2024).

*component f as*

$$\mathscr{P}(f) = \mathbb{E}_{(x,y)\sim\nu}\left[\frac{\|f(x) - f(y)\|_{\mathrm{F}}}{\|x - y\|_{\mathrm{F}}}\right]. \qquad (1)$$

Definition 1 ensures that, for any component $f$, we have $\mathscr{P}(f) \leq \mathrm{Lip}(f)$. The Lipschitz constant is a worst-case estimation that ensures control over each rate of change. To better capture the overall behavior of transformer components over the distribution of input sequences, we compute the average rate of change. This is reminiscent of the notion of average smoothness, defined for functions on a metric probabilistic space in Ashlagi et al. (2021). Two regimes of plasticity can be distinguished. If $\mathscr{P}(f) < 1$, the module $f$ contracts the input discrepancy on average. If $\mathscr{P}(f) > 1$, then $f$ amplifies the change in the input on average and pushes the value of $\mathrm{Lip}(f)$ from below. In the rest of this work, we will say that components in the first regime have low plasticity and are smooth, and that the components in the second regime have high plasticity and low smoothness (or are non-smooth, by abuse of language).

**Connection to finetuning.** The plasticity measure introduced in Definition 1 captures the sensitivity of transformer components to input changes. A high plasticity implies a high Lipschitz constant and thus low smoothness. For a given module $f$ with weights $\theta$, we have from Federer (1969) that $\|\nabla_x f\|_{\mathrm{F}} \leq \mathrm{Lip}(f)$. Let $\mathcal{L}$ be the finetuning loss. During gradient descent, the weights are updated following $\theta \leftarrow \theta - \eta\nabla_\theta\mathcal{L}$, which involves the gradient of $f$ with respect to the parameters via

$$\nabla_\theta\mathcal{L} = (\nabla_f\mathcal{L})\frac{\partial f}{\partial\theta},$$

using the Vector-Jacobian product notation (Béthune et al., 2024; Dagréou et al., 2024). Since our goal is to identify the components that adapt best to downstream data during finetuning, a natural question is: *what is the connection between the gradient with respect to inputs and the gradient with respect to the parameters?* On the theoretical side, Béthune et al. (2024) showed that these notions are two sides of the same coin. More precisely, the authors proved that regularizing the Lipschitz constant with respect to the inputs amounts to bounding the norm of the gradients with respect to the parameters (see Béthune et al., 2024, Theorem 1). As such, too much regularization on the smoothness can impact optimization; conversely, having looser Lipschitz constraints, e.g., thanks to high plasticity, might facilitate the learning process. This has been empirically observed in Newhouse et al. (2025, Section 4.4), where the authors show that reducing the Lipschitz constant negatively impacts performance of a 145M Lipschitz-constrained transformer on FineWeb (Penedo et al., 2024). In particular, matching the NanoGPT baseline (Jordan et al., 2024a) requires a Lipschitz constant of up to $10^{264}$.

**Expected benefits of plasticity.** The connection between input-output and weight-output smoothness hints at the role of plasticity in the learning process. We expect the components with high plasticity, i.e., the non-smooth ones, to allow large gradient norms during finetuning, thus leading to a faster and better adaptation (note that we do not expect a linear relationship with downstream performance, either, as is the case with unsupervised accuracy estimation methods (Deng et al., 2023; Garrido et al., 2023; Xie et al., 2024, 2025)). This process is illustrated in Fig. 1. This can be understood intuitively, with plastic components carrying more information about the downstream data than the smooth ones. Provided our insights are confirmed through experiments (Fig. 2 offers a sneak peek for impatient readers), our perspective would depart from the conventional wisdom that promoting smoothness is beneficial to learning (Miyato et al., 2018; Neyshabur et al., 2017; Rosca et al., 2020; Zhai et al., 2023).

## 4. Theoretical analysis

In this section, we derive upper bounds on the plasticity $\mathscr{P}(f)$. It allows us to compare transformer components in terms of plasticity. The proofs are given in Appendix B. We start with the LayerNorms whose upper bound is stated in the proposition below.

> **Proposition 1** (LayerNorm)**.** *Let $f$ be a LayerNorm with weights $\gamma, \beta \in \mathbb{R}^d$. Assume that all tokens in position $i \in [n]$ have the same mean $\mu_i$ and standard deviation $\sigma_i$ on $\mathbb{R}^d$ and let $\sigma > 0$ be the minimal standard deviation. Then, we have $\mathscr{P}(f) \leq \frac{1}{\sigma}\|\gamma\|_\infty$.*

The requirement on tokens comes from the fact that images are normalized with ImageNet1k statistics during preprocessing (Kolesnikov et al., 2020) and embedded into sequences of tokens with the same layer. This implies that the $\mu_i, \sigma_i$ depend only on the embedding layer. Having $\sigma_i = 0$ for some $i \in [n]$ would force all the tokens in position $i$ to be equal, independently of the embedded images. Since this is not the case, neither at initialization nor after pretraining, we must have $\sigma = \min_{i\in[n]}\sigma_i > 0$. We now proceed to bound the plasticity of the feedforward linear layers. This is reminiscent of the well-known upper bound on the Lipschitz constant of linear operators (Federer, 1969; Virmaux & Scaman, 2018).

> **Proposition 2** (Feedforward layer)**.** *Let $f$ be a feedforward linear layer with weights $W \in \mathbb{R}^{d\times 4d}$ (resp. $W \in \mathbb{R}^{4d\times d}$). Then, we have $\mathscr{P}(f) \leq \|W\|_2$.*

We now proceed to the upper bound for the multi-head self-attention module. Since self-attention is not globally

Lipschitz continuous (Kim et al., 2021), we need to restrict ourselves to sequences $(x_1, \ldots, x_n)$ in $B_r^n$, where $B_r \subset \mathbb{R}^d$ is the closed ball centered in 0 with a radius of $r$.

---

**Proposition 3** (Multi-head self-attention). *Let $f$ be a multi-head self-attention module with weights $(O^h, Q^h, K^h, V^h)_{1 \leq h \leq H}$. Let $A^h = (Q^h)^\top K^h / \sqrt{k}$ and $r > 0$. Assume that sequences of tokens are in $B_r^n$. Then, we have*

$$\mathscr{P}(f) \leq \sum_{h=1}^{H} \|O^h\|_2 \|V^h\|_2 \sqrt{3n + (12n+3)r^4 \|A^h\|_2^2}.$$

---

The setting of bounded tokens has been studied in Castin et al. (2024) and holds in practice (see Darcet et al., 2024, Fig. 4). This can be understood by the fact that images are normalized during preprocessing before being projected in $\mathbb{R}^d$ using a layer with bounded weights. As shown in Castin et al. (2024, Proposition 3.4), the bound in Proposition 3 is tight in terms of sequence length $n$. In a ViT, the average token norm is 20 (see Section 5.1) and the sequence length is around 200. Hence, $r$ and $\sqrt{n}$ have a similar order of magnitude. It leads to an effective growth rate of $r^2 \sqrt{n}$ in Proposition 3, since $\|A^h\|_2 \geq 1$ in practice (see Zhai et al., 2023, Fig. 3). Recalling that the total energy of a digital image is defined as the sum of its squared pixel intensities, the next corollary allows us to obtain a tighter bound with a growth rate in $\sqrt{n}$.

---

**Proposition 4** (Tighter upper bound). *Let $f$ be a multi-head self-attention module with weights $(O^h, Q^h, K^h, V^h)_{1 \leq h \leq H}$. Let $A^h = (Q^h)^\top K^h / \sqrt{k}$ and let $\alpha$ be the spectral norm of the embedding layer. Assume that sequences of tokens are obtained from images with a total energy bounded by $\mathcal{E} > 0$. Then, we have*

$$\mathscr{P}(f) \leq \sum_{h=1}^{H} \|O^h\|_2 \|V^h\|_2 \left( \sqrt{n} + \alpha^2 \mathcal{E} \|A^h\|_2 \right).$$

---

The assumption on input images, discussed in detail in Appendix B.4, holds in a standard signal processing setting (see, e.g., Goodman, 2005; Mallat, 2008); It allows us to bound the Frobenius norm of sequences of tokens. This is key to obtaining the growth rate $\sqrt{n}$ in Proposition 4, further improving Proposition 3. Note that the mean-field limit with $n \to +\infty$ (Castin et al., 2024; Geshkovski et al., 2023; Sander et al., 2022) is interesting from a mathematical perspective. In particular, it leads to upper bounds independent of the sequence length (Castin et al., 2024; Geshkovski et al., 2023). However, this setting is not suitable for vision transformers where $n$ is usually below $10^3$ (Dehghani et al., 2023; Dosovitskiy et al., 2021; Kolesnikov et al., 2020).

**Theoretical ranking.** To compare the modules, we focus on the relative order of their upper bounds. Propositions 1 and 2 imply that the bound over $\mathscr{P}(f)$ is tighter for the normalization than for the linear layers. Indeed, for a vector $\gamma \in \mathbb{R}^d$ and a matrix $W \in \mathbb{R}^{d \times m}$ with entries in a similar range, $\|\gamma\|_\infty$ is comparable to $\|W\|_\infty$, which is smaller than the spectral norm of $W$ since

$$\forall i, j, |W_{ij}| \leq \sqrt{\sum_{i=1}^{k} |W_{ij}|^2} = \|W e_j\| \leq \|W\|_2,$$

with $e_j \in \mathbb{R}^m$ has zero entry everywhere except in $j$-th position, where we used the fact that the spectral norm is the operator norm induced by the Euclidean norm. A similar analysis can be done for the multi-head self-attention module: since spectral norms are above 1 in practice (see Zhai et al., 2023, Fig. 3), the sum over the heads of products of spectral norms and the dependency in the sequence length $n$ of Propositions 3 and 4 imply a looser control over the plasticity of the multi-head self-attention module compared to the LayerNorms and the feedforward. We validate our insight by numerically computing the upper bounds on an 86M pretrained ViT with sequence length $n = 197$ and 12 attention heads; see Appendix E.1.1 for details. In Fig. 7, we can see the ranking between modules: the multi-head self-attention module has the highest upper bound, followed by the feedforward linear layers, and then the LayerNorms. The conclusion of the theoretical analysis is the following:

---

**Takeaway 1.** Our analysis suggests the following plasticity ranking: MHA $\to$ FC1 $\approx$ FC2 $\to$ LN2 $\approx$ LN1.

---

**Extension to large language models.** The results presented in Propositions 1 to 3 also hold for decoder-only models such as large language models (Grattafiori et al., 2024; Kamath et al., 2025; Radford et al., 2019, LLMs). Indeed, decoder blocks (Vaswani et al., 2017) have the same global structure as encoder blocks and differ only at the attention module level, which becomes causal. Fortunately, Proposition 3 is still verified for masked self-attention thanks to Theorem 4.3 in Castin et al. (2024). We will show in Section 5.1 that ViTs and LLMs have similar plasticity patterns.

## 5. Experiments

In this section, we experimentally show that (a) the plasticity of vision transformer components follows the ranking predicted by our theory (Section 5.1) and (b) components with high plasticity lead to better and more stable finetuning accuracy across initializations and learning rates (Section 5.2). Our code is available at github.com/ambroiseodt/vit-plasticity, and reproducibility details are given in Appendix C.

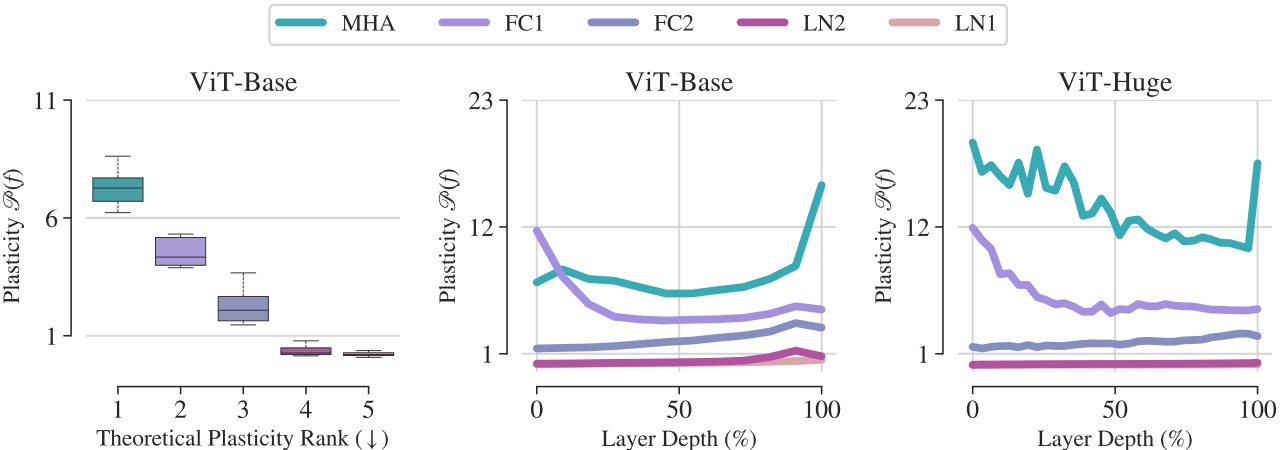

*Figure 3.* **Plasticity analysis on Sketch.** The distribution of rates of change $\|f(x) - f(y)\|_{\mathrm{F}}/\|x - y\|_{\mathrm{F}}$ on ViT-Base (**left**) follows the theoretical ranking of Section 4. We observe along transformer blocks of ViT-Base (**middle**) that the attention module has the highest plasticity $\mathscr{P}(f)$, followed by the first and second linear layers of the feedforward. The LayerNorms are the most rigid, with a plasticity below 1. The same pattern is obtained on ViT-Huge (**right**), where the higher attention plasticity further validates our theory (see Proposition 3) since the sequence length $n$ is larger than with ViT-Base.

**Experimental setup.** Unless otherwise specified, we conduct our experiments on large-scale ViTs of varying sizes (86M, 307M, and 632M parameters), pretrained on ImageNet22k (Deng et al., 2009) (see Appendix D.1 for details). We perform both the plasticity and finetuning studies on a diverse set of 11 commonly used classification benchmarks: Cifar10, Cifar100 (Krizhevsky, 2009); 5 variants from Cifar10-C (Hendrycks & Dietterich, 2019): Contrast, Gaussian Noise, Motion Blur, Snow, Speckle Noise; 2 domains from DomainNet (Peng et al., 2019): Clipart, Sketch; Flowers102 (Nilsback & Zisserman, 2008) and Pets (Parkhi et al., 2012). Images are resized to $224 \times 224$ resolution and preprocessed following the protocol of Dosovitskiy et al. (2021), see Appendix D.2 for details.

### 5.1. Plasticity analysis

In this section, we compute the plasticity measure introduced in Definition 1 on pretrained vision transformers. To allow for diverse discrepancies $\|x - y\|_{\mathrm{F}}$, the sequences $x$ are obtained by embedding 12800 pretraining images from ImageNet1k (Deng et al., 2009), and the sequences $y$ are obtained similarly on various downstream data. The full experimental details are provided in Appendix D.3. Results on Sketch are displayed in Fig. 3, and additional results on all benchmarks are given in Appendix E.1.2.

**Empirical ranking.** In Fig. 3 (left), we display, for each module $f$ of ViT-Base, the distribution of rates of change $\|f(x) - f(y)\|_{\mathrm{F}}/\|x - y\|_{\mathrm{F}}$ on Sketch. We can see that the ranking established in Section 4 correctly predicts the practical behavior of transformer components. In addition, we observe that the first feedforward layer has larger rates of change than the second. Despite being closer, the Layer-

Norms also exhibit distinct plasticity, with the LayerNorm preceding the feedforward layer being less rigid than the one preceding the attention module. Our findings are consistent across all benchmarks, as can be seen in the overall distribution of plasticity displayed in Fig. 2 (middle) and in Figs. 8 to 17. It allows us to refine the ordering established in Section 4 into:

$$\text{MHA} \to \text{FC1} \to \text{FC2} \to \text{LN2} \to \text{LN1}.$$

In the rest of this work, this ordering will define the *plasticity rank* of each component. In Fig. 3 (middle), the evolution of the plasticity $\mathscr{P}(f)$ over the layers of ViT-Base is displayed. The $x$-axis represents the layer depth, denoted as a percentage of the overall depth. The ordering previously mentioned is respected. We can also see the two regimes of plasticity mentioned in Section 3: the attention module and the feedforward linear layers have a high plasticity with values $\mathscr{P}(f) > 1$. In contrast, the LayerNorms have a low plasticity $\mathscr{P}(f) < 1$. Following our terminology, this implies that the attention modules and feedforward layers are non-smooth, contrary to the LayerNorms.

**Remark 5.1** (Smooth normalization layers). *The smoothness and low plasticity of normalization layers can be explained by the fact that, by design, they limit the propagation of perturbations in the input by rescaling the features. This has notably been leveraged in prior works to mitigate the non-stationarity in time series (Kim et al., 2022). As we will see in Section 5.2, this is not a desirable property to adapt to downstream data.*

**Impact of the sequence length.** We further confirm the theoretical insights of Section 4 by conducting a similar plasticity analysis on ViT-Huge, which has a longer sequence length $n = 257$. The results are displayed in Fig. 3

(right). We observe a similar evolution along the depth, with a larger plasticity for the attention module than with ViT-Base. This can be explained by the dependency on the sequence length $n$ in the attention upper bound of Proposition 3. Our findings are consistent across all benchmarks, as displayed in Figs. 8 to 17. This showcases, as hinted by the upper bounds of Section 4, that plasticity is an intrinsic property of the components and their weights. The conclusion of the plasticity analysis is:

> **Takeaway 2.** The empirical *plasticity ranking* of vision transformer modules supports our theoretical insights with: MHA → FC1 → FC2 → LN2 → LN1.

**Beyond supervised vision transformers.** We extend our analysis to other pretraining paradigms and transformer architectures. We consider DINOv3 (Siméoni et al., 2025), a 7B-parameter vision transformer trained in a self-supervised fashion, and GPT2 (Radford et al., 2019), a 124M-parameter decoder-only language model. We follow a similar setup as before, with pretraining and downstream data taken as ImageNet-22k and Cifar10 for DINOv3, and AGNews (Zhang et al., 2015) and WikiText-103 (Merity et al., 2017) for GPT2; details on the choice of datasets are given in Appendix E.1.3). We display the distribution plasticity on DINOv3 and GPT2 in Fig. 4 (see Fig. 18 for the evolution across layers). The observed patterns are consistent with those of supervised ViTs shown in Fig. 3, showcasing the generality of our findings across types of pretraining and transformer architectures.

**Remark 5.2** (Impact of distillation and alignment). *Since the plasticity of transformer components heavily depends on the model's weights (see Section 4), different values may be observed for distilled or instruct-version models.*

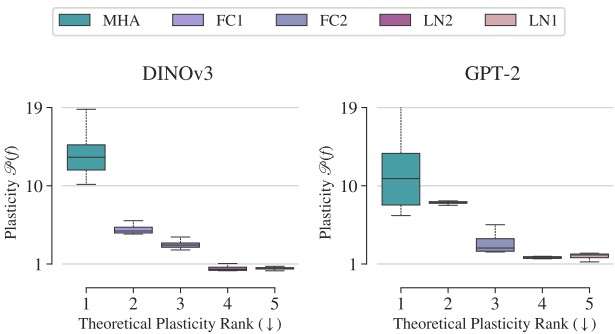

*Figure 4.* **Plasticity analysis on DINOv3 and GPT2.** The distribution of rates of change $\|f(x) - f(y)\|_{\mathrm{F}}/\|x - y\|_{\mathrm{F}}$ on DINOv3 (**left**) and GPT2 (**right**) follows the theoretical ranking of Section 4: the attention module has the highest plasticity $\mathscr{P}(f)$, followed by the first and second linear layers of the feedforward. The LayerNorms are the most rigid, with a plasticity below or near 1. This behavior is consistent with pretrained ViTs shown in Fig. 3.

## 5.2. Benefits of plasticity for finetuning

In this section, each transformer component is finetuned in isolation along the depth of ViT-Base, leading to the 5 configurations in Table 5. The optimization is done with SGD following the protocol of Dosovitskiy et al. (2021) summarized in Table 7. We conduct a sweep over 4 well-spaced learning rates to ensure a fair comparison of modules with different numbers of trainable parameters. Each experiment is done over 3 seeds, leading to a total of $\sim 1000$ finetuning runs. The experimental details are given in Appendix D.4.

**Better performance.** The overall performance on all the 11 benchmarks is displayed in Table 1 (full results in Table 8, deferred to Appendix E.2.1 due to space constraints). We observe that the attention modules and feedforward layers with high plasticity lead to enhanced finetuning performances, surpassing the LayerNorms on most benchmarks. The benefit of plasticity is even more salient on challenging datasets such as Cifar100, Clipart, and Sketch, where MHA, FC1, and FC2 surpass LN1 and LN2 by a large margin. We can see that the performance ordering is aligned with the *plasticity ranking* from Section 5.1: the attention modules and feedforward layers result in higher accuracy than the LayerNorms. These results are consistent with Fig. 2 (right), where we display the relative gain, i.e., the percentage improvement of the finetuning accuracy over the linear probing accuracy. We can see that the ranking is also respected among components of the same size, namely the attention modules and the feedforward layers on the one hand, and the LayerNorms on the other hand. This conclusion holds at a larger scale, as shown on Clipart in Table 9 with ViT-Base (86M parameters) and ViT-Large (307M parameters).

*Table 1.* **Better finetuning performance (11 benchmarks)**. We report the average top-1 accuracy (%) on the test set over 11 diverse image classification benchmarks (↑). Transformer components are ordered in decreasing order of plasticity. The best performance among components is in **bold**, non-smooth components are highlighted in gray, and underlined entries indicate a statistically significant difference with MHA according to a Wilcoxon signed-rank test at a confidence level of 5%. Full results are in Table 8.

| configuration | MHA | FC1 | FC2 | LN2 | LN1 |
|---|---|---|---|---|---|
| *accuracy* | **90.8** | 90.7 | 90.3 | 89.9 | 89.8 |

**A strong baseline.** We can also see in Table 1 that, except for FC1, the improvement of finetuning the MHA is statistically significant compared to the other modules. The adaptability of the attention module to downstream data is reminiscent of Touvron et al. (2022, Section 4), where the authors found that tuning the attention module alone can be beneficial for ViTs models of varying sizes, ranging from 6M to 340M parameters; it notably surpasses the full finetuning baseline on small datasets. This showcases the

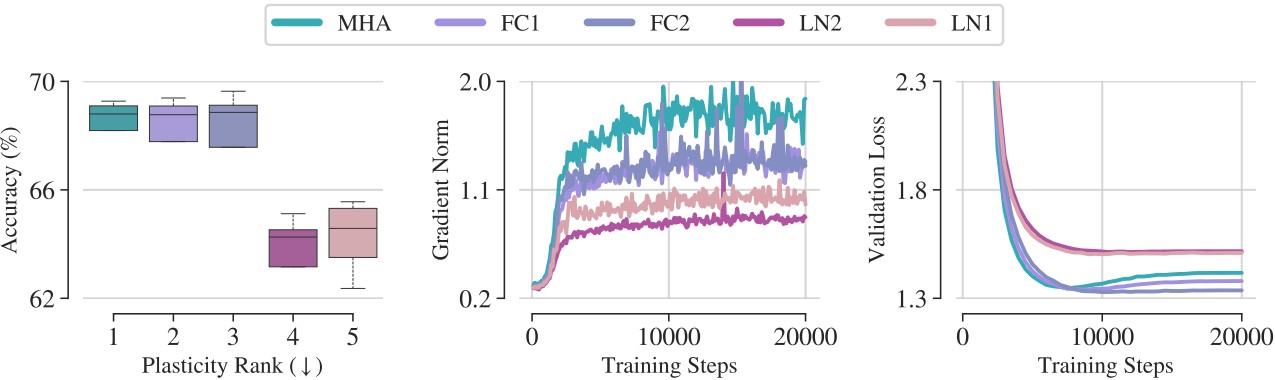

*Figure 5.* **Benefits of plasticity with SGD on Sketch.** Transformer components are ordered in terms of decreasing plasticity. We can see that the performance across learning rates and seeds (**left**) is better and more stable for plastic components. This can be understood by looking at the evolution of the gradient norms (**middle**) and the validation loss (**right**) throughout training: we can see that the higher plasticity, the larger gradient norms, and the better the generalization.

relevance of our plasticity analysis in understanding the behavior of vision transformers. Another interesting insight of our work is that single-component finetuning is competitive with parameter-efficient methods such as LoRA (Hu et al., 2022). For instance, finetuning only the LayerNorms with Adam outperforms LoRA *with* $15\times$ *fewer trainable parameters* on Cifar100: $89.4\%$ with 28K parameters (see Table 2) compared to $88.1\%$ with 400K parameters (see (Bafghi et al., 2024, Table 1)). As a byproduct, it also shows that parameter count alone does not determine the performance.

**Robust adaptation.** The absolute best performance is not the only factor to take into account: being robust to the choices of hyperparameters is also of great importance for practitioners. The main source of variability during finetuning comes from the initialization and the learning rate. In Fig. 5 (left), we report the distribution of top-1 accuracy over the grid of learning rates (see Table 7) and 3 seeds. We can see that the finetuning performance of the components with high plasticity is steadier. In particular, the attention module has the smallest variability. This pattern remains consistent overall, notably for the multi-head self-attention, as can be seen in Fig. 20. Our findings hint at the benefits of plasticity during the optimization process discussed in the next paragraph.

**Interplay between plasticity and optimization.** In Section 1, we argued that, given the interplay between input-output smoothness and weight-output smoothness, a high plasticity should lead to large gradient norms. In Fig. 5, we display the evolution of the gradient norms (middle) and the validation loss (right) for the finetuning run on Sketch that achieves the highest accuracy (this corresponds to learning rate $\eta = 1e{-}2$). This confirms our intuition: the ordering of gradient norms is aligned with the *plasticity ranking* established in Section 5.1. In tandem, the loss descent is steeper for components with high plasticity, such as the at-

tention modules and the feedforward layers. These patterns are consistent across benchmarks, learning rates, and seeds (see Figs. 21 to 53), which confirms the role of plasticity during the optimization process illustrated in Fig. 1. Our findings are reminiscent of the intuition that ResNet layers with larger gradient magnitudes carry more information about the target data (see Lee et al., 2023, Section 4). The benefits of plasticity on the learning process are in accordance with the empirical evidence from Fig. 2 (right). The conclusion of the finetuning analysis can be summarized as:

> **Takeaway 3.** A higher plasticity facilitates the optimization and leads to better and more stable finetuning performance. Our findings indicate that the components to prioritize during finetuning should be the attention module and the first feedforward linear layer.

**Extension to Adam.** We extend the finetuning analysis with Adam (Kingma & Ba, 2014) (more precisely with its decoupled weight-decay version from Loshchilov & Hutter (2019)), which became the de facto choice to train foundation models (Siméoni et al., 2025), notably LLMs (Grattafiori et al., 2024; Orvieto & Gower, 2026). We follow the same setup as with SGD and rescale learning rates by $10^{-2}$ as is standard in the literature on adaptive methods (Dosovitskiy et al., 2021; Kumar et al., 2023; Touvron et al., 2021). Full implementation details are given in Appendix E.2.5. We observe similar benefits than with SGD: finetuning non-smooth components consistently yields higher accuracy and superior stability across learning rates compared to smoother modules. The robust adaptation and the interplay between plasticity and optimization are similar between Adam and SGD, as illustrated in Fig. 54 on Sketch, where the behavior is consistent with Fig. 5.

**Extension to low-data regimes.** We extend the finetuning experiments to low-data regimes by following the standard

*Table 2.* **Consistent benefits with Adam**. We report the average top-1 accuracy (%) on the test set over the learning rate grid of each dataset (↑). Transformer components are ordered in decreasing order of plasticity. The best performance among components is in **bold** and non-smooth components are highlighted in gray.

| configuration | MHA | FC1 | FC2 | LN2 | LN1 |
|---|---|---|---|---|---|
| Cifar100 | 91.0 $_{\pm0.2}$ | **91.3** $_{\pm0.6}$ | 90.6 $_{\pm1.4}$ | 89.4 $_{\pm2.7}$ | 88.4 $_{\pm3.2}$ |
| Motion Blur | **92.4** $_{\pm0.2}$ | 91.8 $_{\pm0.4}$ | 90.7 $_{\pm0.9}$ | 91.7 $_{\pm2.4}$ | 90.7 $_{\pm2.4}$ |
| Clipart | **76.4** $_{\pm0.5}$ | 74.8 $_{\pm0.8}$ | 75.8 $_{\pm0.3}$ | 73.3 $_{\pm1.9}$ | 73.5 $_{\pm1.5}$ |
| Sketch | 69.0 $_{\pm0.5}$ | 67.9 $_{\pm0.6}$ | **69.2** $_{\pm0.4}$ | 63.1 $_{\pm2.9}$ | 64.0 $_{\pm2.6}$ |
| Average | **82.2** | 81.4 | 81.7 | 79.4 | 79.2 |

*Table 3.* **Consistent benefits in low-data regimes.** Study akin to Table 2 but finetuning is done on only 1,000 samples per dataset (VTAB protocol). Transformer components are ordered in decreasing order of plasticity. The best performance among components is in **bold** and non-smooth components are highlighted in gray.

| configuration | MHA | FC1 | FC2 | LN2 | LN1 |
|---|---|---|---|---|---|
| Contrast | **96.7** $_{\pm0.7}$ | 96.1 $_{\pm1.1}$ | 95.4 $_{\pm1.1}$ | 95.8 $_{\pm1.1}$ | 95.6 $_{\pm1.3}$ |
| Motion Blur | **93.6** $_{\pm1.2}$ | 92.7 $_{\pm12.3}$ | 92.0 $_{\pm2.4}$ | 91.8 $_{\pm2.5}$ | 90.7 $_{\pm2.7}$ |
| Pets | **92.5** $_{\pm0.3}$ | 92.4 $_{\pm0.7}$ | 92.2 $_{\pm0.8}$ | 92.5 $_{\pm0.9}$ | 92.1 $_{\pm0.9}$ |
| Snow | **94.9** $_{\pm0.6}$ | 94.1 $_{\pm1.9}$ | 93.4 $_{\pm2.1}$ | 93.7 $_{\pm1.8}$ | 93.4 $_{\pm1.9}$ |
| Average | **94.4** | 93.8 | 93.3 | 93.4 | 93.0 |

Visual Task Adaptation Benchmark protocol (Zhai et al., 2020, VTAB), where training is restricted to 1,000 samples per dataset. Following the original ViT paper setup (Dosovitskiy et al., 2021), we finetuned models for 2,500 steps on 1,000 training samples. Results are displayed in Table 3. We can see that prioritizing high-plasticity components, especially the attention module MHA, yields overall superior performance compared to low-plasticity components (LN1, LN2). This remains consistent with the full-dataset setting.

## 6. Related work

**Smoothness.** Smoothness has been studied extensively in deep learning, e.g., in generalization (Bartlett et al., 2017; Jukić & Šnajder, 2025; Rosca et al., 2020), training stability (Zhai et al., 2023), generative modeling (Miyato et al., 2016; Szegedy et al., 2014), adversarial robustness (Hein & Andriushchenko, 2017; Jia et al., 2024; Tsuzuku et al., 2018; Weng et al., 2018), and differential privacy (Béthune et al., 2024). Common practices in deep learning, such as weight decay (Hanson & Pratt, 1988), dropout (Srivastava et al., 2014), and early stopping (Hardt et al., 2016), encourage smoothness. We extend this discussion in Appendix A. In our work, we identify the components with low smoothness and showcase the benefits of "non-smooth" components for finetuning. However, we do not promote smoothness during the learning process in any way.

**Lipschitz constant estimation.** Estimating the Lipschitz constant of neural networks is a hard problem (Virmaux & Scaman, 2018). Theoretical bounds are often loose, except for simple blocks such as linear maps and activation (Béthune et al., 2024). For transformers, the non-linear nature of self-attention makes the estimation more involved. Notably, Kim et al. (2021) showed that vanilla attention is not globally Lipschitz. Tight upper bounds have been obtained when restricted to sequences of bounded tokens Castin et al. (2024). Imposing Lipschitz constraints is a common way to promote smoothness (Newhouse et al., 2025; Rosca et al., 2020). Note that the proof techniques to derive the upper bounds in Section 4 are akin to those used to bound the Lipschitz constant of the modules.

**Parameter-efficient finetuning.** There exists a plethora of PEFT methods (Zhang et al., 2025). Our work is in line with the selective approaches, common in vision models, where only a subset of the parameters is finetuned (Guo et al., 2019; Lee et al., 2019, 2023; Liu et al., 2021; Wang et al., 2021; Xu et al., 2021). Another widely used category consists of additive methods, where small adapters, such as normalization layers, are inserted in the model (Houlsby et al., 2019; Lian et al., 2022; Pfeiffer et al., 2021). The well-known LoRa (Hu et al., 2022) method belongs to the reparameterization methods, where the weights are decomposed and reparameterized to adapt to fewer parameters. While those approaches are performance-oriented and often tune several types of modules together, we conduct a component-wise analysis with the aim of theoretically understanding the adaptability of each transformer module.

## 7. Discussion

This paper investigates the plasticity of vision transformer components by analyzing their average smoothness. Our theoretical and empirical analysis demonstrate the benefits of this approach to identify the components to prioritize during finetuning. In particular, finetuning non-smooth components (with high plasticity), namely the attention modules and the feedforward layers, consistently results in better and more stable performance. Our findings offer a novel perspective on the role of smoothness in finetuning transformers. We hope our work can help the design of more efficient adaptation methods and contribute to the effort towards better understanding the transformer architecture (see, e.g., Jelassi et al., 2022; Raghu et al., 2021; Von Oswald et al., 2023; Zekri et al., 2025).

**Limitations and future work.** To extend our analysis, a promising approach would be to study the effect of a tailored optimization (e.g., adaptive learning rates and scheduler) on the performance of each module. Moreover, since our methodology and theoretical insights naturally extend to decoder-only models, our paper could serve as groundwork to explore the adaptability and plasticity of LLMs.

## Acknowledgements

The authors would like to thank Zehao Xiao, Abdelhakim Benechehab, Vasilii Feofanov, and Albert Thomas for insightful comments about early versions of this work, as well as Théo Moutakanni and Guillaume Carlier for fruitful discussions that led to this project. The authors would also like to thank the anonymous reviewers and the area chair for their time and constructive feedback that helped us improve our work.

## Impact statement

This paper presents work whose goal is to advance the field of Machine Learning. There are many potential societal consequences of our work, none of which we feel must be specifically highlighted here.

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

# Appendix

**Roadmap.**    In this appendix, we discuss additional related work in Appendix A. We detail the proofs of our theoretical results in Appendix B. The open-source code and carbon footprint of the project are given in Appendix C. We provide the full implementation details in Appendix D and, finally, extensive additional experiments are provided in Appendix E. We display the corresponding table of contents below.

# Table of Contents

# A. Extended related work

In this section, we extend the discussion of prior works related to our paper.

**Smoothness in neural networks.**  Neural networks smoothness, typically quantified via Lipschitz constants and spectral norms, has been studied in the context of in-domain generalization (Bartlett et al., 2017; Jukić & Šnajder, 2025; Luxburg & Bousquet, 2004; Neyshabur et al., 2017; Novak et al., 2018; Rosca et al., 2020; Sokolić et al., 2017), training stability (Miyato et al., 2018; Zhai et al., 2023), generative modeling (Miyato et al., 2016; Szegedy et al., 2014), adversarial robustness (Anil et al., 2019; Hein & Andriushchenko, 2017; Jia et al., 2024; Rosca et al., 2020; Tsuzuku et al., 2018; Weng et al., 2018) and differential privacy (Béthune et al., 2024). Neyshabur et al. (2017) discusses the interplay between complexity measures based on norms, margin control, Lipschitz constants, and sharpness. These works discuss the benefits of promoting smoothness by regularizing Lipschitz constants (Rosca et al., 2020) or spectral norms (Zhai et al., 2023) during training. Common practices in deep learning such as weight decay (Bartlett, 1996; Hanson & Pratt, 1988; Krogh & Hertz, 1991), dropout (Srivastava et al., 2014) and early stopping (Hardt et al., 2016) encourage neural networks smoothness.

**Smoothness at scale.**  Recently, smoothness has been studied in the context of stabilizing large models, such as LLMs (Zhai et al., 2023). During training at such a large scale, many instabilities appear and notable loss spikes (Chowdhery et al., 2023; Hernández-Cano et al., 2025; Marin, 2025). They can cause the model to diverge and have been the subject of many studies. Inspired by mechanistic interpretability (Elhage et al., 2021), companion work proposed to study the gradient descent of the transformer (Odonnat et al., 2024, 2025) on the sparse modular addition problem (Nanda et al., 2023). It provides a simple yet sufficient testbed to observe involved optimization dynamics at a small scale. In more realistic settings, methods to control the gradient norms have been proposed, such as QK-Norm (Dehghani et al., 2023; Wortsman et al., 2024), or constraining the representation space on the hypersphere (Loshchilov et al., 2025), and are used to train industry-level LLMs. These approaches are reminiscent of generalization bounds based on margins and spectral norms (Bartlett et al., 2017; Neyshabur et al., 2017). Recently, the Muon optimizer (Jordan et al., 2024b), that normalize the spectral norms of the weights, has shown tremendous benefits in solving training instabilities. In addition, Newhouse et al. (2025) that Muon allows for optimizing Lipschitz-constrained neural networks at scale. The combination of these approaches can also be beneficial: the Kimi team managed to train on over 1T tokens without any loss spikes thanks to MuonClip (Kimi Team et al., 2025) that combines Muon with QK-Norm.

**Lipschitz constant estimation.**  A lot of effort has been put into estimating the Lipschitz constants of neural networks. While linear and activation layers have a known tight Lipschitz constant (Béthune et al., 2024; Castin et al., 2024; Virmaux & Scaman, 2018), estimating the Lipschitz constant of feedforward networks is NP-hard (Virmaux & Scaman, 2018) with loose theoretical upper bounds (Virmaux & Scaman, 2018). The non-linear nature of the self-attention module makes the estimation of its Lipschitz constant more involved. Kim et al. (2021) showed that the vanilla attention is not globally Lipschitz, while the dot-product attention is. Tight upper bounds on the attention module restricted to sequences of bounded tokens have been provided in Castin et al. (2024). Dasoulas et al. (2021) showed the benefits of enforcing Lipschitz continuity in self-attention for graph neural networks. Imposing Lipschitz constraints on neural networks has also been done in generative modeling (Arjovsky et al., 2017) or to ensure robustness and explainability, e.g., with 1-Lipschitz neural networks (Serrurier et al., 2021, 2023).

**Parameter-efficient finetuning.**  PEFT methods can be categorized into 5 main families (Zhang et al., 2025). Selective methods, common in vision models, aim to finetune only a subset of the parameters (Guo et al., 2019; Lee et al., 2019, 2023; Liu et al., 2021; Wang et al., 2021; Xu et al., 2021). Additive methods insert small adapter networks between the model's layers to be trained during finetuning (Houlsby et al., 2019; Lian et al., 2022; Pfeiffer et al., 2021). Prompt methods, commonly used for large language models (Gu et al., 2022; Li & Liang, 2021; Liu et al., 2023), involve learning soft commands to guide the model. Reparameterization methods, such as LoRa (Hu et al., 2022) and its companions (Gu et al., 2023; Zi et al., 2023), decompose and reparameterize the model's weights to adapt fewer parameters. These methods can be combined, leading to the last family of hybrid approaches (Mao et al., 2022). We note that the recent shift of large language models from static predictors towards dynamic, context-aware agents redefines finetuning methods. The benefits of learning to use tools instead of incorporating the knowledge in the model weights have been demonstrated in (Houliston et al., 2025). This explain the superiority and scalability of approaches such as Toolformer (Schick et al., 2023), Retrieval-Augmented Generation (RAG, Lewis et al., 2020) and a plethora of other approaches (Qu et al., 2025).

# B. Proofs

In this section, we detail the proofs of our theoretical results, which involve simple manipulations of matrix norms.

**Notations.** Throughout the paper, we use the notation $[n]$ to represent the set $\{1, \ldots, n\}$. The Euclidean norm of $\mathbb{R}^n$ is denoted by $\| \cdot \|$ and its $\ell_\infty$ norm is denoted by $\| \cdot \|_\infty$. Entries of a matrix $A \in \mathbb{R}^{n \times m}$ write $A_{ij}$, its rows write $A_i$ and its columns write $A_{\cdot,j}$. The Frobenius norm of a matrix $A \in \mathbb{R}^{n \times m}$ writes $\|A\|_{\mathrm{F}} = \left( \sum_{i=1}^n \sum_{j=1}^m A_{ij}^2 \right)^{1/2}$ and its spectral norm writes $\|A\|_2 = \sigma_{\max}(A)$, defined as the largest singular value of $A$. We denote by $B_r \subset \mathbb{R}^d$ the closed ball centered at 0 with radius $r > 0$.

**Useful properties.** The next lemma recalls some well-known properties of the Frobenius norm and its connection with the spectral norms (see Horn & Johnson, 2012, p.364, Section 5.6.P20), which will be used in our proofs.

**Lemma 1.** *For any matrices $A \in \mathbb{R}^{n \times m}$ and $B \in \mathbb{R}^{m \times p}$, we have*

$$\begin{cases} \|AB\|_{\mathrm{F}} \leq \|A\|_{\mathrm{F}} \|B\|_{\mathrm{F}} \\ \|AB\|_{\mathrm{F}} \leq \|A\|_2 \|B\|_{\mathrm{F}} \\ \|AB\|_{\mathrm{F}} \leq \|A\|_{\mathrm{F}} \|B\|_2, \end{cases}$$

*where the first property is referred to as the submultiplicativity of the Frobenius norm.*

*Proof.* Let $C = AB$. The entries of $C$ writes $C_{ij} = \sum_{k=1}^m A_{ik} B_{kj} = A_i^\top B_{\cdot,j}$. Applying Cauchy-Schwartz leads to:

$$\|C_{ij}\|^2 = \|A_i^\top B_{\cdot,j}\|^2 \leq \|A_i\|^2 \|B_{\cdot,j}\|^2.$$

Hence, the Frobenius norm of $AB$ verifies

$$
\begin{aligned}
\|AB\|_{\mathrm{F}}^2 = \|C\|_{\mathrm{F}}^2 &= \sum_{i=1}^n \sum_{j=1}^p \|C_{ij}\|^2 \\
&\leq \sum_{i=1}^n \sum_{j=1}^p \|A_i\|^2 \|B_{\cdot,j}\|^2 \\
&= \sum_{i=1}^n \|A_i\|^2 \sum_{j=1}^p \|B_{\cdot,j}\|^2 \\
&= \sum_{i=1}^n \sum_{k=1}^m A_{ik}^2 \sum_{j=1}^p \sum_{l=1}^m B_{lj}^2 \\
&= \sum_{i=1}^n \sum_{k=1}^m A_{ik}^2 \sum_{l=1}^m \sum_{j=1}^p B_{lj}^2 \\
&= \|A\|_{\mathrm{F}}^2 \|B\|_{\mathrm{F}}^2.
\end{aligned}
$$

Taking the square root concludes the proof by monotonicity. For the second result, using the same notations, we recall that the columns of $C$ write

$$C_{\cdot,j} = A(B_{\cdot,j}).$$

Recalling that the spectral norm $\| \cdot \|_2$ is the operator norm induced by $\| \cdots \|$ on $\mathbb{R}^n$, it leads to

$$
\begin{aligned}
\|AB\|_{\mathrm{F}} = \|C\|_{\mathrm{F}} &= \sum_{i=1}^{n} \sum_{j=1}^{p} C_{ij}^2 \\
&= \sum_{j=1}^{p} \|C_{\cdot,j}\|^2 \\
&= \sum_{j=1}^{p} \|A(B_{\cdot,j})\|^2 \\
&\leq \sum_{j=1}^{p} \|A\|_2^2 \|(B_{\cdot,j})\|^2 \qquad\qquad \text{(operator norm property of } \| \cdot \|_2) \\
&= \|A\|_2^2 \sum_{j=1}^{p} \|B_{\cdot,j}\|^2 \\
&= \|A\|_2^2 \sum_{j=1}^{p} \sum_{l=1}^{m} B_{lj}^2 \\
&= \|A\|_2^2 \sum_{l=1}^{m} \sum_{j=1}^{p} B_{lj}^2 \\
&= \|A\|_2^2 \|B\|_{\mathrm{F}}^2 .
\end{aligned}
$$

Taking the square root concludes the proof by monotonicity. For the last result, we use the previous one by using the fact that the Frobenius norm is the sum of singular values, which are invariant by transposition, and that the spectral norm is the maximal singular value. This leads to the Frobenius and the spectral norm to remain invariant under transposition. As such, we have

$$
\|AB\|_{\mathrm{F}} = \|(AB)^\top\|_{\mathrm{F}} = \|B^\top A^\top\|_{\mathrm{F}} \leq \|B^\top\|_2 \|A^\top\|_{\mathrm{F}} = \|A\|_{\mathrm{F}} \|B\|_2 ,
$$

which concludes the proof. $\qquad\square$

### B.1. Proof of Proposition 1

*Proof.* We start by upper-bounding the plasticity of LayerNorms. Let $f$ be a LayerNorm with weights $\gamma, \beta \in \mathbb{R}^d$ and let $\nu$ be the uniform distribution over the set of distinct pairs of sequences of tokens in $(\mathbb{R}^d)^n$. Let $(x, y)$ be a pair of two distinct sequences of tokens sampled according to $\nu$. By assumption, we have for any $i \in [n]$ that

$$
\forall 1 \leq i \leq n, \quad \mu(x_i) = \mu(y_i) = \mu_i \text{ and } \sigma(x_i) = \sigma(y_i) = \sigma_i > 0,
$$

with $\mu(x_i), \sigma(x_i)$ (respectively $\mu(y_i), \sigma(y_i)$) the mean and standard deviation of the token $x_i \in \mathbb{R}^d$ (respectively of $y_i$). From the definition of LayerNorm (see Section 2), it leads to

$$
\begin{aligned}
f(x) - f(y) &= \left( \gamma \odot \frac{x_1 - \mu(x_1)}{\sigma(x_1)} + \beta, \ldots, \gamma \odot \frac{x_n - \mu(x_n)}{\sigma(x_n)} + \beta \right) \\
&\quad - \left( \gamma \odot \frac{y_1 - \mu(y_1)}{\sigma(y_1)} + \beta, \ldots, \gamma \odot \frac{y_n - \mu(y_n)}{\sigma(y_n)} + \beta \right) \\
&= \left( \gamma \odot \frac{x_1 - \mu_1}{\sigma_1} + \beta, \ldots, \gamma \odot \frac{x_n - \mu_n}{\sigma_n} + \beta \right) \\
&\quad - \left( \gamma \odot \frac{y_1 - \mu_1}{\sigma_1} + \beta, \ldots, \gamma \odot \frac{y_n - \mu_n}{\sigma_n} + \beta \right) \\
&= \left( \gamma \odot \frac{x_1 - y_1}{\sigma_1}, \ldots, \gamma \odot \frac{x_n - y_n}{\sigma_n} \right) .
\end{aligned}
$$

Denoting $\tilde{x}$, respectively $\tilde{y}$, the sequence with entries $\left(\frac{x_i}{\sigma_i}\right)_{i=1}^n$, respectively $\left(\frac{y_i}{\sigma_i}\right)_{i=1}^n$, we have

$$\begin{aligned}
\|f(x) - f(y)\|_{\mathrm{F}} &= \|\gamma \odot (\tilde{x} - \tilde{y})\|_{\mathrm{F}} \\
&= \|\Gamma(\tilde{x} - \tilde{y})\|_{\mathrm{F}} \\
&\leq \|\Gamma\|_2 \|\tilde{x} - \tilde{y}\|_{\mathrm{F}},
\end{aligned}$$

where the second line comes from defining $\Gamma \in \mathbb{R}^{d \times d}$ as a diagonal matrix with values the entries of $\gamma \in \mathbb{R}^d$ and replacing the element-wise product by a matrix product, and the last line comes from using Lemma 1. We recall that

$$\|\tilde{x} - \tilde{y}\|_{\mathbf{F}}^2 = \sum_{i=1}^n \|(x_i - y_i)/\sigma_i\|_2 \leq \left(\frac{1}{\min_{i=1}^n \sigma_i}\right)^2 \sum_{i=1}^n \|(x_i - y_i)\|_2 = \frac{1}{\sigma^2}\|x - y\|_{\mathbf{F}}^2,$$

where $\sigma = \min_{i=1}^n \sigma_i > 0$, and that $\|\Gamma\|_2 = \max_{i=1}^d |\gamma_i| = \|\gamma\|_\infty$ by the definition of the spectral norm of a diagonal matrix. We then obtain

$$\|f(x) - f(y)\|_{\mathrm{F}} \leq \frac{1}{\sigma}\|\gamma\|_\infty \|x - y\|_{\mathrm{F}}.$$

Since the result holds for randomly sampled sequences $x, y$, and by assumption, the $\sigma_i$ depend on the embedding layer and not on the input sequences, we can upper bound the rate of change and take the expectation over distinct sequences of tokens $x, y$. We have

$$\mathcal{P}(f) = \mathbb{E}_{(x,y) \sim \mu}\left[\frac{\|f(x) - f(y)\|_{\mathrm{F}}}{\|x - y\|_{\mathrm{F}}}\right] \leq \frac{1}{\sigma}\|\gamma\|_\infty,$$

which concludes the proof for the LayerNorms. $\qquad\square$

### B.2. Proof of Proposition 2

*Proof.* The proof derivation is simple for the case of linear layers. We detail it below for consistency. Let $f$ be a linear layer with weights $W_1 \in \mathbb{R}^{d \times 4d}$ and let $\nu$ be the uniform distribution over the set of distinct pairs of sequences of tokens in $(\mathbb{R}^d)^n$. Let $(x, y)$ be a pair of two distinct sequences of tokens sampled according to $\nu$. Let $x, y \in \mathbb{R}^{n \times d}$ be two distinct sequences of tokens sampled according to $\mu$.. By definition of the linear layer, a simple application of Lemma 1 leads to

$$\|f(x) - f(y)\|_{\mathbf{F}} = \|W_1(x - y)\|_{\mathbf{F}} \leq \|W_1\|_2\|x - y\|_{\mathbf{F}}.$$

We obtain a similar result for the second linear layer with weights in $\mathbb{R}^{4d \times d}$. As in Appendix B.1, since the upper bound holds for randomly sampled sequences $x, y$, we can bound the rate of change and take the expectation to conclude the proof. $\qquad\square$

### B.3. Proof of Proposition 3

*Proof.* Let $f$ be a multihead self-attention module with weights $(O^h, Q^h, K^h, V^h)_{1 \leq h \leq H}$, with $A^h = (Q^h)^\top K^h / \sqrt{k}$, and let $\nu$ be the uniform distribution over the set of distinct pairs of sequences of tokens in $(\mathbb{R}^d)^n$. Let $(x, y)$ be a pair of two distinct sequences of tokens sampled according to $\nu$. Let $x, y \in \mathbb{R}^{n \times d}$ be two distinct sequences of tokens sampled according to $\mu$. By the definition of multihead self-attention (see Section 2), we have

$$\begin{aligned}
\|f(x) - f(y)\|_{\mathrm{F}} &= \|\sum_{h=1}^H O^h\big(f_{\mathrm{att}}^h(x) - f_{\mathrm{att}}^h(x)\big)\|_{\mathrm{F}} \\
&\leq \sum_{h=1}^H \|O^h\big(f_{\mathrm{att}}^h(x) - f_{\mathrm{att}}^h(x)\big)\|_{\mathrm{F}} \qquad\qquad \text{(triangular inequality)} \\
&\leq \sum_{h=1}^H \|O^h\|_2 \|f_{\mathrm{att}}^h(x) - f_{\mathrm{att}}^h(x)\|_{\mathrm{F}}. \qquad\qquad \text{(Lemma 1)}
\end{aligned}$$

We recall that following Castin et al. (2024), the Lipschitz constant of $f$ on $\mathcal{X} \subset (\mathbb{R}^d)^n$ writes

$$\mathrm{Lip}(f_{|\mathcal{X}}) = \sup_{\substack{x,y \in \mathcal{X} \\ x \neq y}} \frac{\|f(x) - f(y)\|_{\mathrm{F}}}{\|x - y\|_{\mathrm{F}}}. \qquad\qquad (2)$$

We recall the following result on the Lipschitz constant of self-attention.

**Theorem B.1.** *(Castin et al., 2024, Theorem 3.3) Let $Q, K, V \in \mathbb{R}^{k \times d}$ and $A = Q^\top K / \sqrt{k}$. Let $r > 0$ and $n \in \mathbb{N}$. A self-attention module $f_{\mathrm{att}}$ with weights $Q, K, V$ is Lipschitz continuous on $B_r^n$, with*

$$\mathrm{Lip}(f_{\mathrm{att}}|B_r^n) \leq \sqrt{3}\|V\|_2 \sqrt{\|A\|_2^2 r^4 (4n+1) + n}.$$

By assumption, the sequences of tokens are restricted to $B_r^n$. We can thus apply Theorem B.1 on each self-attention module $f_{\mathrm{att}}^h$ with weights $Q^h, K^h, V^h$. Using the fact that the Lipschitz constant in Eq. (2) is a supremum over individual rates of change, we have

$$\|f(x) - f(y)\|_{\mathrm{F}} \leq \sum_{h=1}^{H} \|O^h\|_2 \cdot \mathrm{Lip}\big(f_{\mathrm{att}}^h|B_r^n\big)\|x - y\|_{\mathrm{F}}$$

$$\leq \left(\sum_{h=1}^{H} \|O^h\|_2 \sqrt{3}\|V^h\|_2 \sqrt{\|A^h\|_2^2 r^4 (4n+1) + n}\right)\|x - y\|_{\mathrm{F}}.$$

As in Appendix B.1, since the upper bound holds for randomly sampled sequences $x, y$, we can bound the rate of change and take the expectation to conclude the proof. $\square$

## B.4. Proof of Proposition 4

*Proof.* Let $f$ be a multihead self-attention module with weights $(O^h, Q^h, K^h, V^h)_{1 \leq h \leq H}$, with $A^h = (Q^h)^\top K^h / \sqrt{k}$, and let $\nu$ be the uniform distribution over the set of distinct pairs of sequences of tokens in $(\mathbb{R}^d)^n$. Let $(x, y)$ be a pair of two distinct sequences of tokens sampled according to $\nu$. We first show that the assumption on the energy of images leads to sequences of tokens with bounded Frobenius norm. The digital image embedded to obtain the sequence of tokens $x$ can be seen as a discretization of its continuous intensity, denoted by $I \colon \Omega \subset \mathbb{R}^2 \to \mathbb{R}_+$ (for convenience, we consider a grayscale image, but similar derivations are straightforward for an RGB image). The total energy of the image is defined as the sum of the squared intensities over pixels (see Mallat, 2008, Chapter 1, page 2). In line with the signal processing literature (Goodman, 2005; Mallat, 2008), the image has a finite energy $\mathcal{E}_x \geq 0$, which is bounded by $\mathcal{E}$ by assumption. Summing over pixels, we have

$$\mathcal{E}_x = \sum_{u,v \in \Omega} |I(u,v)|^2 \leq \mathcal{E} < \infty.$$

In vision transformers, images are split into $n$ square patches of size $P$. The $i$-th patch, denoted by $p_i \in \mathbb{R}^{P \times P}$, covers an area $\Omega_i$ and has an energy $\mathcal{E}_i \geq 0$. We have:

$$\mathcal{E} = \sum_{u,v \in \Omega} |I(u,v)|^2 = \sum_{i=1}^{n} \sum_{u,v \in \Omega_i} |I(u,v)|^2 = \sum_{i=1}^{n} \mathcal{E}_i.$$

Denoting by $x = (x_1, \ldots, x_n) \in (\mathbb{R}^d)^n$ the sequence of tokens obtained after embedding the image, where the $i$-th token $x_i$ is obtained by flattening the $i$-th patch $p_i$ and linearly projecting it in $\mathbb{R}^d$. Since the input images have dimensions $H \times W \times C$ (see Dosovitskiy et al., 2021, Section 3.1), the flattened patches have a dimension of $m = P^2 \times C$. We denote by $E \in \mathbb{R}^{d \times m}$ the weights of the embedding layer. Using the property of the spectral norm $\|\cdot\|_2$ (which is the operator norm induced by the Euclidean norm $\|\cdot\|$), we have

$$\|x_i\| = \|E\mathrm{vec}(p_i)\| \leq \|E\|_2 \|\mathrm{vec}(p_i)\|,$$

where $\mathrm{vec}(\cdot)$ denotes the operator that transforms a matrix into a column vector. By definition, we have

$$\|\mathrm{vec}(p_i)\|^2 = \|p_i\|_{\mathrm{F}}^2 = \sum_{u,v \in \Omega_i} |I(u,v)|^2 du dv = \mathcal{E}_i.$$

As such, the Frobenius norm of the sequence of tokens $x$ verifies

$$\|x\|_{\mathrm{F}} = \sqrt{\sum_{i=1}^{n} \|x_i\|^2} \leq \sqrt{\sum_{i=1}^{n} \|E\|_2^2 \mathrm{vec}(p_i)\|^2} = \|E\|_2 \sqrt{\sum_{i=1}^{n} \|\mathrm{vec}(p_i)\|^2} = \|E\|_2 \sqrt{\sum_{i=1}^{n} \mathcal{E}_i} = \|E\|_2 \cdot \sqrt{\mathcal{E}}, \quad (3)$$

as intended. In the rest of the proof, we denote $R = \alpha \cdot \sqrt{\mathcal{E}}$, with $\alpha$ the spectral norm of $E$. This implies that the sequences of tokens are in $B_R^n$, which corresponds to the setting of Proposition 3. Indeed, each token verifies $\|x_i\| \leq R$; otherwise, the Frobenius norm would be greater than $R$. We now proceed to bound $\|f(x) - f(y)\|_{\mathrm{F}}$. For any $h \in [H]$, we introduce for convenience the function $S^h \colon \mathbb{R}^{d \times n} \to \mathbb{R}^{n \times n}$ as

$$S^h(x) = \mathrm{softmax}\left(\frac{(Qx)^\top Kx}{\sqrt{k}}\right) = \mathrm{softmax}(x^\top A^h x).$$

Similarly to the proof of Proposition 3, we have used the triangular inequality and Lemma 1 that

$$
\begin{aligned}
\|f(x) - f(y)\|_{\mathrm{F}} &= \|\sum_{h=1}^{H} O^h\big(f_{\mathrm{att}}^h(x) - f_{\mathrm{att}}^h(x)\big)\|_{\mathrm{F}} \\
&\leq \sum_{h=1}^{H} \|O^h\big(f_{\mathrm{att}}^h(x) - f_{\mathrm{att}}^h(x)\big)\|_{\mathrm{F}} \\
&\leq \sum_{h=1}^{H} \|O^h\|_2 \|f_{\mathrm{att}}^h(x) - f_{\mathrm{att}}^h(x)\|_{\mathrm{F}}.
\end{aligned}
\tag{4}
$$

Moreover, by the definition of the self-attention layer, we have

$$\|f_{\mathrm{att}}^h(x) - f_{\mathrm{att}}^h(y)\|_{\mathrm{F}} = \|(V^h x)S^h(x) - (V^h y)S^h(y)\|_{\mathrm{F}} \leq \|V^h\|_2 \|xS^h(x) - yS^h(y)\|_{\mathrm{F}}, \tag{5}$$

where we used Lemma 1 for the inequality. Moreover, we have

$$
\begin{aligned}
\|xS^h(x) - yS^h(y)\|_{\mathrm{F}} &= \|x\big(S^h(x) - S^h(y)\big) + (x - y)S^h(y)\|_{\mathrm{F}} \\
&\leq \|x\big(S^h(x) - S^h(y)\big)\|_{\mathrm{F}} + \|(x - y)S^h(y)\|_{\mathrm{F}} \qquad \text{(triangular inequality)} \\
&\leq \|x\|_{\mathrm{F}} \|S^h(x) - S^h(y)\|_{\mathrm{F}} + \|x - y\|_{\mathrm{F}} \|S^h(y)\|_{\mathrm{F}},
\end{aligned}
$$

where we used Lemma 1 for the last inequality. We recall that we have $\|x\|_{\mathrm{F}} \leq R$ from Eq. (3). Recalling the fact that $S$ is row-stochastic leads to $\|S^h(y)\|_{\mathrm{F}} \leq \sqrt{n}$ via the simple derivation

$$\|S^h(y)\|_{\mathrm{F}}^2 = \|S\|_{\mathrm{F}}^2 = \sum_{i=1}^{n}\sum_{j=1}^{n} S_{ij}^2 \leq \sum_{i=1}^{n}\underbrace{\sum_{j=1}^{n} S_{ij}}_{=1} \leq n,$$

where we used $S = S^h(y)$ to alleviate the notations, this leads to

$$\|xS^h(x) - yS^h(y)\|_{\mathrm{F}} \leq R\|S^h(x) - S^h(y)\|_{\mathrm{F}} + \sqrt{n}\|x - y\|_{\mathrm{F}}. \tag{6}$$

We now proceed to bound the term $\|S^h(x) - S^h(y)\|_{\mathrm{F}}$. Since the softmax operator is applied row-wise, the $i$-th row of $S^h(x)$ writes $g(x)_i = \mathrm{softmax}\big((x^\top A^h x)_i\big) \in \mathbb{R}^{1 \times n}$. We define $g(y)_i$ similarly. Then, we have

$$\|S^h(x) - S^h(y)\|_{\mathrm{F}}^2 = \sum_{i=1}^{n}\sum_{j=1}^{n}(S^h(x) - S^h(y))_{ij}^2 = \sum_{i=1}^{n} \|g(x)_i - g(y)_i\|^2.$$

We recall the following result on the Lipschitz constant of the softmax operator with respect to the Euclidean norm, i.e., the $\ell_2$ norm (we note that Nair (2026) states the result for any $\ell_p$ norm).

**Theorem B.2.** *(Nair, 2026, Theorem 1) Let $n \in \mathbb{N}$ and $u, v \in \mathbb{R}^n$. Then, $\|\mathrm{softmax}(u) - \mathrm{softmax}(v)\| \leq \frac{1}{2}\|u - v\|$.*

Applying Theorem B.2 leads for any $i \in [n]$ to

$$\|g(x)_i - g(y)_i\| = \|\mathrm{softmax}\big((x^\top A^h x)_i\big) - \mathrm{softmax}\big((y^\top A^h y)_i\big)\| \leq \frac{1}{2}\|(x^\top A^h x)_i - (y^\top A^h y)_i\|.$$

Hence, we obtain using the fact that $\sqrt{\cdot}$ is monotonically increasing that

$$
\begin{aligned}
\|S^h(x) - S^h(y)\|_{\mathrm{F}} &\leq \left( \frac{1}{4} \sum_{i=1}^{n} \|(x^\top A^h x)_i - (y^\top A^h y)_i\|^2 \right)^{1/2} \\
&= \frac{1}{2} \left( \sum_{i=1}^{n} \|(x^\top A^h x - y^\top A^h y)_i\|^2 \right)^{1/2} \\
&= \frac{1}{2} \|x^\top A^h x - y^\top A^h y\|_{\mathrm{F}} \\
&= \frac{1}{2} \|(x - y)^\top A^h x - y^\top A^h (x - y)\|_{\mathrm{F}} \\
&\leq \frac{1}{2} \left( \|(x - y)^\top A^h x\|_{\mathrm{F}} + \|y^\top A^h (x - y)\|_{\mathrm{F}} \right) \\
&\leq \frac{1}{2} \left( \|(x - y)^\top \|_{\mathrm{F}} \|A^h x\|_{\mathrm{F}} + \|y^\top A^h\|_{\mathrm{F}} \|x - y\|_{\mathrm{F}} \right) && \text{(Lemma 1)} \\
&\leq \frac{1}{2} \left( \|(x - y)^\top \|_{\mathrm{F}} \|A^h\|_2 \|x\|_{\mathrm{F}} + \|y^\top \|_{\mathrm{F}} \|A^h\|_2 \|x - y\|_{\mathrm{F}} \right). && \text{(Lemma 1)}
\end{aligned}
$$

Since singular values are invariant to transposition and the Frobenius norm can be expressed as the sum of the singular values, we know that $\|(x - y)^\top \|_{\mathrm{F}} = \|x - y\|_{\mathrm{F}}$ and that $\|y^\top \|_{\mathrm{F}} = \|y\|_{\mathrm{F}}$. Recalling that by assumption, the Frobenius norm of sequences of tokens is bounded by $R$ from Eq. (3), we have

$$
\begin{aligned}
\|S^h(x) - S^h(y)\|_{\mathrm{F}} &\leq \frac{1}{2} \left( \|x - y\|_{\mathrm{F}} \|A^h\|_2 \|x\|_{\mathrm{F}} + \|y\|_{\mathrm{F}} \|A^h\|_{\mathrm{F}} \|x - y\|_{\mathrm{F}} \right) \\
&\leq R \|A^h\|_2 \|x - y\|_{\mathrm{F}}.
\end{aligned}
$$

From Eq. (6), we obtain
$$
\|xS^h(x) - yS^h(y)\|_{\mathrm{F}} \leq \left( R^2 \|A^h\|_2 + \sqrt{n} \right) \|x - y\|_{\mathrm{F}},
$$

and from Eq. (5) we obtain

$$
\|f_{\mathrm{att}}^h(x) - f_{\mathrm{att}}^h(y)\|_{\mathrm{F}} \leq \|V^h\|_2 \left( R^2 \|A^h\|_2 + \sqrt{n} \right) \|x - y\|_{\mathrm{F}},
$$

This leads using Eq. (4) to

$$
\|f(x) - f(y)\|_{\mathrm{F}} \leq \sum_{h=1}^{H} \|O^h\|_2 \|V^h\|_2 \left( R^2 \|A^h\|_2 + \sqrt{n} \right) \|x - y\|_{\mathrm{F}},
$$

with and $R = \alpha \sqrt{\mathcal{E}}$. As in Appendix B.1, since the upper bound holds for randomly sampled sequences $x, y$, we can bound the rate of change and take the expectation to conclude the proof. $\qquad \square$

# C. Reproducibility

In this section, we provide details to reproduce our work.

## C.1. Open-source code

To facilitate knowledge transfer and minimize redundant training runs within the community, our code and findings have been made publicly available at `github.com/ambroiseodt/vit-plasticity`. Our library is built such that researchers can adapt all or part of the code for their specific use cases. We notably hope it can be used to further extend the study to large language models.

## C.2. Carbon footprint

This project required comprehensive, large-scale experiments with around 1000 finetuning runs for an equivalent of 3700 GPU hours. With public cloud providers such as Azure or Amazon Web Services, this can cost up to $40,000$. With a carbon efficiency of $0.1$ kgCO$_2$eq/kWh in the France region, the total emissions are estimated to be roughly 259 kgCO$_2$eq. This is equivalent to a round-trip flight from Paris to Madrid with a Boeing 737. We note that this number is low because France's electricity grid relies heavily on nuclear and renewable energy. For similar GPU-hours in regions using coal and gas, such as Germany, carbon emissions would be much higher. Estimations were conducted using the ML Impact calculator presented in Lacoste et al. (2019).

# D. Implementation details

In this section, we provide full implementation details.

## D.1. Vision transformers

This section is focused on the vision transformer implementation.

**Architecture.** In vision transformers (ViT, Dosovitskiy et al., 2021), inputs are 2D images that are split into square patches of size $P$, which are flattened and linearly embedded in dimension $d$. A classification token CLS is prepended to the sequence of tokens before adding positional embeddings. The obtained sequence of tokens $x = (x_1, \ldots, x_n) \in (\mathbb{R}^d)^n$ is fed to a succession of transformer encoders (Vaswani et al., 2017). Each block consists of a multihead self-attention module followed by a feedforward network implemented as a two-layer MLP with GeLU activation (Hendrycks & Gimpel, 2016) and a hidden dimension taken as 4 times the embedding dimension (Dosovitskiy et al., 2021; Vaswani et al., 2017). A LayerNorm (Ba et al., 2016) is applied before each block, and a residual connection is applied after each block. It leads to the 5 modules displayed in Fig. 2 (left).

**Implementation.** In our experiments, we use ViT models of size 86M, 307M, and 632M with patch sizes 16, 16, and 14, respectively. Models are pretrained on ImageNet22k. Their characteristics are given in Table 4. In our code, we follow the original ViT implementation from Dosovitskiy et al. (2021) and use a convolutional layer to embed images (see Dosovitskiy et al., 2021, §"Hybrid Architecture"). This is also the standard in the implementation from HuggingFace (2025). In Fig. 6, we display the implementation of the ViT-Base model with a classification head for 10 classes obtained using our modular library `vitef`.

*Table 4.* Details of ViT variants (Dosovitskiy et al., 2021) with the patch size, the number of layers, the number of attention heads, the embedding dimension, the number of parameters, and the link to the pretrained weights.

| model | patch size $P$ | seq. length $n$ | layers | heads $H$ | embedding $d$ | FFN hidden dimension | parameters |
|---|---|---|---|---|---|---|---|
| ViT-Base | 16 | 197 | 12 | 12 | 768 | 3072 | 86M |
| ViT-Large | 16 | 197 | 24 | 16 | 1024 | 4096 | 307M |
| ViT-Huge | 14 | 257 | 32 | 16 | 1280 | 5120 | 632M |

```python
# Python snippet to print the ViT architecture
from vitef.models import build_model

model = build_model(implementation="vit", model_name="base", n_classes=10)
print(model)

# Corresponding output
Transformer(
  (embedding): Embedding(
    (patching): PatchImages(
      (patching): Sequential(
        (0): Conv2d(3, 768, kernel_size=(16, 16), stride=(16, 16))
        (1): Flatten(start_dim=2, end_dim=-1)
      )
    )
  )
  (blocks): ModuleList(
    (0-11): 12 x TransformerBlock(
      (attn_norm): LayerNorm((768,), eps=1e-12, elementwise_affine=True)
      (attn): SelfAttention(
        (qkv_mat): Linear(in_features=768, out_features=2304, bias=True)
        (output): Linear(in_features=768, out_features=768, bias=True)
      )
      (ffn_norm): LayerNorm((768,), eps=1e-12, elementwise_affine=True)
      (ffn): FeedForward(
        (fc1): Linear(in_features=768, out_features=3072, bias=True)
        (fc2): Linear(in_features=3072, out_features=768, bias=True)
      )
    )
  )
  (output): Output(
    (output_layer): ClassificationLayer(
      (output_norm): LayerNorm((768,), eps=1e-12, elementwise_affine=True)
      (output): Linear(in_features=768, out_features=10, bias=True)
    )
  )
)
```

*Figure 6.* ViT-Base Implementation.

### D.2. Data preprocessing

All our experiments are conducted on a varied collection of 11 classification benchmarks: Cifar10, Cifar100 (Krizhevsky, 2009); variants from Cifar10-C (Hendrycks & Dietterich, 2019) with severity 5: Contrast, Gaussian Noise, Motion Blur, Snow, Speckle Noise; 2 domains from DomainNet (Peng et al., 2019), a challenging benchmark typically used for domain generalization: Clipart, Sketch; Flowers102 (Nilsback & Zisserman, 2008) and Pets (Parkhi et al., 2012).

The preprocessing follows Dosovitskiy et al. (2021) and Kolesnikov et al. (2020): for training data, we apply random cropping, a 224×224 image resizing, and random horizontal flip for training images. For validation and test data, the 224×224 image resizing is applied before center cropping images. All images are normalized using the ImageNet1k (Deng et al., 2009) statistics. It ensures images with mean $[0.485, 0.456, 0.406]$ and standard deviation $[0.229, 0.224, 0.225]$. For datasets that do not have predefined training and test sets (i.e., datasets from Cifar10-C and DomainNet), we manually create *deterministic* training and test sets following a $80\% - 20\%$ split. The deterministic part is crucial to ensure no data contamination.

### D.3. Plasticity setup

This section is focused on the plasticity analysis.

**Realistic setting.**   In real-world applications, the discrepancy between the pretraining and downstream data is not known a priori. This motivates us to compute the plasticity images coming from the pretraining distribution and various downstream distributions, without any additional assumption. This differs from prior work, where the distribution shift can be categorized, e.g., into natural, subpopulation, or synthetic shift (Deng et al., 2023; Lee et al., 2023; Xie et al., 2024, 2025).

**Practical implementation.**   The sequences of tokens $x, y$ are obtained by embedding preprocessed images with the pretrained model studied. We loop over $N$ batches of size $b$ with forward passes on the GPU and store high-dimensional outputs on the CPU. This ensures a fast computation and avoids out-of-memory issues. The total number of samples used to compute the plasticity is equal to $N \times b$. We note that all the transformer components take as input sequences of tokens in $\mathbb{R}^d$, except for the second layer of the feedforward $f_{\text{fc2}}$, where the tokens must be in $\mathbb{R}^{4d}$. Akin to how a vector in the plane can be mapped to a 3D vector $(u_1, u_2, 0)$, we lift each token $x_i$ into $\mathbb{R}^{4d}$ by padding the remaining entries with zeros.

## D.4. Finetuning setup

The finetuning experiments of Section 5.2 follow the protocol from Dosovitskiy et al. (2021) with a resolution of $224 \times 224$.

**Configurations.**   We consider pretrained models like ViT-Base and ViT-Large and finetune each of their trainable components in isolation: we freeze all the weights of each model, except the group studied, which is optimized across the depth: the attention norm (LN1), the attention module (MHA), the feedforward norm (LN2), the first feedforward layer (FC1), and the second feedforward layer (FC2). The classification head is randomly initialized following Dosovitskiy et al. (2021). This leads to the 5 configurations described in Table 5 along with their corresponding number of trainable parameters. We add as a baseline the full-finetuning (All), where all the model's parameters are trainable.

*Table 5.* **Finetuning configurations**. Configurations are denoted by the name of the trainable transformer component and ordered in terms of plasticity ranking (see Section 5.1). We report the number of trainable parameters on ViT-Base.

| configuration | MHA | FC1 | FC2 | LN2 | LN1 |
|---|---|---|---|---|---|
| parameters | 28M | 28M | 28M | 18K | 18K |
| % of total | 33 | 33 | 33 | 0.02 | 0.02 |

**Memory load.**   The finetuning configurations have the same inference cost since they share the same ViT architecture. However, the number of trainable parameters differs. The GPU usage of training a model consists of the memory load to store the model parameters, the optimizer states, the gradients, and the activations (Thor, 2025). In our setting, the memory load is the same between configurations except for the optimizer and the gradient computation. For a model with $P$ parameters and a precision of $b$ bytes, the memory required to store the gradients is $Pb$ because backpropagation computes a gradient per parameter. The same memory is needed for the optimizer states with SGD (and the double for Adam (Kingma & Ba, 2014; Loshchilov & Hutter, 2019), which also computes the variance). In Table 6, the memory usage for one training step on Cifar10 for each configuration with a default FP32 precision.

*Table 6.* **Memory load comparison**. Memory usage of the optimizer and gradients for one training step (in MB).

| configuration | MHA | FC1 | FC2 | LN2 | LN1 |
|---|---|---|---|---|---|
| *memory load* | 220 | 220 | 220 | 0.14 | 0.14 |

**Optimization.**   We optimize models with the Stochastic Gradient Descent (SGD), a momentum of 0.9, no weight decay, a cosine learning rate decay, a warmup of 2000 steps, a batch size of 512, and gradient clipping at norm 1. The finetuning resolution is of 224. For each pair of dataset - configuration, we perform a sweep over 4 learning rates, as summarized in Table 7, and conduct 3 runs with different seeds relative to network initialization and dataloaders.

**Performance.**   For each run, we monitor the training using a validation set (20% of the training set). The final performance is the test accuracy of the checkpoint that achieves the best validation accuracy.

*Table 7.* **Finetuning hyperparameters**. We report the choice of optimizer, batch size, training steps, and learning rates.

| dataset | optimizer | batch size | training steps | learning rates $\eta$ |
|---|---|---|---|---|
| Cifar10 | SGD | 512 | 10000 | {1e−3, 3e−3, 1e−2, 3e−2} |
| Cifar100 | SGD | 512 | 10000 | {1e−3, 3e−3, 1e−2, 3e−2} |
| Contrast | SGD | 512 | 10000 | {1e−3, 3e−3, 1e−2, 3e−2} |
| Gaussian Noise | SGD | 512 | 10000 | {1e−3, 3e−3, 1e−2, 3e−2} |
| Motion Blur | SGD | 512 | 10000 | {1e−3, 3e−3, 1e−2, 3e−2} |
| Snow | SGD | 512 | 10000 | {1e−3, 3e−3, 1e−2, 3e−2} |
| Speckle Noise | SGD | 512 | 10000 | {1e−3, 3e−3, 1e−2, 3e−2} |
| Clipart | SGD | 512 | 20000 | {3e−3, 1e−2, 3e−2, 6e−2} |
| Sketch | SGD | 512 | 20000 | {3e−3, 1e−2, 3e−2, 6e−2} |
| Flowers102 | SGD | 512 | 5000 | {1e−3, 3e−3, 1e−2, 3e−2} |
| Pets | SGD | 512 | 4000 | {1e−3, 3e−3, 1e−2, 3e−2} |

# E. Additional experiments

In this section, we report the detailed results corresponding to the figures presented in the paper, along with additional experiments not shown in the main due to space constraints. For reproducibility, we provide the carbon footprint of our project.

## E.1. Plasticity analysis

In this section, we present the additional figures and experiments related to Sections 4 and 5.1.

### E.1.1. THEORETICAL PLASTICITY RANKING

We numerically compute the plasticity upper bounds of Section 4 on ViT-Base. The sequence length is $n = 197$, and the number of attention heads is $H = 12$. Following Castin et al. (2024, Section 5), the average radius is computed over input sequences $x = (x_1, \ldots, x_n)$ as $r = \sqrt{\frac{1}{n} \sum_{i=1}^{n} \|x_i\|^2}$. The value $r = 19.4$ obtained on Cifar10 (Krizhevsky, 2009) is used as the reference for the computation of the bounds in Propositions 1 to 3. We display the upper bounds in Fig. 7. The upper bounds ranking follows our theoretical insight,s with the attention module having the largest upper bound, followed by the first and second feedforward layer, the LayerNorm preceding the feedforward network, and finally, the LayerNorm preceding the attention module. We note that the upper bound of the attention module is several orders of magnitude larger than the other components. Even with the dependency in $n^{1/4}$ empirically observed in Castin et al. (2024, Fig. 1), the order of the bound remains $10^6$. We attribute this scale to the dependency of the bound on the number of heads, the radius $r$, and the sequence length $n$. As explained in Section 4, the bound is tight in terms of dependency in $n$, the numerical values of $r$ and $n$ being close leads to a large bound in practice. We notice in Section 5.1 that the plasticity scales are more similar between modules than the upper bounds. This further confirms that the difference in scale between the upper bounds is due to the difficulty of bounding the self-attention Lipschitz constant. In particular, we observe in Section 5.1 that the plasticity computed as an average rate of change follows the same ranking but with lower magnitude, notably for the attention module. This is reminiscent of Ashlagi et al. (2021) where the authors showed that the gap between the Lipschitz constant and the average rate of change can be considerable.

### E.1.2. PLASTICITY EXPERIMENTS OF ALL BENCHMARKS

We extend the analysis of Section 5.1 to additional datasets and display the results in Figs. 8 to 17. Our findings are aligned with the theoretical analysis in Section 4 and shows that the attention module has the highest plasticity, followed by the first feedforward linear layer, then the second feedforward linear layer. The LayerNorms are more rigid with a plasticity below 1.

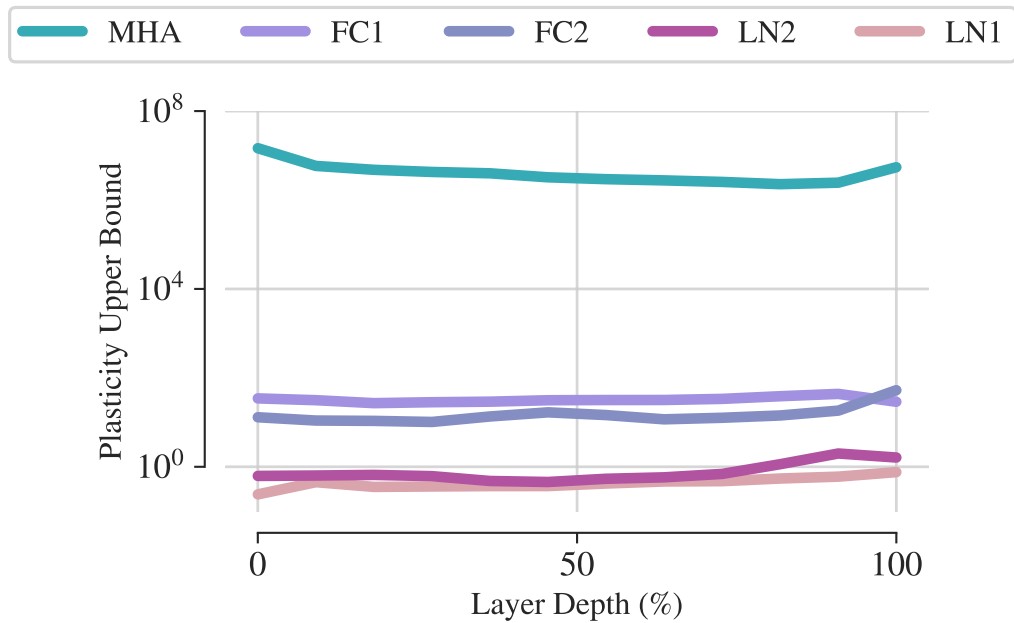

*Figure 7.* **Plasticity upper bounds on ViT-Base.** The sequence length is $n = 197$, the number of heads is $H = 12$ and the average radius is computed over input sequences $x = (x_1, \ldots, x_n)$ as $r = \sqrt{\frac{1}{n} \sum_{i=1}^{n} \|x_i\|^2}$. We obtain a value of $r = 19.4$. We can see that the attention module has the highest plasticity, followed by the first and second feedforward layers, the LayerNorm preceding the feedforward, and finally the LayerNorm preceding the attention module. This aligns with our theoretical insights.

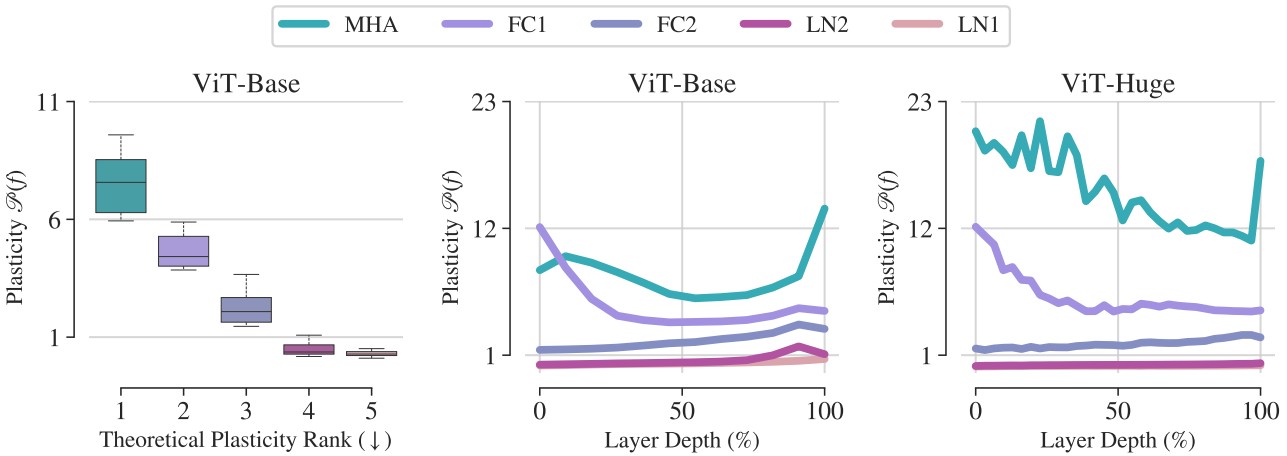

*Figure 8.* **Plasticity analysis on Cifar10.** The distribution of rates of change $\|f(x) - f(y)\|_{\mathrm{F}} / \|x - y\|_{\mathrm{F}}$ on ViT-Base (**left**) follow the upper bound ranking predicted by our theory in Section 4. We observe along transformer blocks of ViT-Base (**middle**) that the attention module has the highest plasticity $\mathcal{P}(f)$, followed by the first and second linear layers of the feedforward. The LayerNorms are the most rigid, with a plasticity below 1. The same pattern is obtained on ViT-Huge (**right**), where the higher attention plasticity further validates our theory (see Proposition 3) since the sequence length $n$ is larger than with ViT-Base.

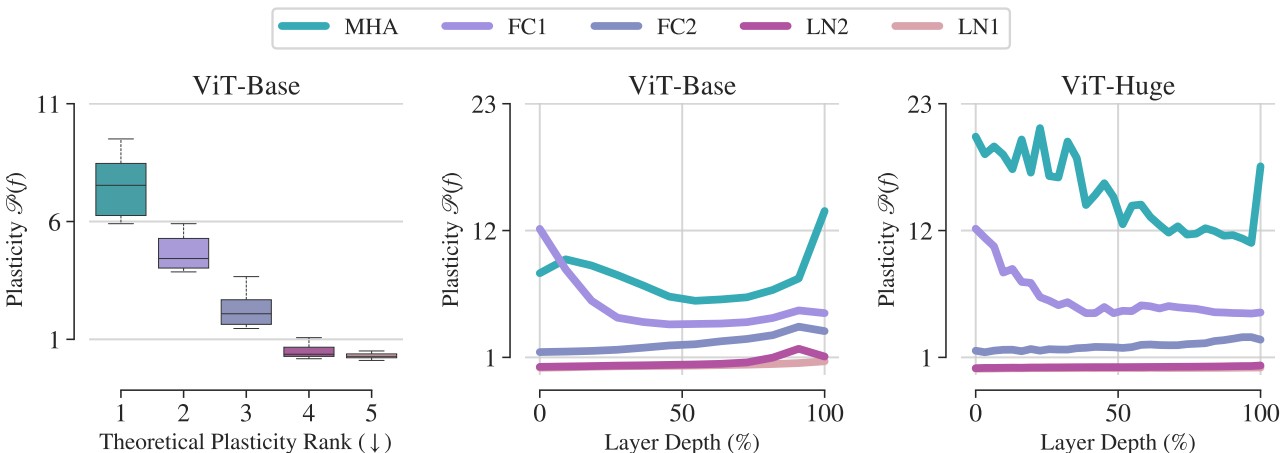

*Figure 9.* **Plasticity analysis on Cifar100.** The distribution of rates of change $\|f(x) - f(y)\|_F / \|x - y\|_F$ on ViT-Base (**left**) follow the upper bound ranking predicted by our theory in Section 4. We observe along transformer blocks of ViT-Base (**middle**) that the attention module has the highest plasticity $\mathcal{P}(f)$, followed by the first and second linear layers of the feedforward. The LayerNorms are the most rigid, with a plasticity below 1. The same pattern is obtained on ViT-Huge (**right**), where the higher attention plasticity further validates our theory (see Proposition 3) since the sequence length $n$ is larger than with ViT-Base.

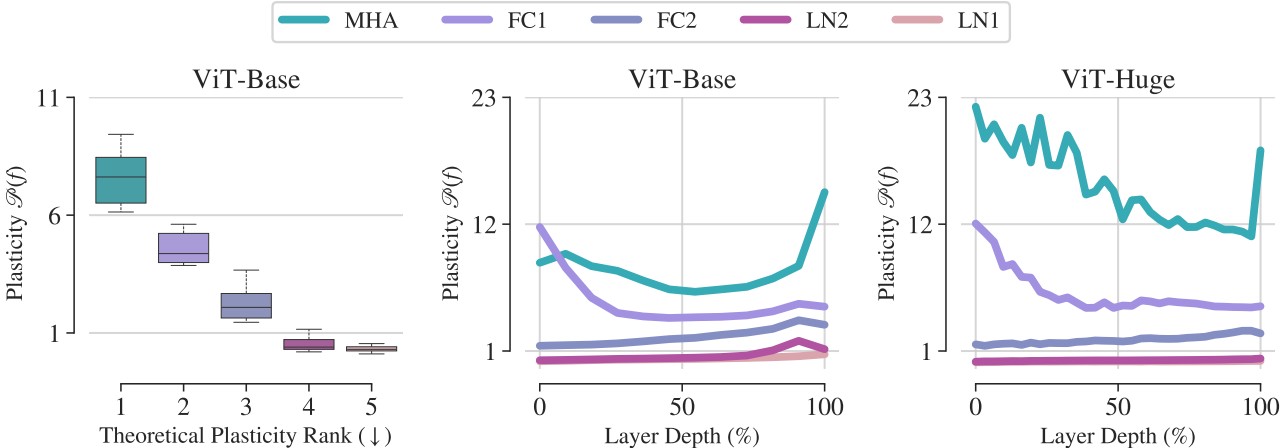

*Figure 10.* **Plasticity analysis on Contrast.** The distribution of rates of change $\|f(x) - f(y)\|_F / \|x - y\|_F$ on ViT-Base (**left**) follow the upper bound ranking predicted by our theory in Section 4. We observe along transformer blocks of ViT-Base (**middle**) that the attention module has the highest plasticity $\mathcal{P}(f)$, followed by the first and second linear layers of the feedforward. The LayerNorms are the most rigid, with a plasticity below 1. The same pattern is obtained on ViT-Huge (**right**), where the higher attention plasticity further validates our theory (see Proposition 3) since the sequence length $n$ is larger than with ViT-Base.

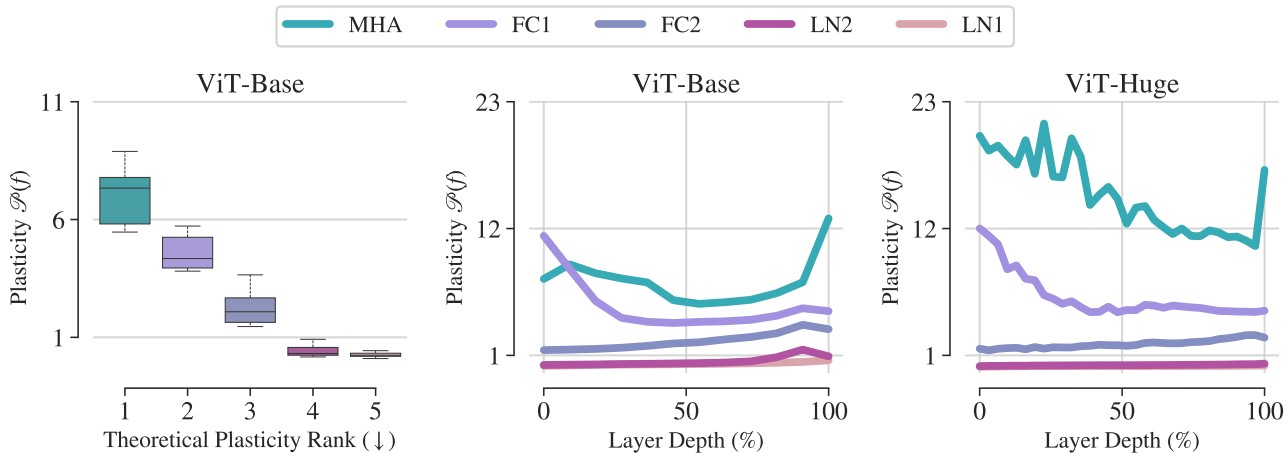

*Figure 11.* **Plasticity analysis on Gaussian Noise.** The distribution of rates of change $\|f(x) - f(y)\|_F / \|x - y\|_F$ on ViT-Base (**left**) follow the upper bound ranking predicted by our theory in Section 4. We observe along transformer blocks of ViT-Base (**middle**) that the attention module has the highest plasticity $\mathcal{P}(f)$, followed by the first and second linear layers of the feedforward. The LayerNorms are the most rigid, with a plasticity below 1. The same pattern is obtained on ViT-Huge (**right**), where the higher attention plasticity further validates our theory (see Proposition 3) since the sequence length $n$ is larger than with ViT-Base.

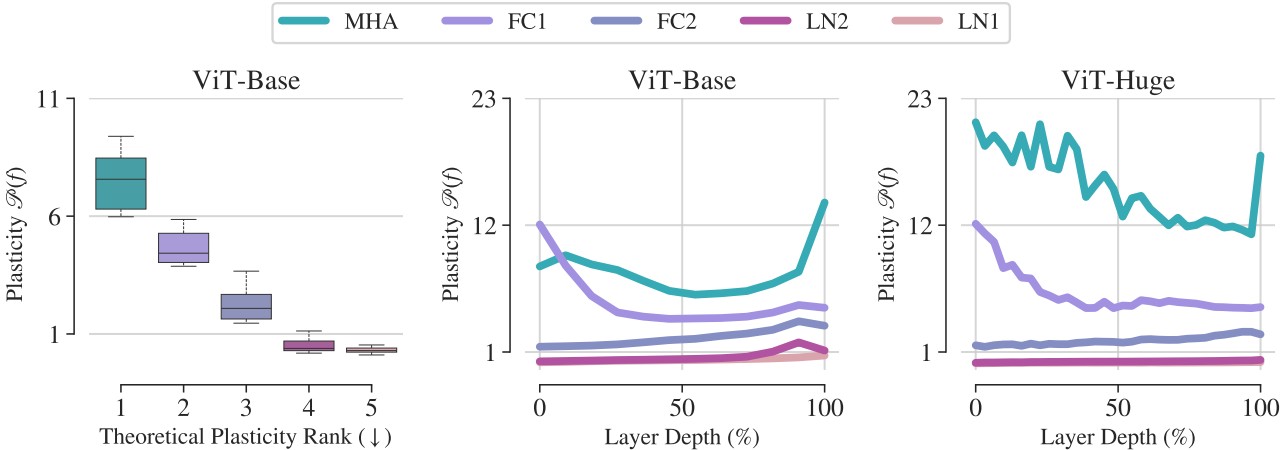

*Figure 12.* **Plasticity analysis on Motion Blur.** The distribution of rates of change $\|f(x) - f(y)\|_F / \|x - y\|_F$ on ViT-Base (**left**) follow the upper bound ranking predicted by our theory in Section 4. We observe along transformer blocks of ViT-Base (**middle**) that the attention module has the highest plasticity $\mathcal{P}(f)$, followed by the first and second linear layers of the feedforward. The LayerNorms are the most rigid, with a plasticity below 1. The same pattern is obtained on ViT-Huge (**right**), where the higher attention plasticity further validates our theory (see Proposition 3) since the sequence length $n$ is larger than with ViT-Base.

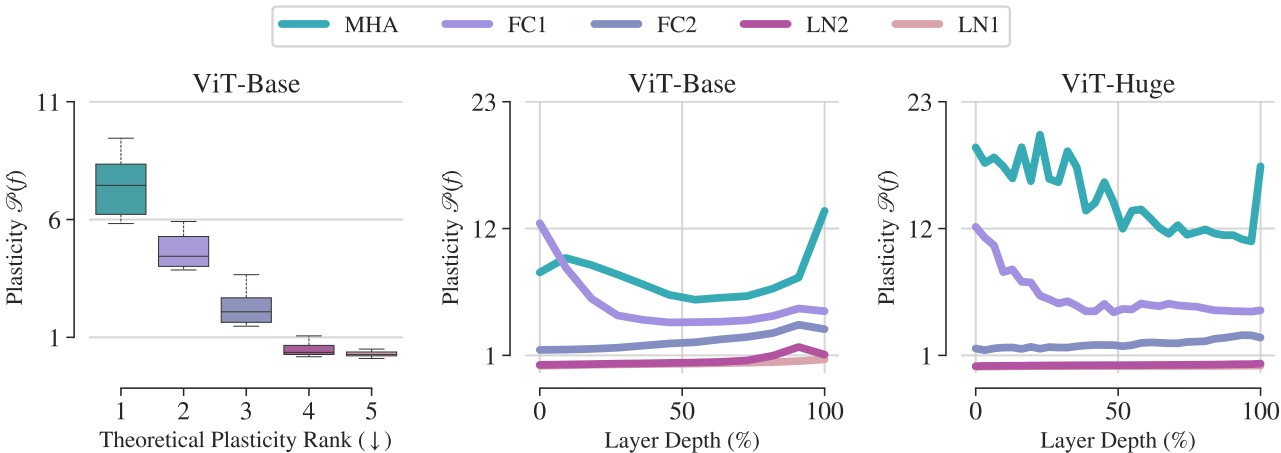

*Figure 13.* **Plasticity analysis on Snow.** The distribution of rates of change $\|f(x) - f(y)\|_{\mathrm{F}}/\|x - y\|_{\mathrm{F}}$ on ViT-Base (**left**) follow the upper bound ranking predicted by our theory in Section 4. We observe along transformer blocks of ViT-Base (**middle**) that the attention module has the highest plasticity $\mathcal{P}(f)$, followed by the first and second linear layers of the feedforward. The LayerNorms are the most rigid, with a plasticity below 1. The same pattern is obtained on ViT-Huge (**right**), where the higher attention plasticity further validates our theory (see Proposition 3) since the sequence length $n$ is larger than with ViT-Base.

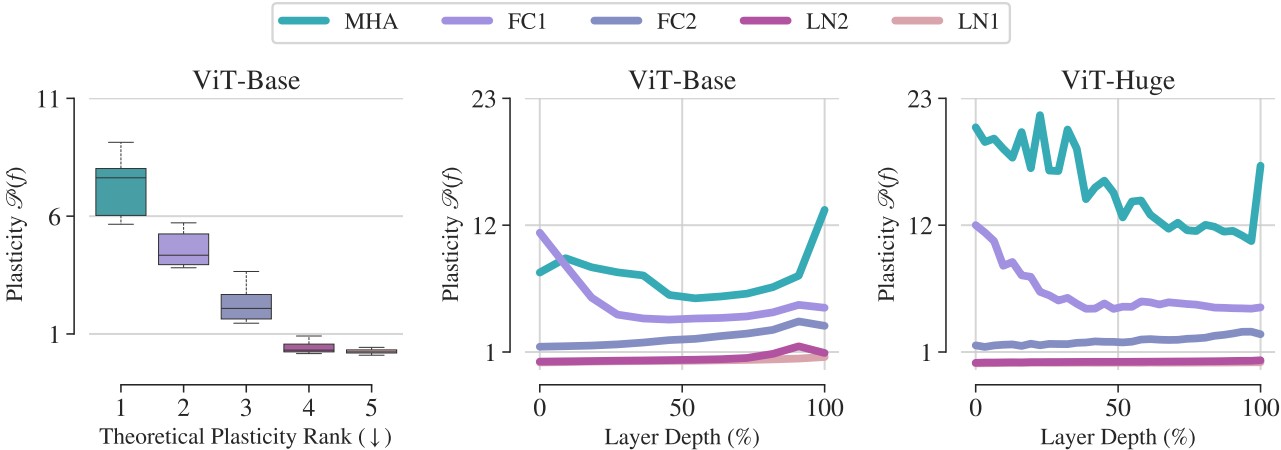

*Figure 14.* **Plasticity analysis on Speckle Noise.** The distribution of rates of change $\|f(x) - f(y)\|_{\mathrm{F}}/\|x - y\|_{\mathrm{F}}$ on ViT-Base (**left**) follow the upper bound ranking predicted by our theory in Section 4. We observe along transformer blocks of ViT-Base (**middle**) that the attention module has the highest plasticity $\mathcal{P}(f)$, followed by the first and second linear layers of the feedforward. The LayerNorms are the most rigid, with a plasticity below 1. The same pattern is obtained on ViT-Huge (**right**), where the higher attention plasticity further validates our theory (see Proposition 3) since the sequence length $n$ is larger than with ViT-Base.

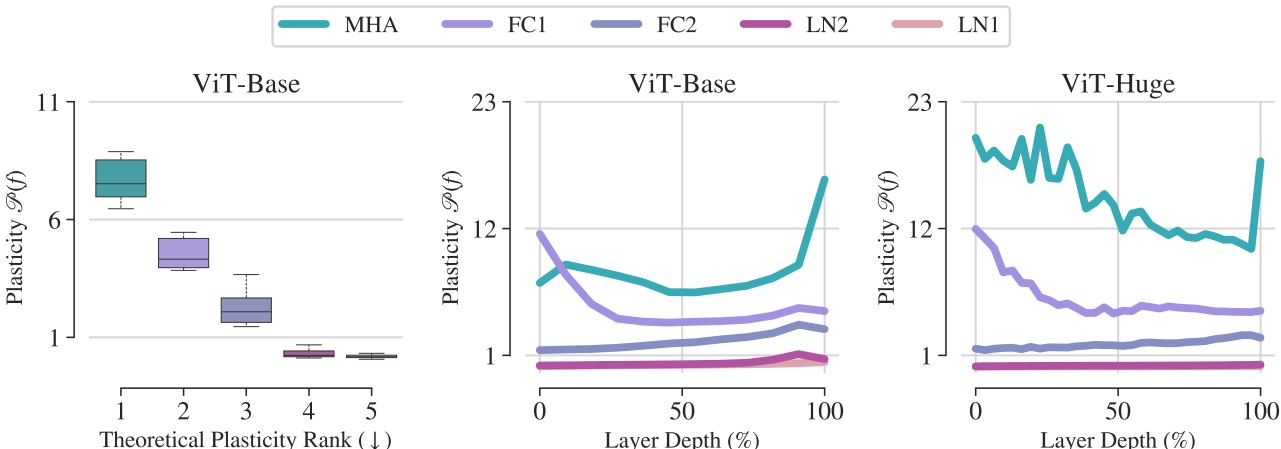

*Figure 15.* **Plasticity analysis on Clipart.** The distribution of rates of change $\|f(x) - f(y)\|_{\mathrm{F}}/\|x - y\|_{\mathrm{F}}$ on ViT-Base (**left**) follow the upper bound ranking predicted by our theory in Section 4. We observe along transformer blocks of ViT-Base (**middle**) that the attention module has the highest plasticity $\mathcal{P}(f)$, followed by the first and second linear layers of the feedforward. The LayerNorms are the most rigid, with a plasticity below 1. The same pattern is obtained on ViT-Huge (**right**), where the higher attention plasticity further validates our theory (see Proposition 3) since the sequence length $n$ is larger than with ViT-Base.

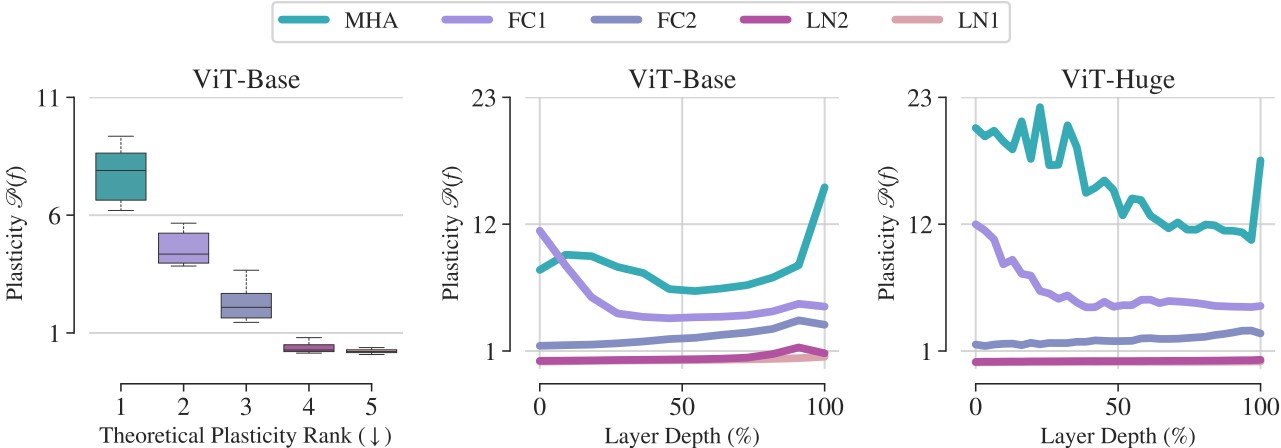

*Figure 16.* **Plasticity analysis on Flowers102.** The distribution of rates of change $\|f(x) - f(y)\|_{\mathrm{F}}/\|x - y\|_{\mathrm{F}}$ on ViT-Base (**left**) follow the upper bound ranking predicted by our theory in Section 4. We observe along transformer blocks of ViT-Base (**middle**) that the attention module has the highest plasticity $\mathcal{P}(f)$, followed by the first and second linear layers of the feedforward. The LayerNorms are the most rigid, with a plasticity below 1. The same pattern is obtained on ViT-Huge (**right**), where the higher attention plasticity further validates our theory (see Proposition 3) since the sequence length $n$ is larger than with ViT-Base.

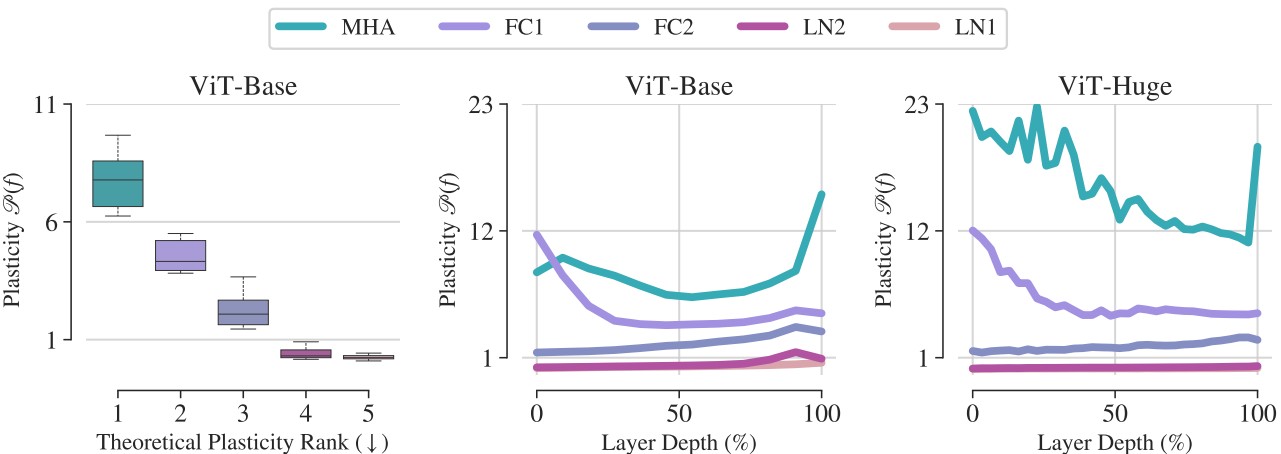

*Figure 17.* **Plasticity analysis on Pet.** The distribution of rates of change $\|f(x) - f(y)\|_F / \|x - y\|_F$ on ViT-Base (**left**) follow the upper bound ranking predicted by our theory in Section 4. We observe along transformer blocks of ViT-Base (**middle**) that the attention module has the highest plasticity $\mathcal{P}(f)$, followed by the first and second linear layers of the feedforward. The LayerNorms are the most rigid, with a plasticity below 1. The same pattern is obtained on ViT-Huge (**right**), where the higher attention plasticity further validates our theory (see Proposition 3) since the sequence length $n$ is larger than with ViT-Base.

### E.1.3. PLASTICITY EXPERIMENTS ON DINOv3 AND GPT2

We extend our analysis to DINOv3 (Siméoni et al., 2025), a 7B-parameter vision transformer trained in a self-supervised fashion, and GPT2 (Radford et al., 2019), a 124M-parameter decoder-only language model. Since ImageNet22k is part of the pretrained set of DINOv3, we follow the same experimental setup as with pretrained ViTs and report the results on Cifar10. To extend the analysis to GPT2, we use the fact that the pretraining set WebText (Radford et al., 2019) was built by scraping the web and includes all high-quality outbound links from Reddit (at least 3 karma). Since it contains mainstream news articles, political commentary, and tech journalism, this motivates us to use AGNews (Zhang et al., 2015), a dataset of news articles about the world, sports, business, and science, as a representative proxy for the pre-training data distribution. Moreover, the authors explicitly "*removed all Wikipedia documents* (Radford et al., 2019) from WebText. This allows us to consider the WikiText-103 (Merity et al., 2017) as the downstream dataset. We display the distribution and evolution of plasticity across layers of DINOv3 and GPT2 in Fig. 18. The observed patterns are consistent with those of supervised ViTs shown in Fig. 3.

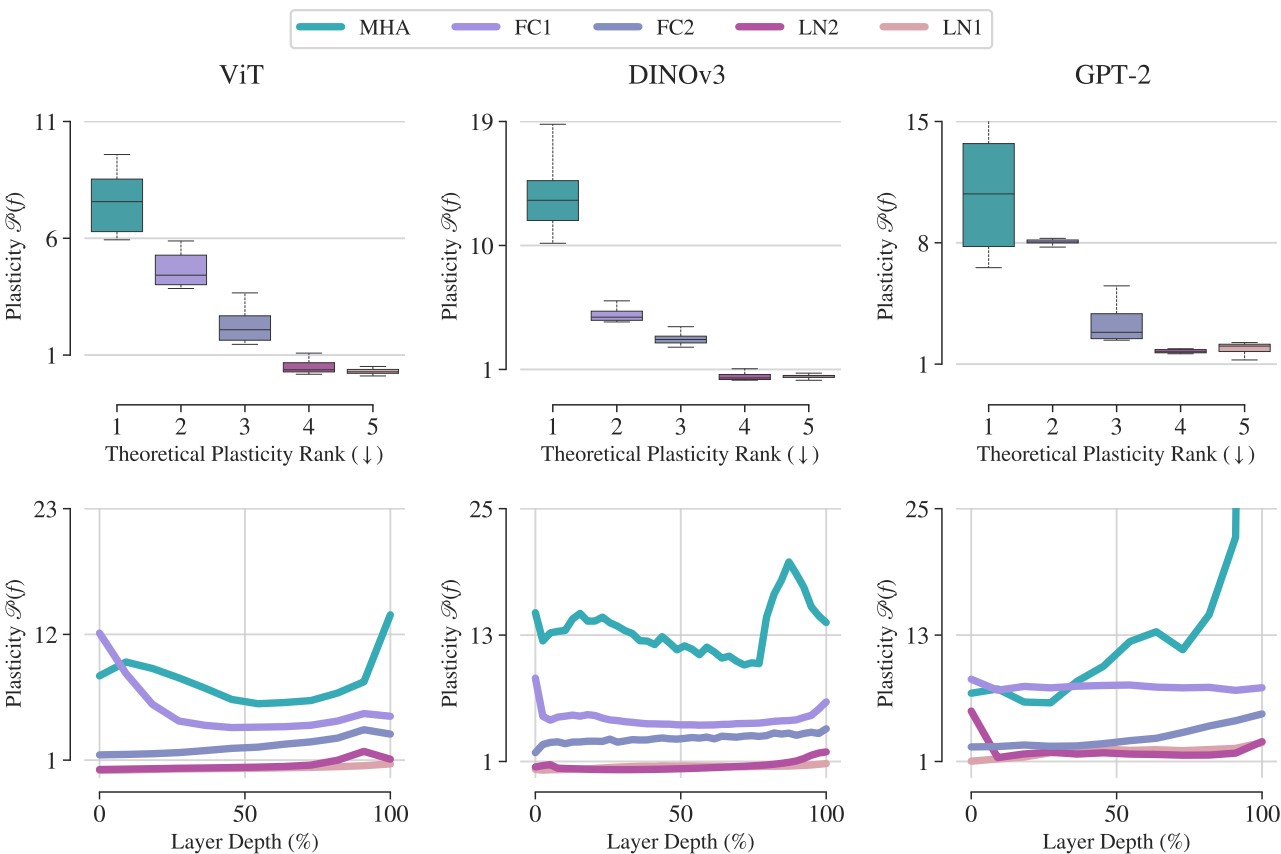

*Figure 18.* **Plasticity analysis on ViT-Base, DINOv3, and GPT2.** The distribution of rates of change $\|f(x) - f(y)\|_{\mathrm{F}} / \|x - y\|_{\mathrm{F}}$ on ViT-Base (**top left**), DINOv3-7B (**top middle**), and GPT2 (**top right**) follows the theoretical ranking of Section 4. We observe along transformer blocks of ViT-Base (**top left**), DINOv3-7B (**bottom middle**), and GPT2 (**bottom right**) that the attention module has the highest plasticity $\mathscr{P}(f)$, followed by the first and second linear layers of the feedforward. The LayerNorms are the most rigid, with a plasticity below 1.

## E.2. Finetuning analysis

In this section, we provide the additional results, figures, and experiments related to Section 5.2.

### E.2.1. PERFORMANCE COMPARISON

In Table 8, we gather the finetuning performance for each configuration and dataset. It is used to obtain Fig. 2 (right). The relative gain metric is computed as the percentage improvement between the finetuning and the linear probing performance. For visualization purposes, we display in Fig. 19 the overall performance of each configuration on the 11 benchmarks. We

*Table 8.* **Full finetuning results.** We investigate the benefits of plasticity by evaluating finetuning the trainable components of Table 5 on a diverse set of 11 image classification benchmarks. We report the best top-1 accuracy (%) on the test set over the learning rate grid of each dataset (↑). Entries show the mean and standard deviation over three finetuning runs with different seeds. Transformer components are ordered in terms of decreasing plasticity. The best overall performance among the transformer components configurations of Table 5 is in **bold**. The *full-finetuning* configuration lets the 86M parameters of ViT-Base be trainable. The *linear probing* performance is obtained over the hidden representation of the last layer following Caron et al. (2021).

| configuration | MHA | FC1 | FC2 | LN2 | LN1 | *full finetuning* | *linear probing* |
|---|---|---|---|---|---|---|---|
| Cifar10 | $98.91_{\pm0.07}$ | $99.09_{\pm0.05}$ | $98.91_{\pm0.06}$ | $98.72_{\pm0.05}$ | $98.67_{\pm0.03}$ | $99.02_{\pm0.02}$ | 92.07 |
| Cifar100 | $92.65_{\pm0.07}$ | $92.85_{\pm0.07}$ | $92.31_{\pm0.11}$ | $91.93_{\pm0.11}$ | $91.43_{\pm0.07}$ | $92.74_{\pm0.05}$ | 67.56 |
| Contrast | $97.09_{\pm0.11}$ | $97.06_{\pm0.08}$ | $96.28_{\pm0.11}$ | $96.67_{\pm0.20}$ | $96.89_{\pm0.19}$ | $97.23_{\pm0.18}$ | 75.55 |
| Gaussian Noise | $89.41_{\pm0.53}$ | $89.49_{\pm0.16}$ | $88.49_{\pm0.51}$ | $89.55_{\pm0.04}$ | $88.99_{\pm0.24}$ | $87.14_{\pm1.16}$ | 50.35 |
| Motion Blur | $94.72_{\pm0.21}$ | $94.53_{\pm0.06}$ | $94.04_{\pm0.16}$ | $93.95_{\pm0.34}$ | $93.25_{\pm0.29}$ | $94.67_{\pm0.14}$ | 62.75 |
| Snow | $95.47_{\pm0.13}$ | $95.52_{\pm0.20}$ | $95.27_{\pm0.29}$ | $95.51_{\pm0.11}$ | $95.15_{\pm0.10}$ | $95.42_{\pm0.13}$ | 61.85 |
| Speckle Noise | $90.07_{\pm0.32}$ | $89.85_{\pm0.34}$ | $89.22_{\pm0.31}$ | $89.71_{\pm0.17}$ | $89.74_{\pm0.31}$ | $89.58_{\pm0.43}$ | 52.10 |
| Clipart | $77.31_{\pm0.41}$ | $76.47_{\pm0.24}$ | $76.54_{\pm0.17}$ | $74.37_{\pm0.08}$ | $74.65_{\pm0.16}$ | $78.50_{\pm0.49}$ | 45.86 |
| Sketch | $69.23_{\pm0.05}$ | $69.31_{\pm0.18}$ | $69.49_{\pm0.20}$ | $65.27_{\pm0.15}$ | $65.76_{\pm0.10}$ | $71.30_{\pm0.26}$ | 30.16 |
| Flowers102 | $99.03_{\pm0.08}$ | $99.05_{\pm0.06}$ | $98.86_{\pm0.06}$ | $99.21_{\pm0.07}$ | $98.99_{\pm0.20}$ | $99.15_{\pm0.05}$ | 96.62 |
| Pet | $94.37_{\pm0.13}$ | $94.26_{\pm0.26}$ | $93.98_{\pm0.20}$ | $94.39_{\pm0.13}$ | $94.46_{\pm0.11}$ | $94.57_{\pm0.29}$ | 89.18 |
| Avg. | **90.75** | 90.68 | 90.31 | 89.93 | 89.82 | 90.90 | 64.22 |

observe similar patterns than with the relative gain in Fig. 2 (right): the higher the plasticity, the better the finetuning.

### E.2.2. EXTENSION TO VIT-LARGE (307M)

We extend the finetuning analysis to ViT-Large (307M parameters) and obtain similar conclusions. Results on Clipart are displayed in Table 9.

*Table 9.* **Consistent benefits across model sizes.** We report the average top-1 accuracy (%) on the test set over the learning rate grid of Clipart (↑). Entries show the mean and standard deviation over learning rates. The best overall performance among the transformer components configurations of Table 5 is in **bold** and non-smooth components are highlighted in gray.

| configuration | MHA | FC1 | FC2 | LN2 | LN1 |
|---|---|---|---|---|---|
| ViT-Base (86M) | **76.9** $_{\pm0.6}$ | 75.7 $_{\pm0.6}$ | 75.9 $_{\pm0.8}$ | 73.6 $_{\pm0.8}$ | 74.0 $_{\pm0.5}$ |
| ViT-Large (307M) | **78.4** $_{\pm0.2}$ | 77.3 $_{\pm0.2}$ | 78.3 $_{\pm0.3}$ | 76.1 $_{\pm0.3}$ | 76.8 $_{\pm0.3}$ |

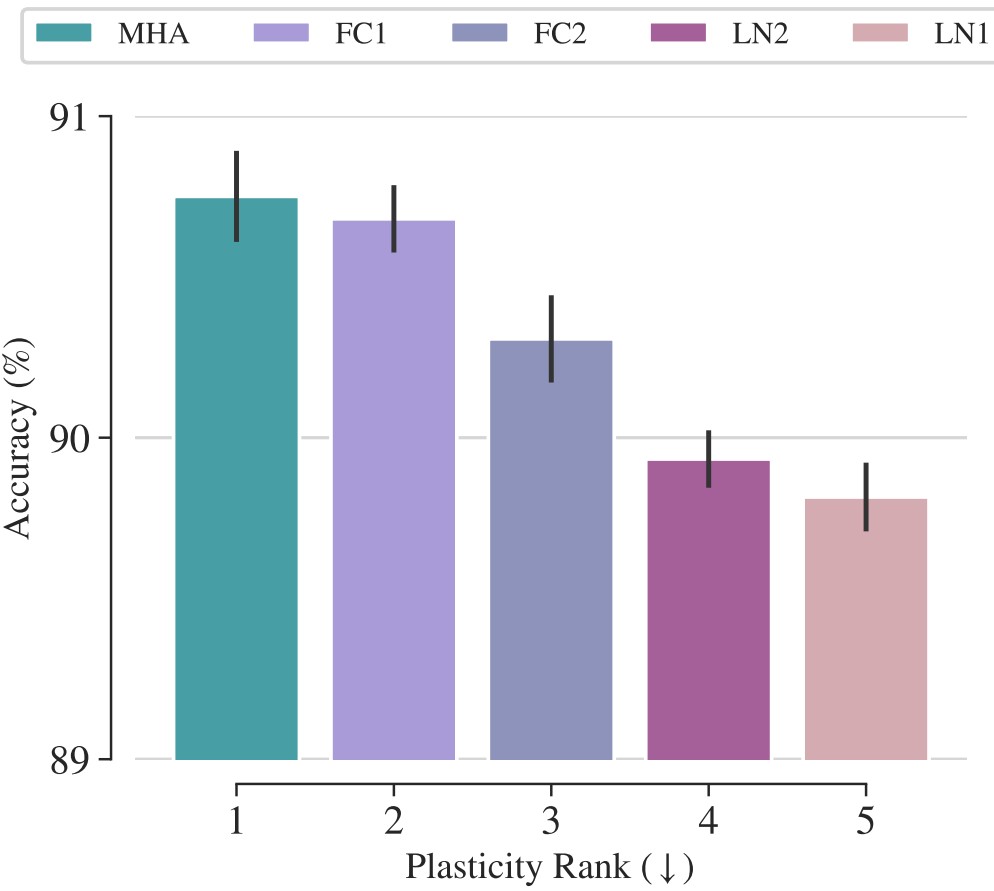

*Figure 19.* **Better performance (11 benchmarks).** We compare transformer components, ordered in terms of decreasing plasticity, and report the average top-1 accuracy over a diverse set of 11 benchmarks, with the pooled standard error computed over 3 finetuning runs. We can see that high plasticity results in better performance.

### E.2.3. ROBUSTNESS ANALYSIS

In Fig. 20, we display the finetuning performance over learning rates and seeds for all benchmarks. Overall, we observe similar patterns to those in Fig. 5 (left), with plastic components resulting in more stable performance. In particular, we consistently see that finetuning the attention module leads to better and more stable performance.

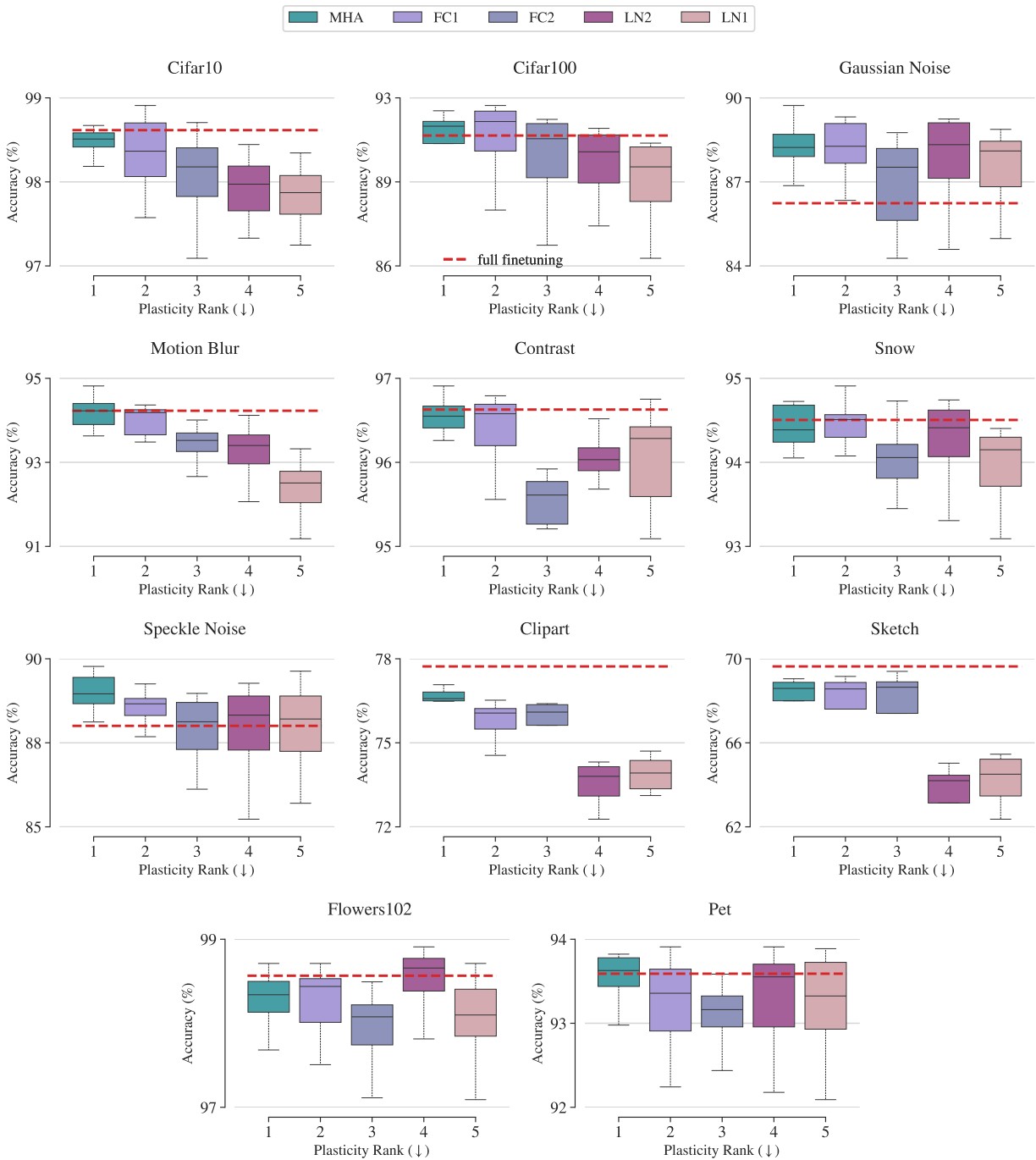

*Figure 20.* **Robustness comparison (11 benchmarks).** We display the distribution of the finetuning performance over the learning rates from Table 7 and 3 seeds relative to network initialization and dataloaders. We compare transformer components, ordered in terms of decreasing plasticity. Overall, plastic components result in more stable performance. In particular, the attention module consistently displays a small performance variation.

### E.2.4. GRADIENT NORM ANALYSIS

In Figs. 21 to 53, we display the evolution of the gradient norms and validation loss on all benchmarks, learning rates and seeds. We consistently see that the scale of gradient norms and validation loss descent follows the plasticity ranking from Section 5.1: we observe a faster and better convergence for plastic components, with even more salient benefits on challenging datasets such as Cifar100, Clipart, and Sketch. In particular, we observe lower gradient norms and a less steep descent for the LayerNorms, which is aggravated for low learning rates $\eta$. This showcases the plasticity of LayerNorms to the choice of learning rates compared to components with higher plasticity. For components with high plasticity, we observe a rather expected training evolution for low learning rates with increasing gradient norms and decreasing validation loss. However, for higher learning rates, we can see that the gradient norms first increase in a steep fashion before slowly decreasing. This can be understood by the model escaping the pretraining minima, passing through sharp regions of the loss landscape before converging to a flat local minima. This behavior can be seen on Clipart Fig. 42, for instance, and is particularly salient for the attention module. This is reminiscent of Park & Kim (2022), who showed that the multihead self-attention module flattens the loss landscape, which leads to better generalization (Chen et al., 2022; Foret et al., 2021; Ilbert et al., 2024).

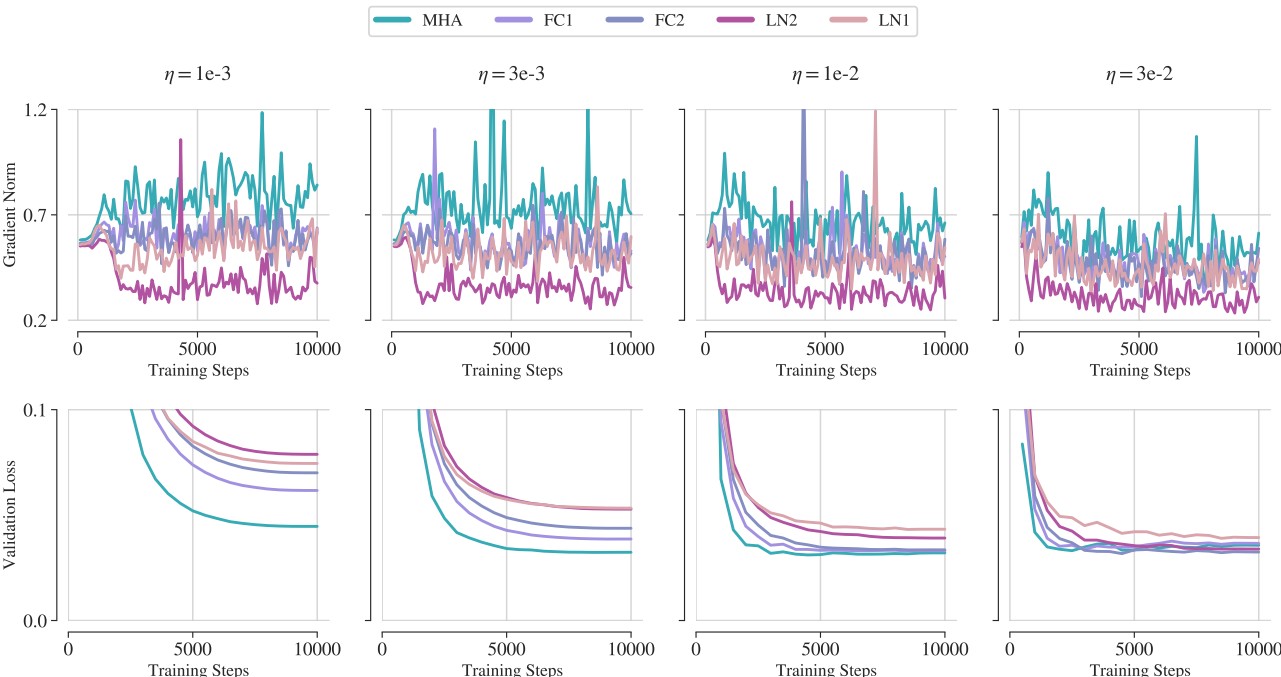

*Figure 21.* **Training dynamics on Cifar10 with seed** 0. We display the evolution during training of the gradient norms (**top**) and the validation loss (**bottom**) of each finetuning configuration of Table 5, with increasing learning rate $\eta$ from **left to right**. Components are ordered in terms of decreasing plasticity in the legend. Plastic components have higher gradient norms, which leads to a steeper descent in the validation loss and better downstream performance. The benefits of plasticity are even more salient with low learning rates. Overall, higher plasticity leads to better optimization and generalization.

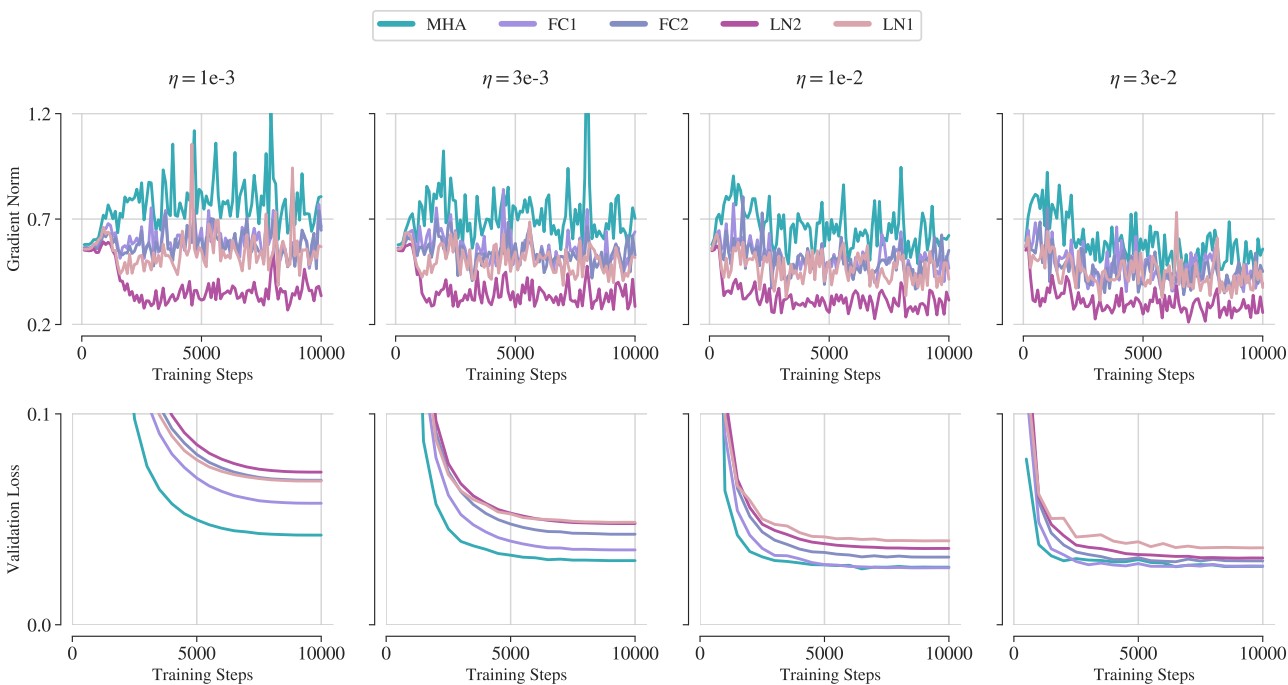

*Figure 22.* **Training dynamics on Cifar10 with seed** 42**.** Akin to Fig. 21, we observe the same consistent pattern of faster and better convergence for components with high plasticity.

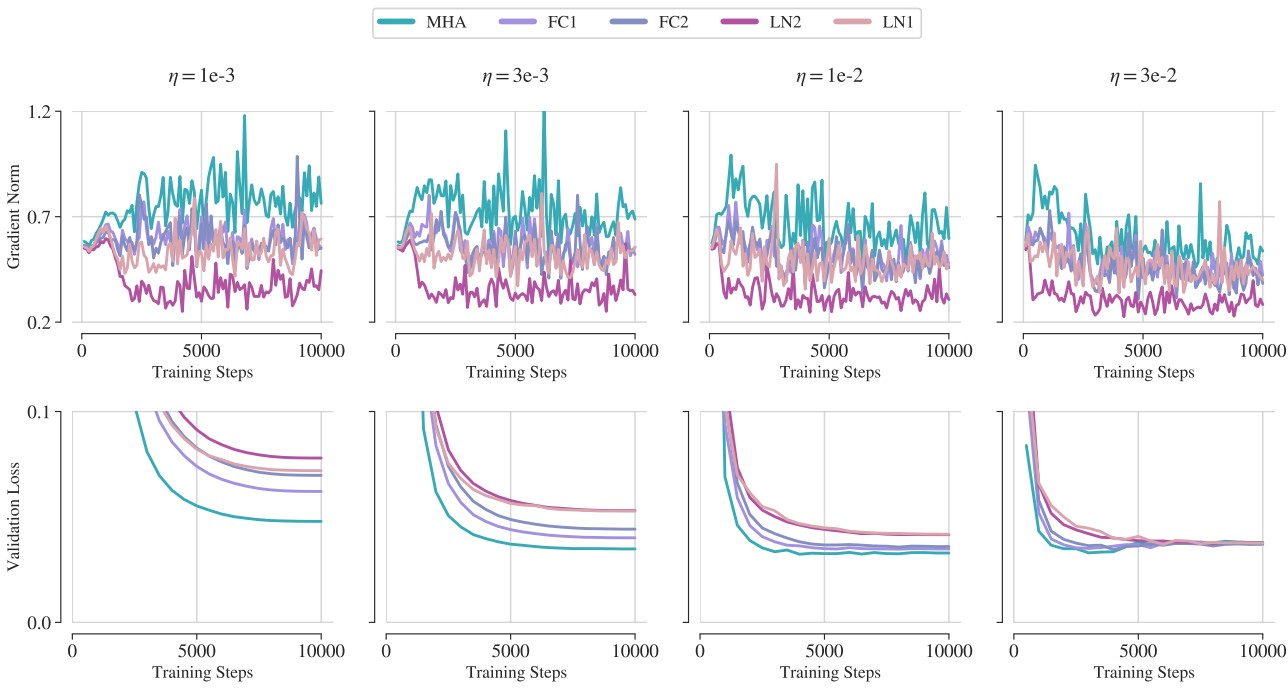

*Figure 23.* **Training dynamics on Cifar10 with seed** 3407**.** Akin to Fig. 21, we observe the same consistent pattern of faster and better convergence for components with high plasticity.

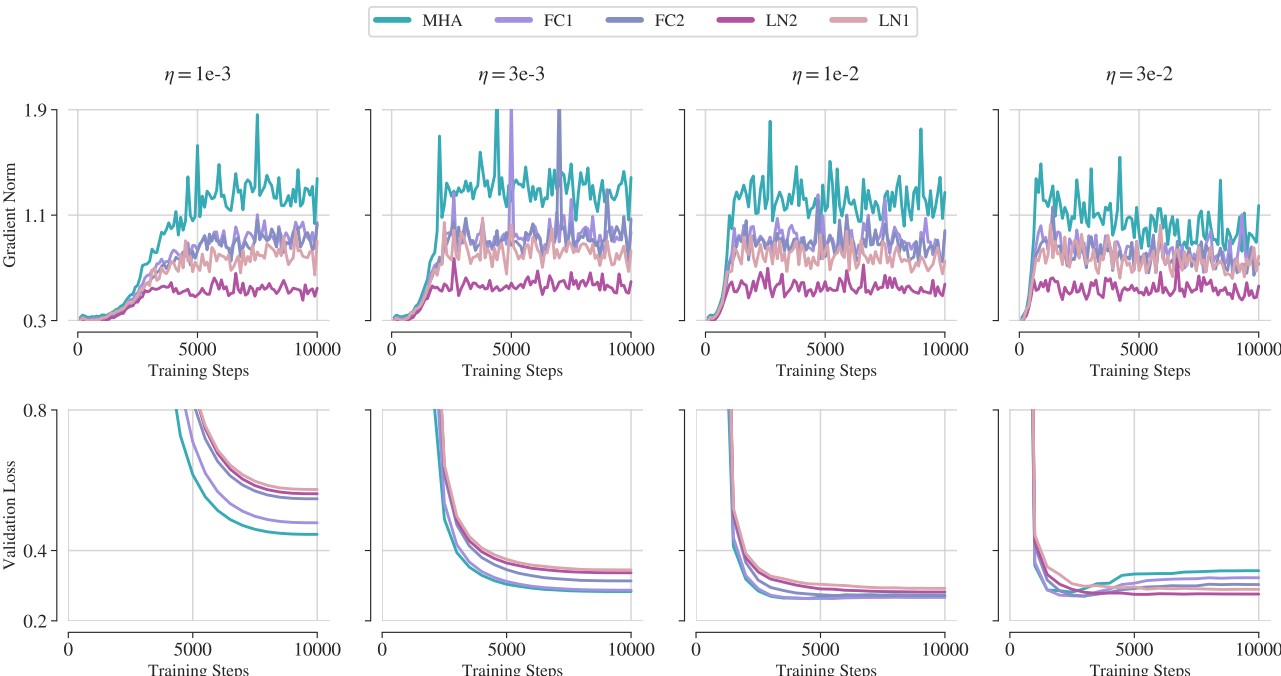

*Figure 24.* **Training dynamics on Cifar100 with seed** 0. We display the evolution during training of the gradient norms (**top**) and the validation loss (**bottom**) of each finetuning configuration of Table 5, with increasing learning rate $\eta$ from **left to right**. Components are ordered in terms of decreasing plasticity in the legend. Plastic components have higher gradient norms, which leads to a steeper descent in the validation loss and better downstream performance. The benefits of plasticity are even more salient with low learning rates. Overall, higher plasticity leads to better optimization and generalization.

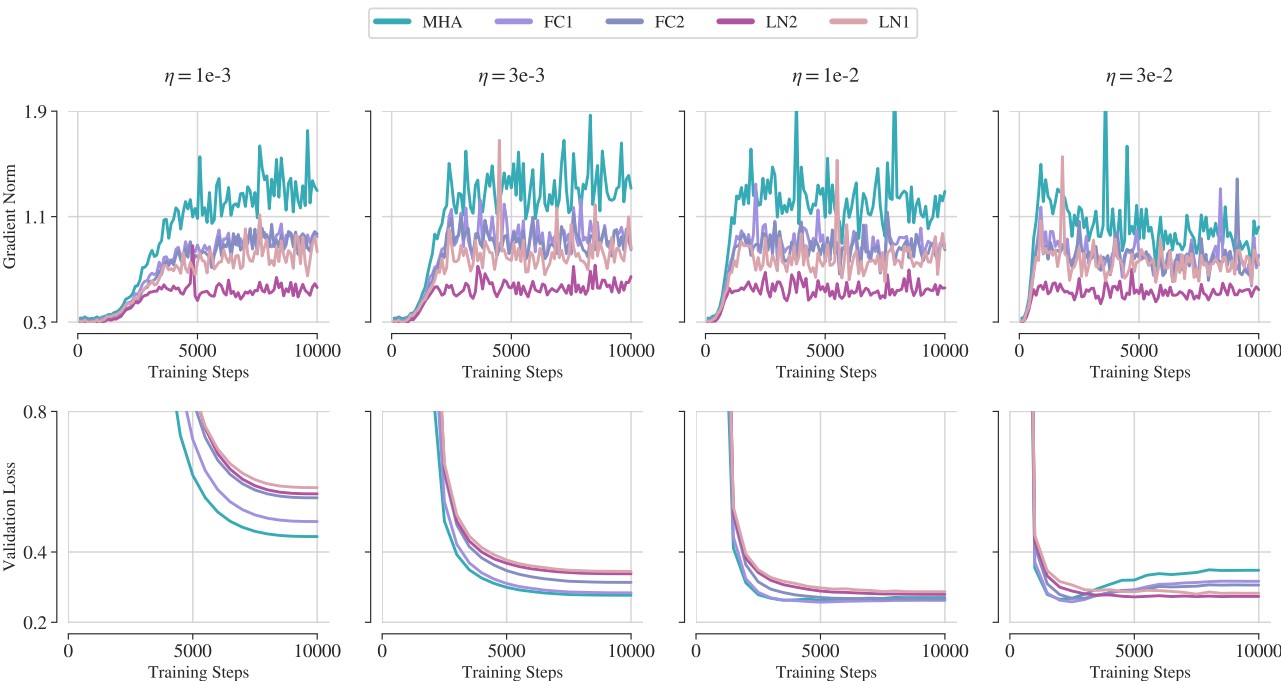

*Figure 25.* **Training dynamics on Cifar100 with seed** 42. Akin to Fig. 24, we observe the same consistent pattern of faster and better convergence for components with high plasticity.

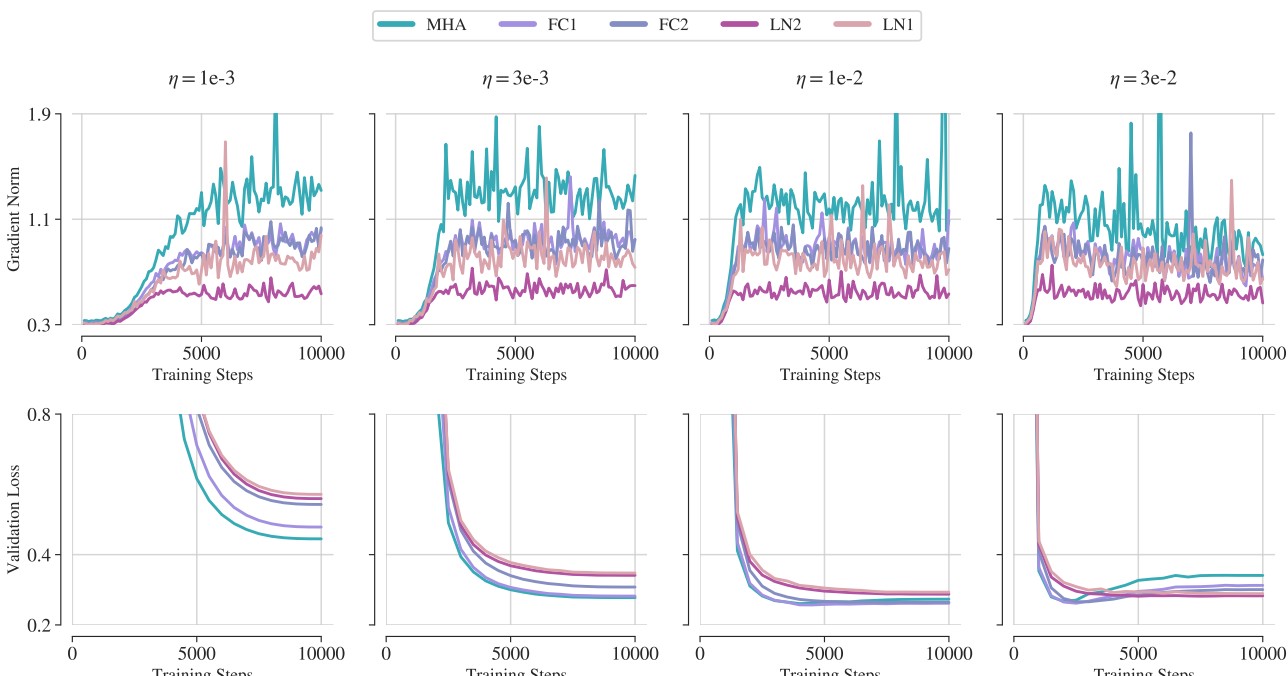

*Figure 26.* **Training dynamics on Cifar100 with seed** 3407**.** Akin to Fig. 24, we observe the same consistent pattern of faster and better convergence for components with high plasticity.

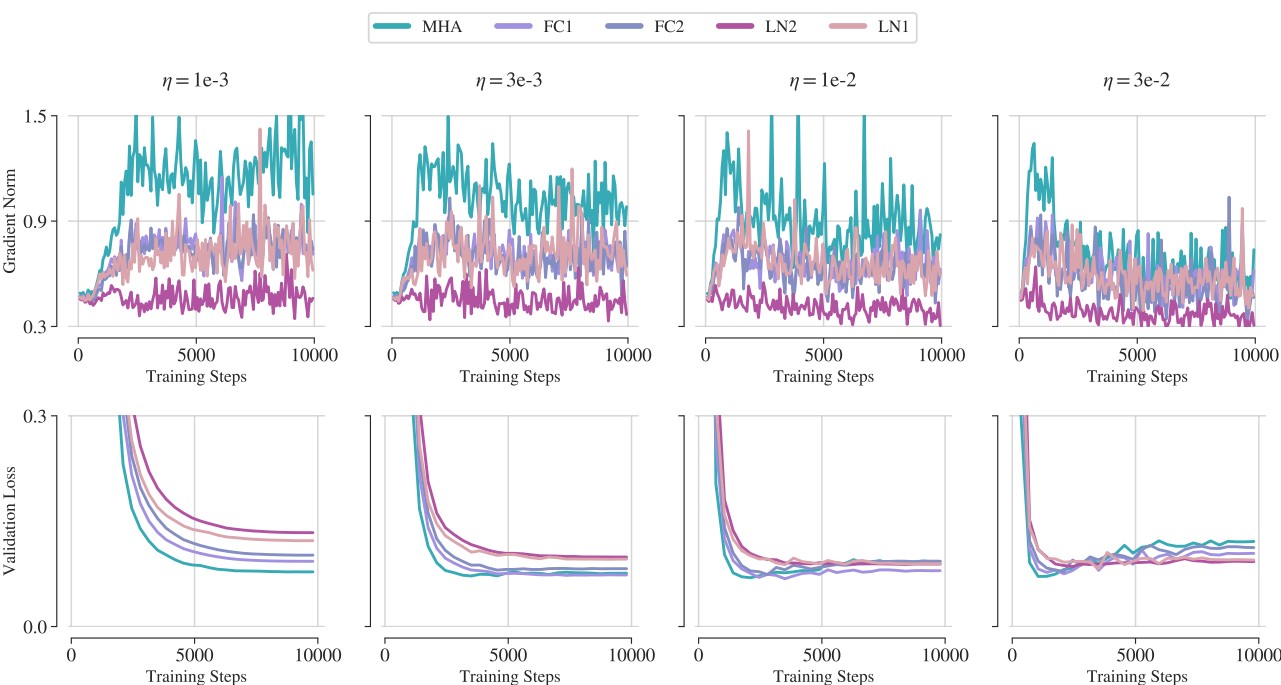

*Figure 27.* **Training dynamics on Contrast with seed** 0**.** We display the evolution during training of the gradient norms (**top**) and the validation loss (**bottom**) of each finetuning configuration of Table 5, with increasing learning rate $\eta$ from **left to right**. Components are ordered in terms of decreasing plasticity in the legend. Plastic components have higher gradient norms, which leads to a steeper descent in the validation loss and better downstream performance. The benefits of plasticity are even more salient with low learning rates. Overall, higher plasticity leads to better optimization and generalization.

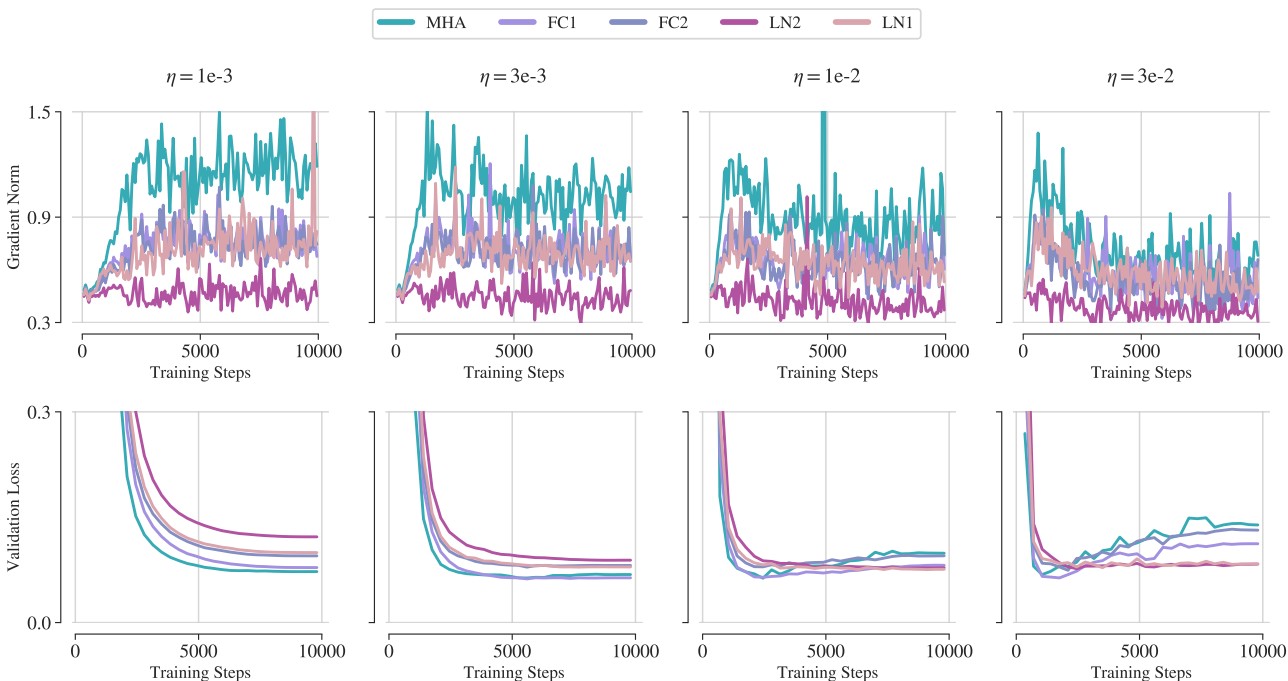

*Figure 28.* **Training dynamics on Contrast with seed** 42**.** Akin to Fig. 27, we observe the same consistent pattern of faster and better convergence for components with high plasticity.

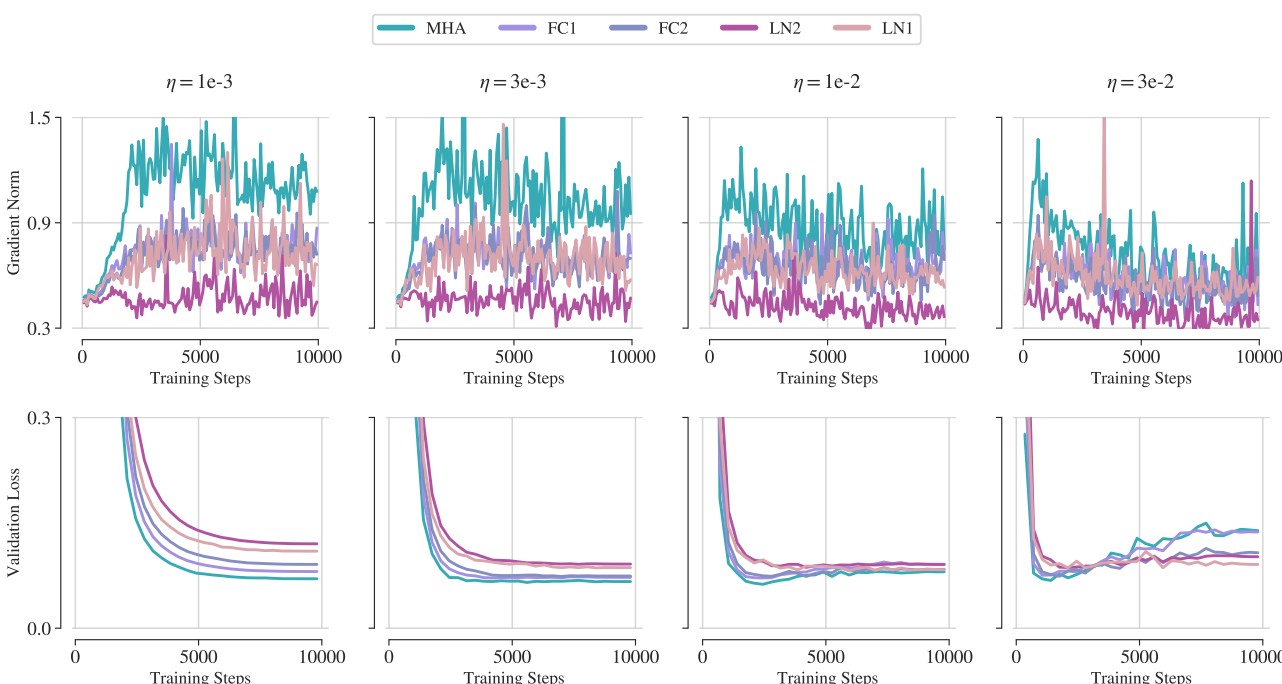

*Figure 29.* **Training dynamics on Contrast with seed** 3407**.** Akin to Fig. 27, we observe the same consistent pattern of faster and better convergence for components with high plasticity.

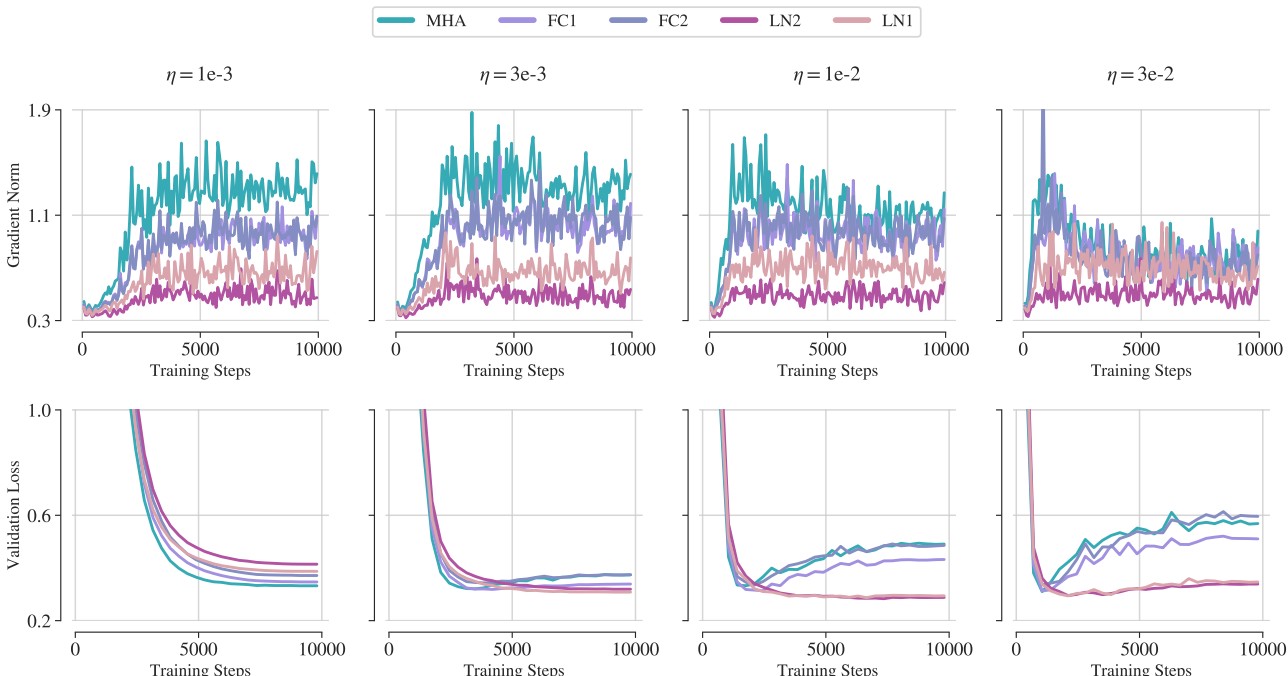

*Figure 30.* **Training dynamics on Gaussian Noise with seed** 0. We display the evolution during training of the gradient norms (**top**) and the validation loss (**bottom**) of each finetuning configuration of Table 5, with increasing learning rate $\eta$ from **left to right**. Components are ordered in terms of decreasing plasticity in the legend. Plastic components have higher gradient norms, which leads to a steeper descent in the validation loss and better downstream performance. The benefits of plasticity are even more salient with low learning rates. Overall, higher plasticity leads to better optimization and generalization.

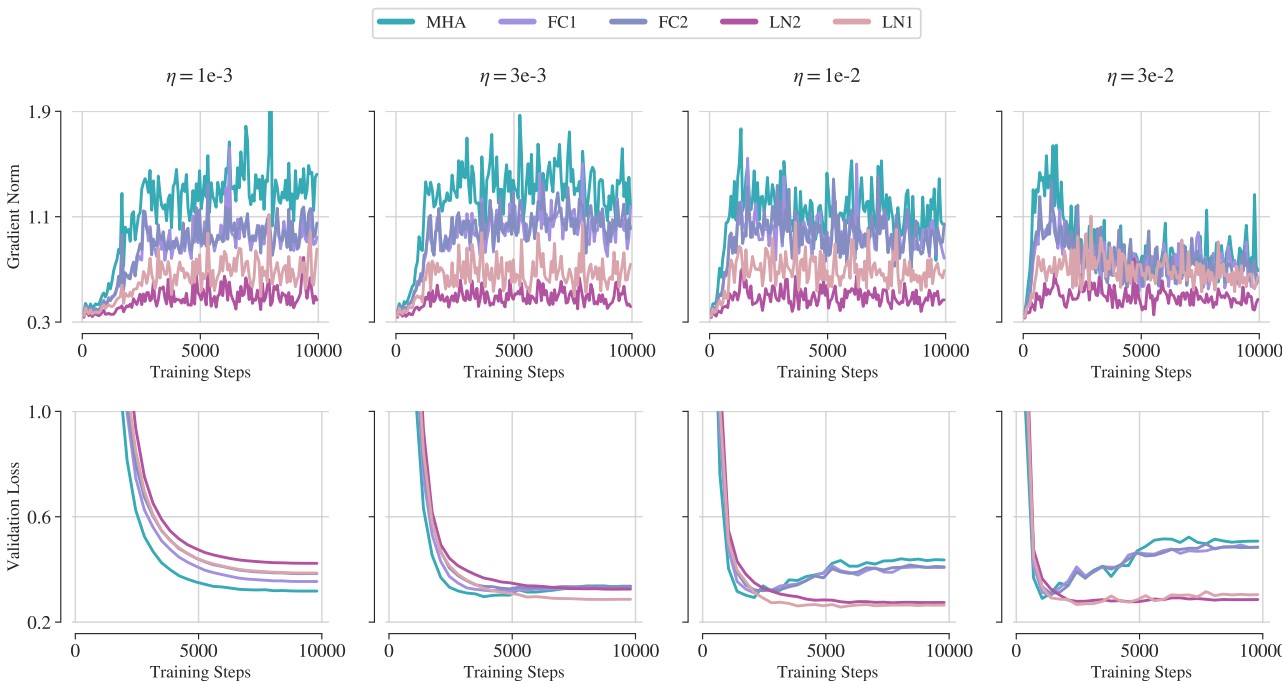

*Figure 31.* **Training dynamics on Gaussian Noise with seed** 42. Akin to Fig. 30, we observe the same consistent pattern of faster and better convergence for components with high plasticity.

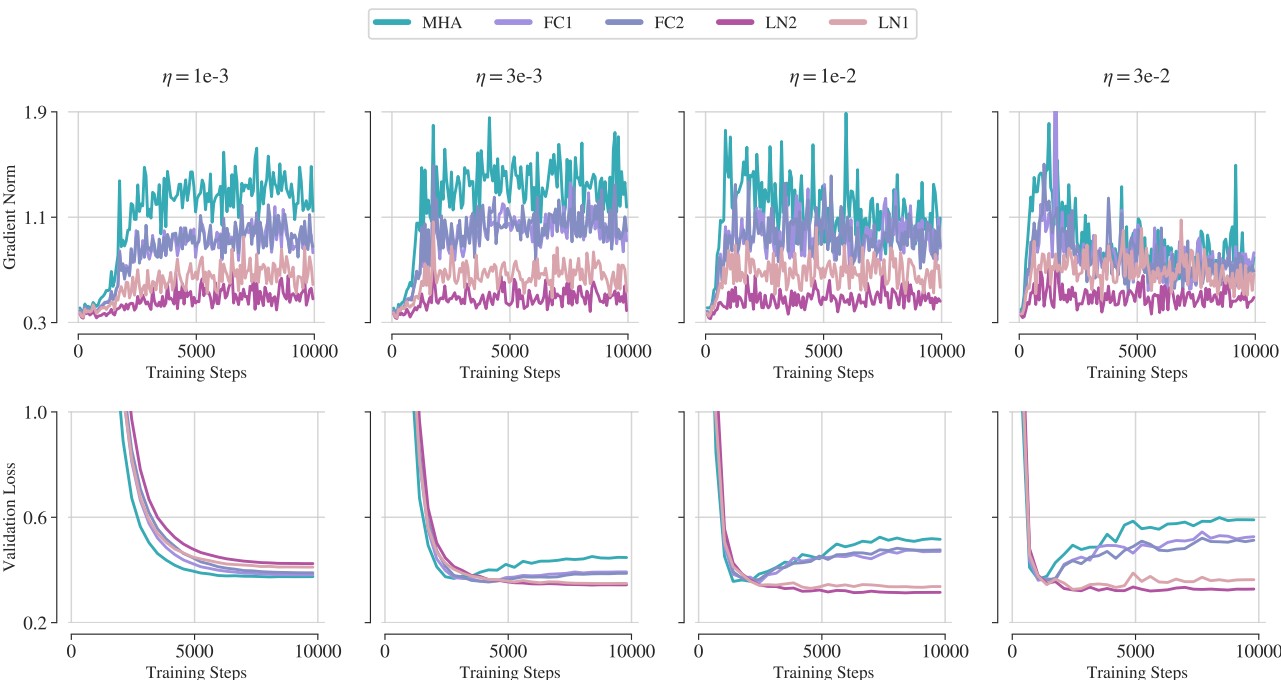

*Figure 32.* **Training dynamics on Gaussian Noise with seed** 3407**.** Akin to Fig. 30, we observe the same consistent pattern of faster and better convergence for components with high plasticity.

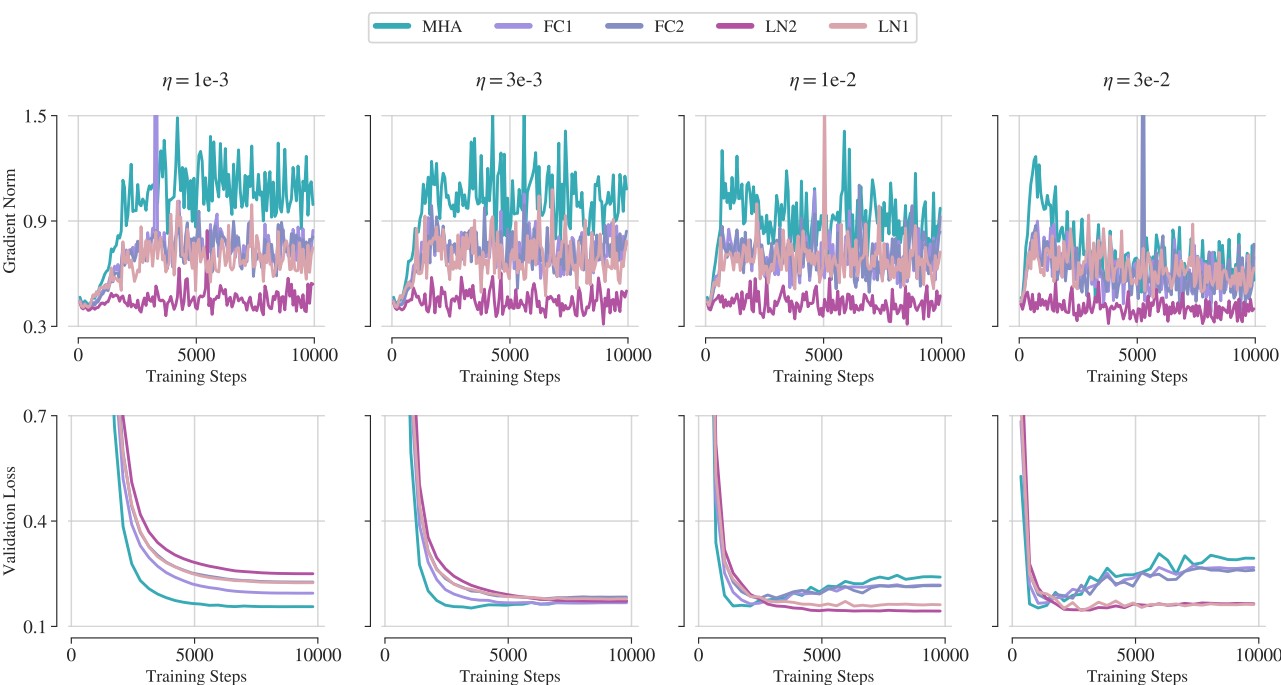

*Figure 33.* **Training dynamics on Motion Blur with seed** 0**.** We display the evolution during training of the gradient norms (**top**) and the validation loss (**bottom**) of each finetuning configuration of Table 5, with increasing learning rate $\eta$ from **left to right**. Components are ordered in terms of decreasing plasticity in the legend. Plastic components have higher gradient norms, which leads to a steeper descent in the validation loss and better downstream performance. The benefits of plasticity are even more salient with low learning rates. Overall, higher plasticity leads to better optimization and generalization.

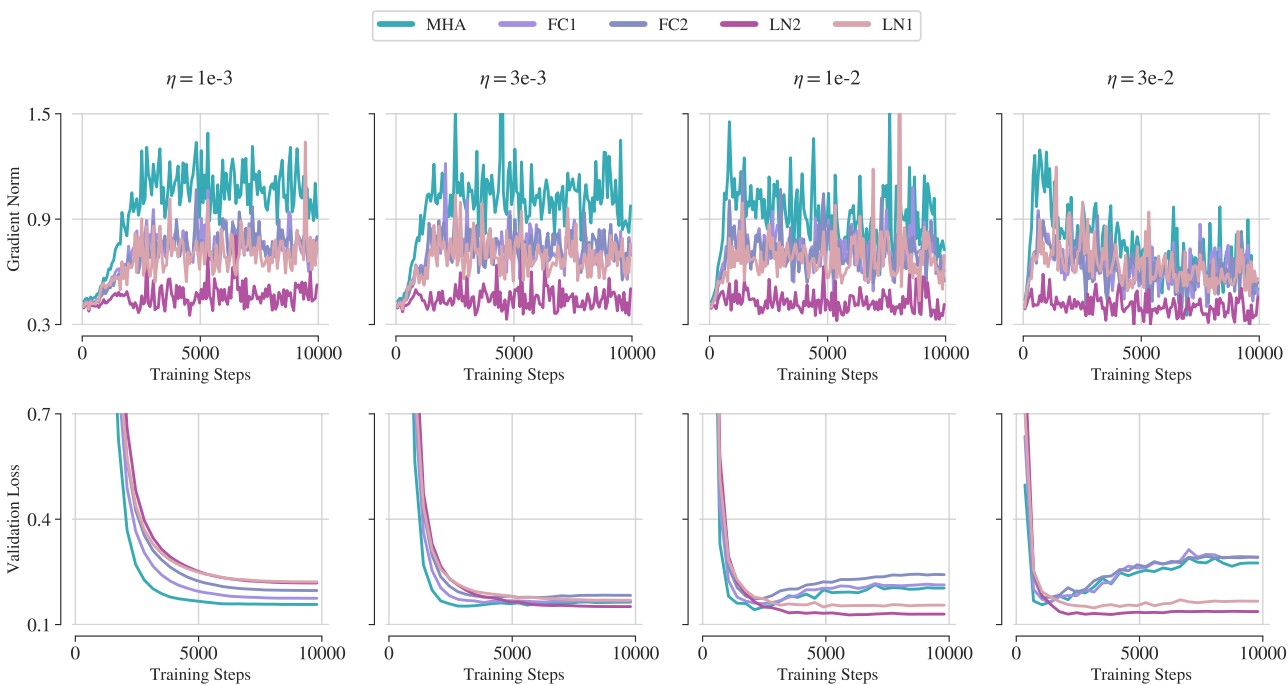

*Figure 34.* **Training dynamics on Motion Blur with seed** 42. Akin to Fig. 33, we observe the same consistent pattern of faster and better convergence for components with high plasticity.

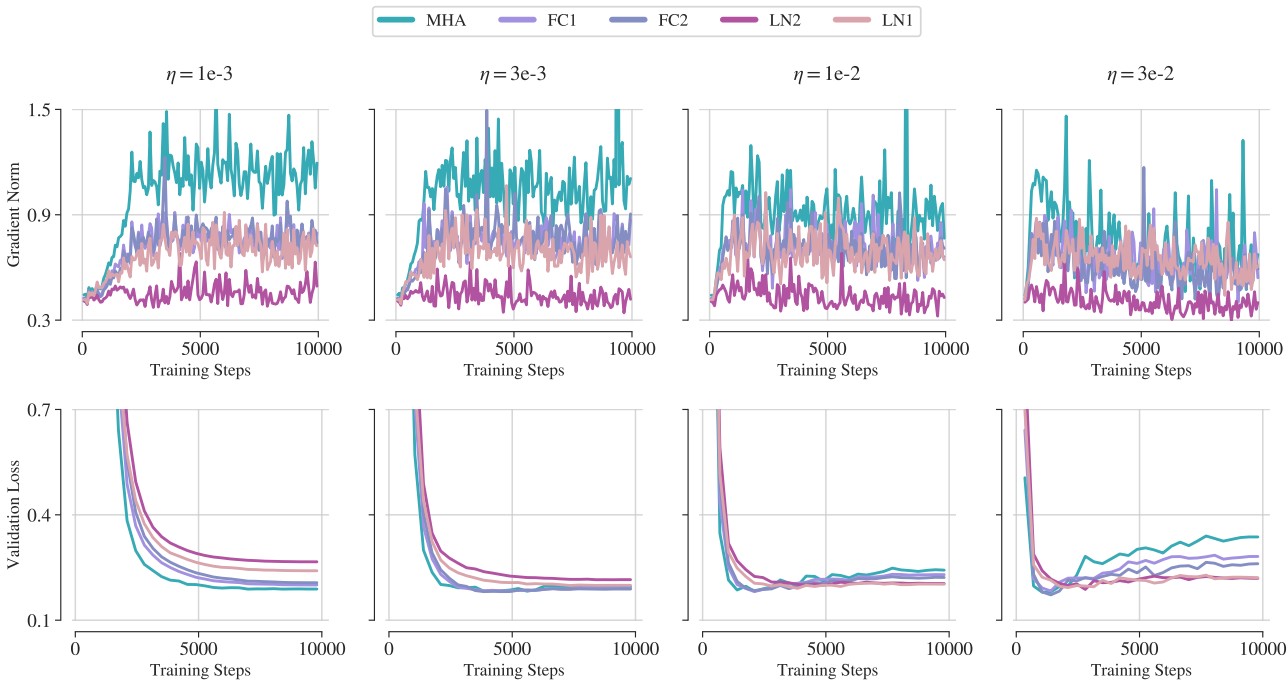

*Figure 35.* **Training dynamics on Motion Blur with seed** 3407. Akin to Fig. 33, we observe the same consistent pattern of faster and better convergence for components with high plasticity.

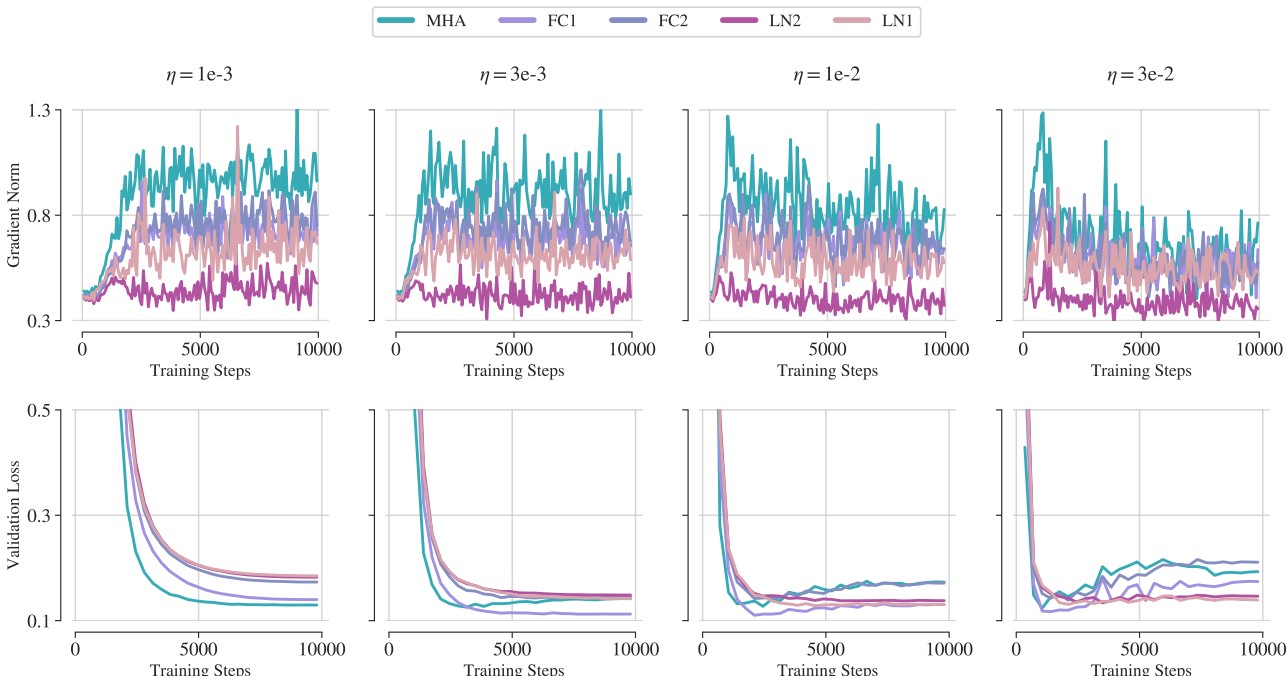

*Figure 36.* **Training dynamics on Snow with seed** 0. We display the evolution during training of the gradient norms (**top**) and the validation loss (**bottom**) of each finetuning configuration of Table 5, with increasing learning rate $\eta$ from **left to right**. Components are ordered in terms of decreasing plasticity in the legend. Plastic components have higher gradient norms, which leads to a steeper descent in the validation loss and better downstream performance. The benefits of plasticity are even more salient with low learning rates. Overall, higher plasticity leads to better optimization and generalization.

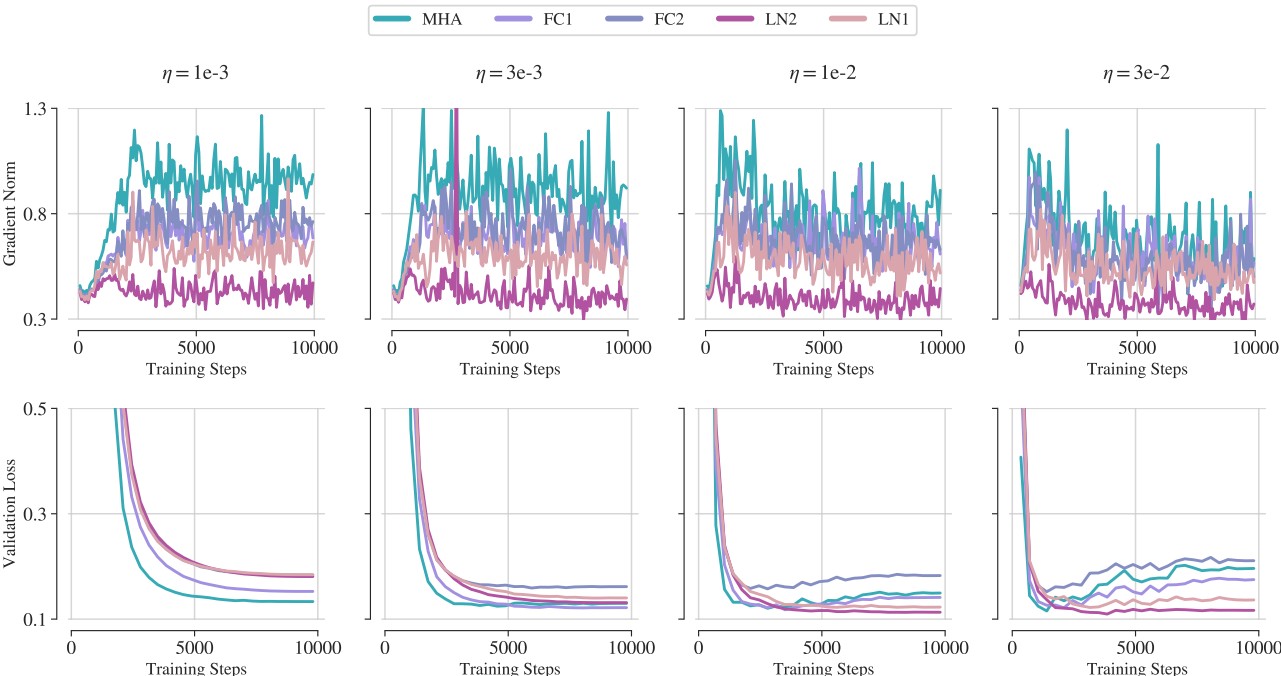

*Figure 37.* **Training dynamics on Snow with seed** 42. Akin to Fig. 36, we observe the same consistent pattern of faster and better convergence for components with high plasticity.

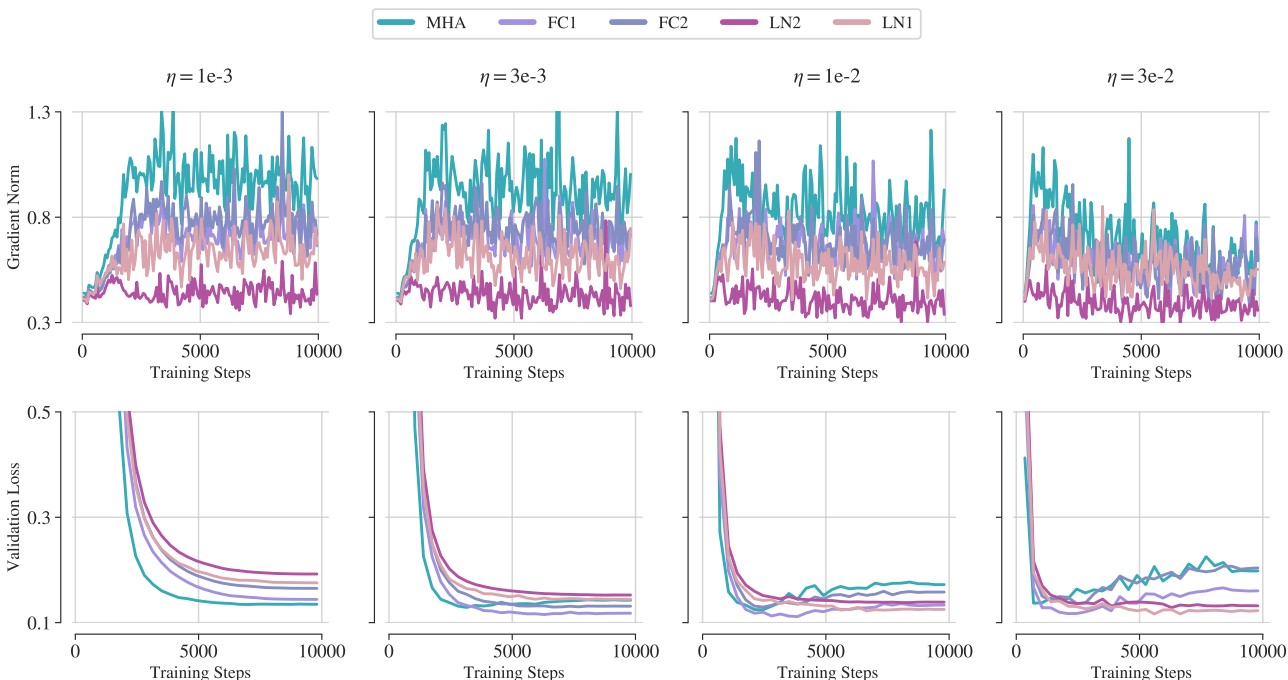

*Figure 38.* **Training dynamics on Snow with seed** 3407. Akin to Fig. 36, we observe the same consistent pattern of faster and better convergence for components with high plasticity.

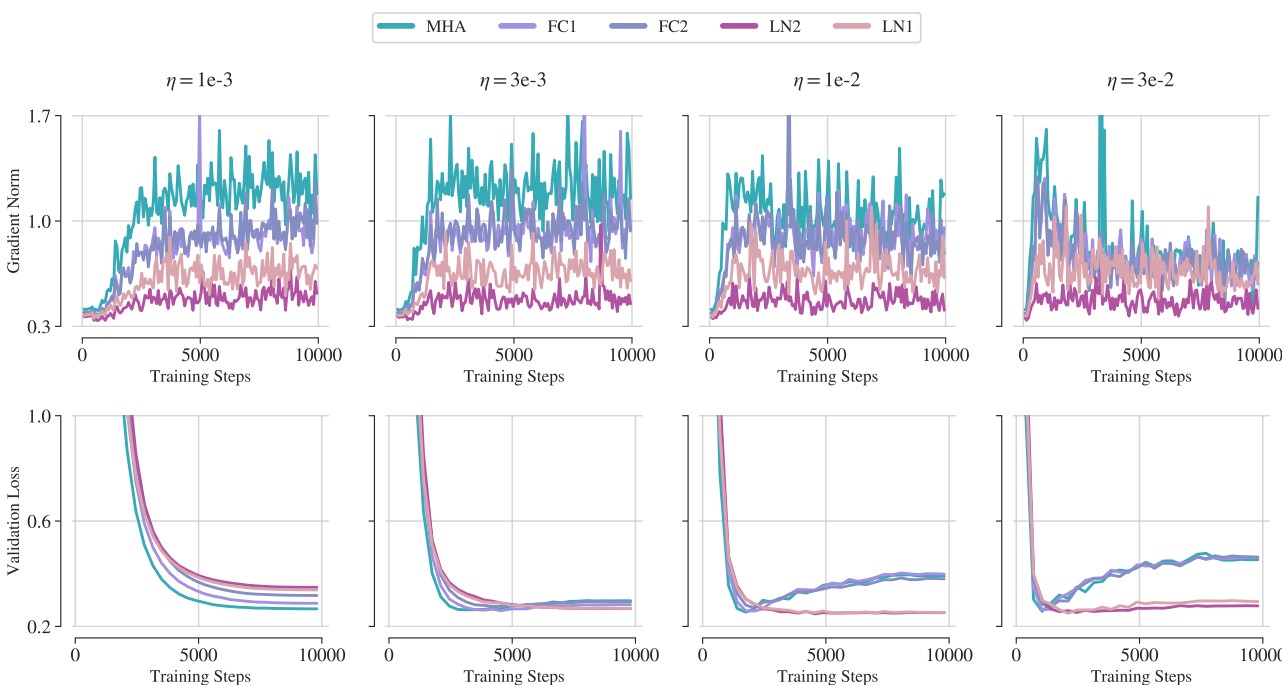

*Figure 39.* **Training dynamics on Speckle Noise with seed** 0. We display the evolution during training of the gradient norms (**top**) and the validation loss (**bottom**) of each finetuning configuration of Table 5, with increasing learning rate $\eta$ from **left to right**. Components are ordered in terms of decreasing plasticity in the legend. Plastic components have higher gradient norms, which leads to a steeper descent in the validation loss and better downstream performance. The benefits of plasticity are even more salient with low learning rates. Overall, higher plasticity leads to better optimization and generalization.

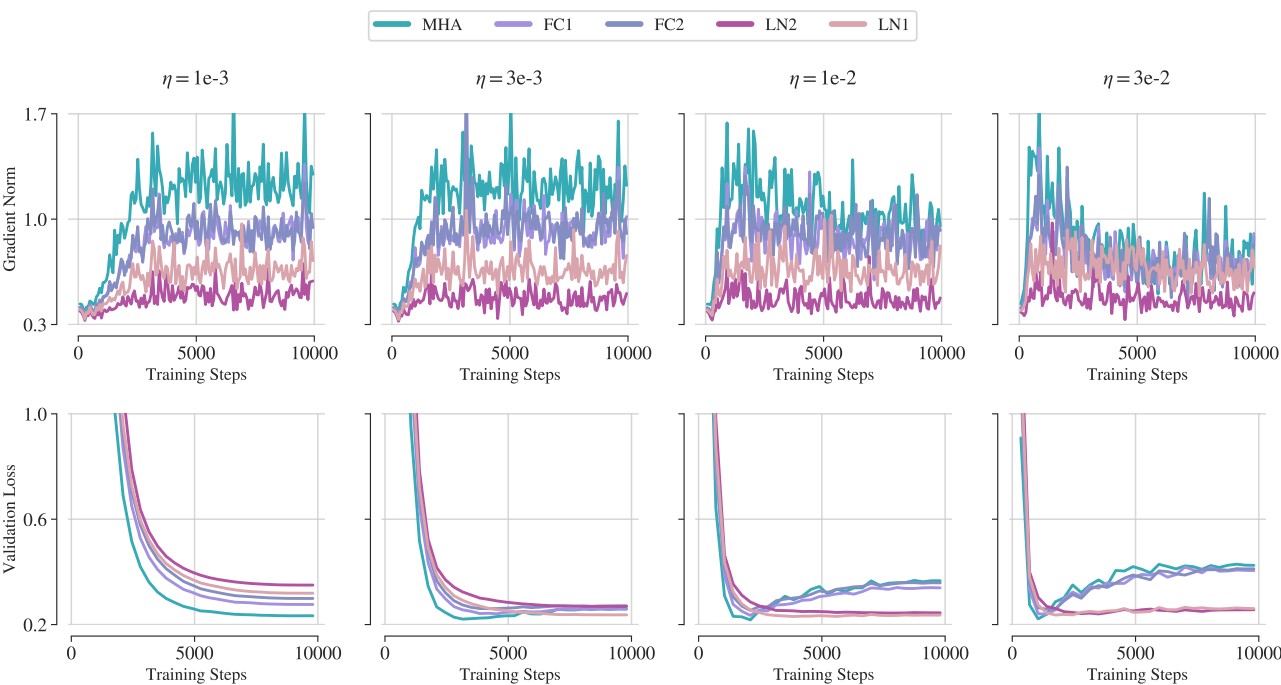

*Figure 40.* **Training dynamics on Speckle Noise with seed** 42. Akin to Fig. 39, we observe the same consistent pattern of faster and better convergence for components with high plasticity.

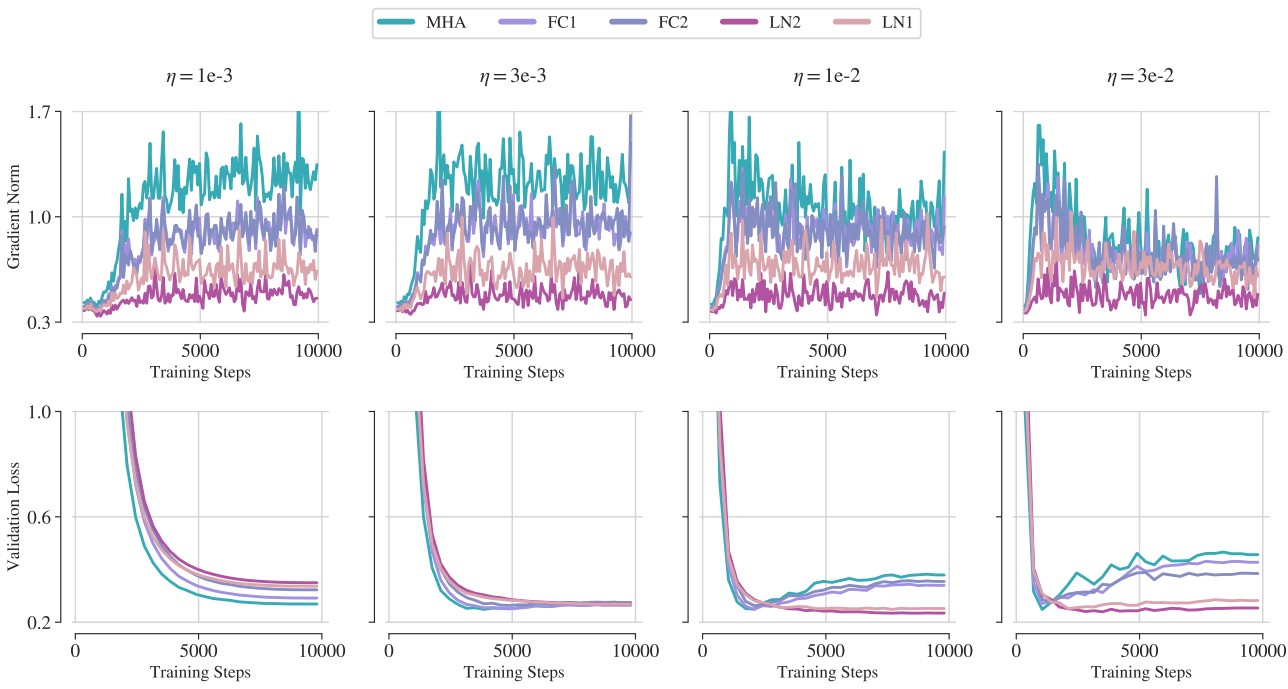

*Figure 41.* **Training dynamics on Speckle Noise with seed** 3407. Akin to Fig. 39, we observe the same consistent pattern of faster and better convergence for components with high plasticity.

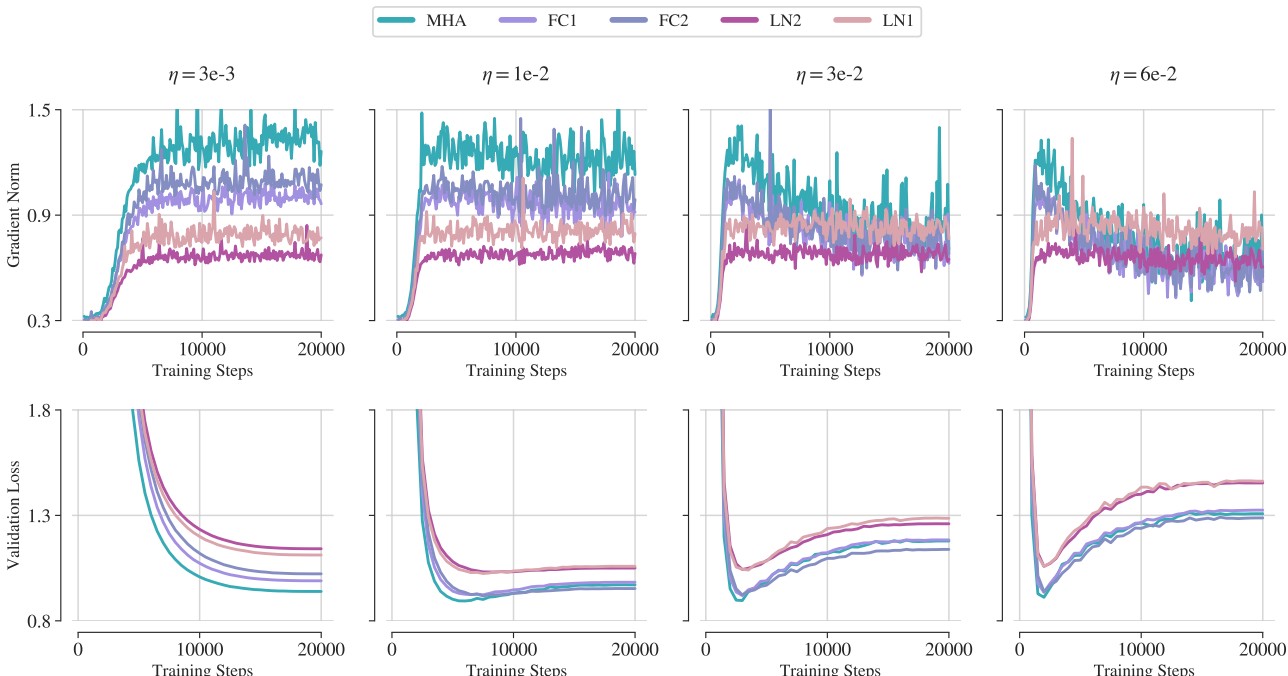

*Figure 42.* **Training dynamics on Clipart with seed** 0. We display the evolution during training of the gradient norms (**top**) and the validation loss (**bottom**) of each finetuning configuration of Table 5, with increasing learning rate $\eta$ from **left to right**. Components are ordered in terms of decreasing plasticity in the legend. Plastic components have higher gradient norms, which leads to a steeper descent in the validation loss and better downstream performance. The benefits of plasticity are salient across all learning rates. Overall, higher plasticity leads to better optimization and generalization.

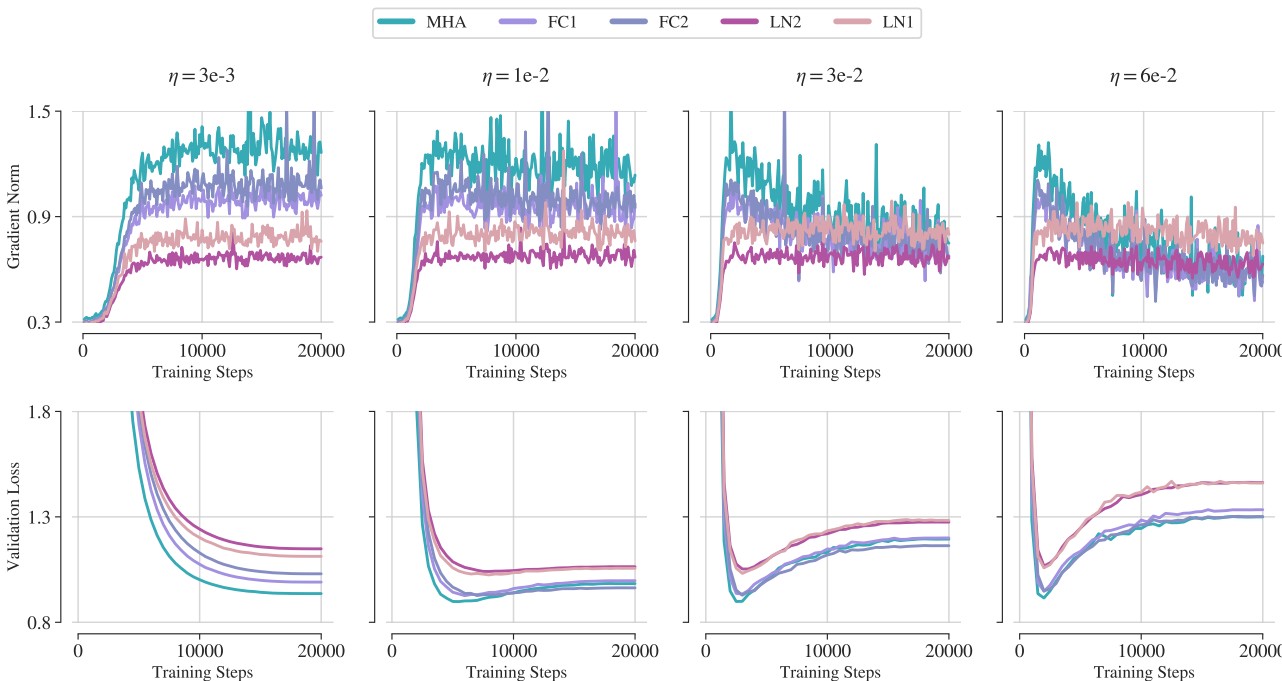

*Figure 43.* **Training dynamics on Clipart with seed** 42. Akin to Fig. 42, we observe the same consistent pattern of faster and better convergence for components with high plasticity.

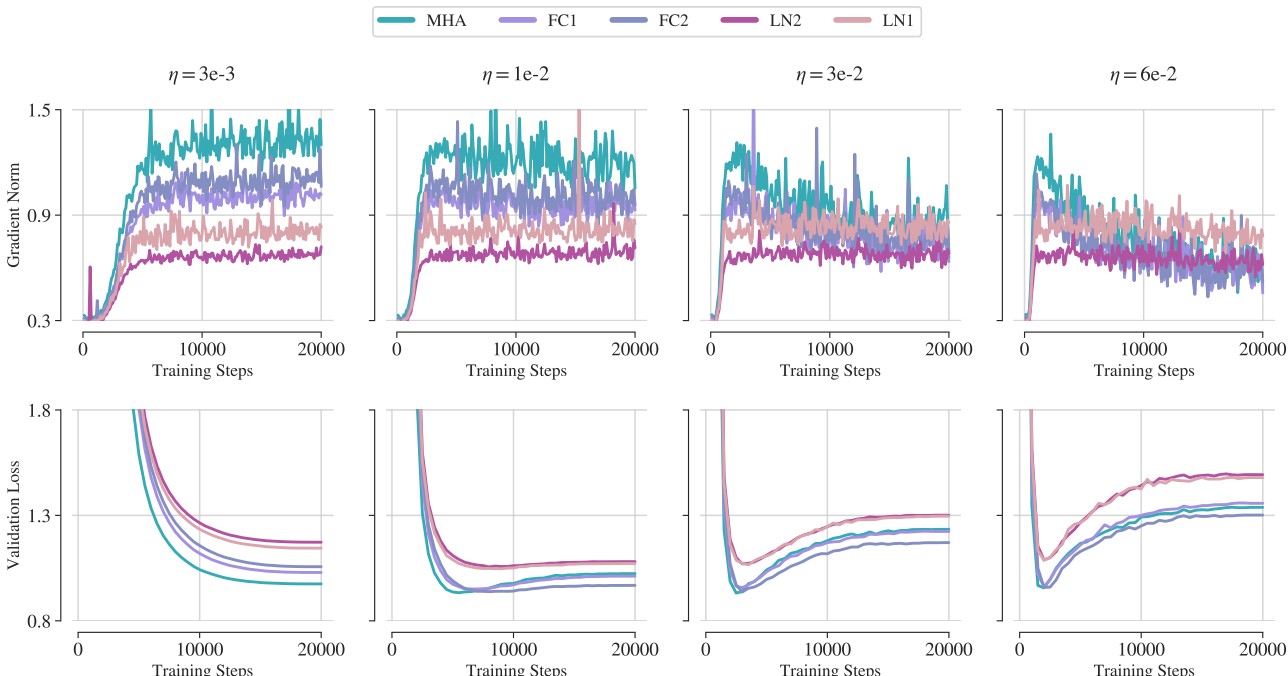

*Figure 44.* **Training dynamics on Clipart with seed** 3407**.** Akin to Fig. 42, we observe the same consistent pattern of faster and better convergence for components with high plasticity.

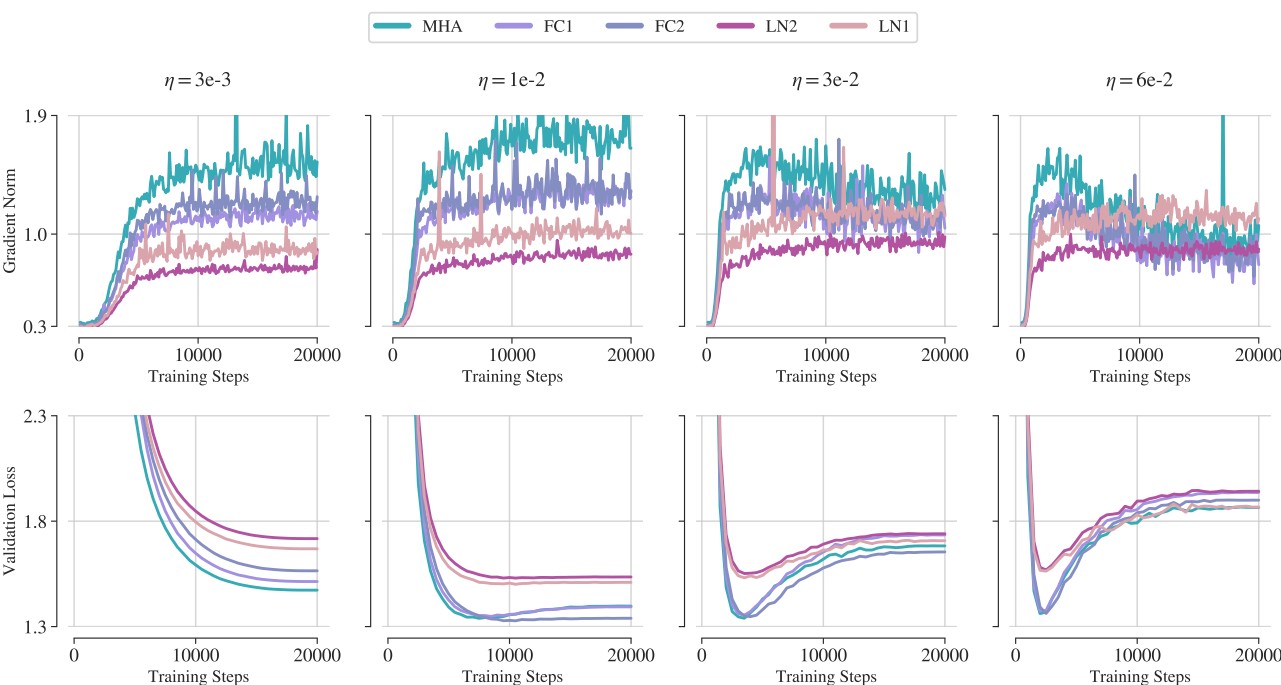

*Figure 45.* **Training dynamics on Sketch with seed** 0**.** We display the evolution during training of the gradient norms (**top**) and the validation loss (**bottom**) of each finetuning configuration of Table 5, with increasing learning rate $\eta$ from **left to right**. Components are ordered in terms of decreasing plasticity in the legend. Plastic components have higher gradient norms, which leads to a steeper descent in the validation loss and better downstream performance. The benefits of plasticity are salient across all learning rates. Overall, higher plasticity leads to better optimization and generalization.

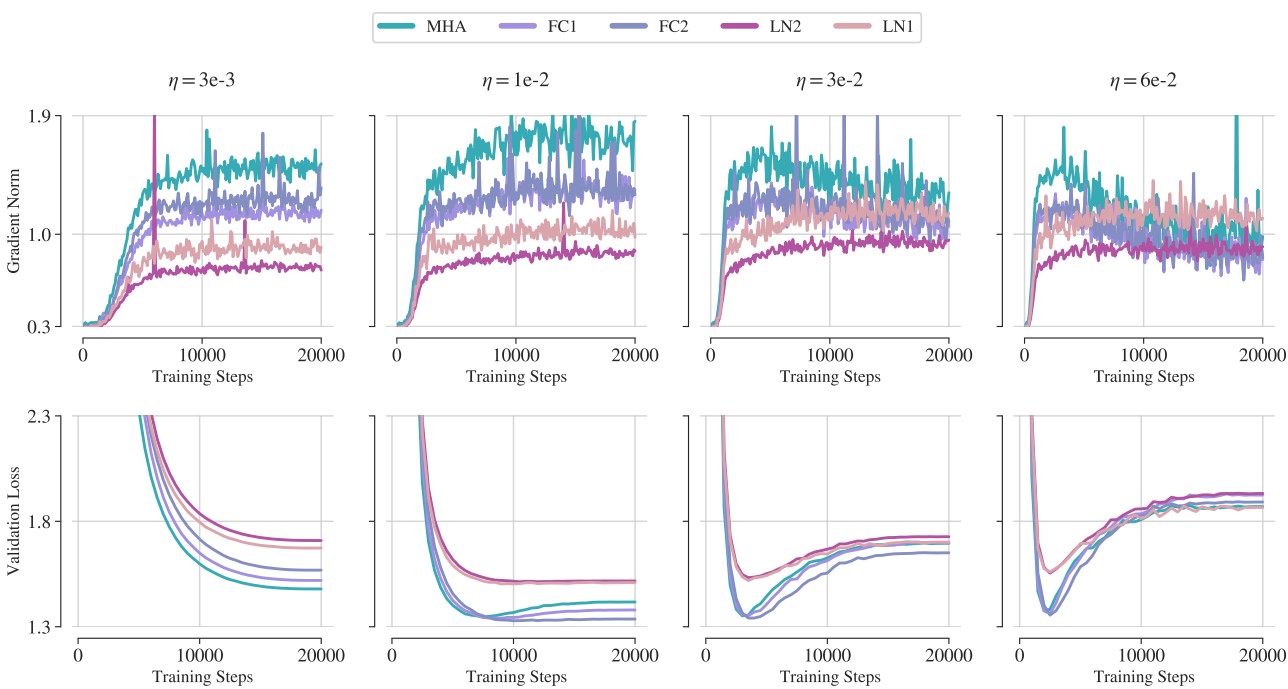

*Figure 46.* **Training dynamics on Sketch with seed** 42**.** Akin to Fig. 45, we observe the same consistent pattern of faster and better convergence for components with high plasticity.

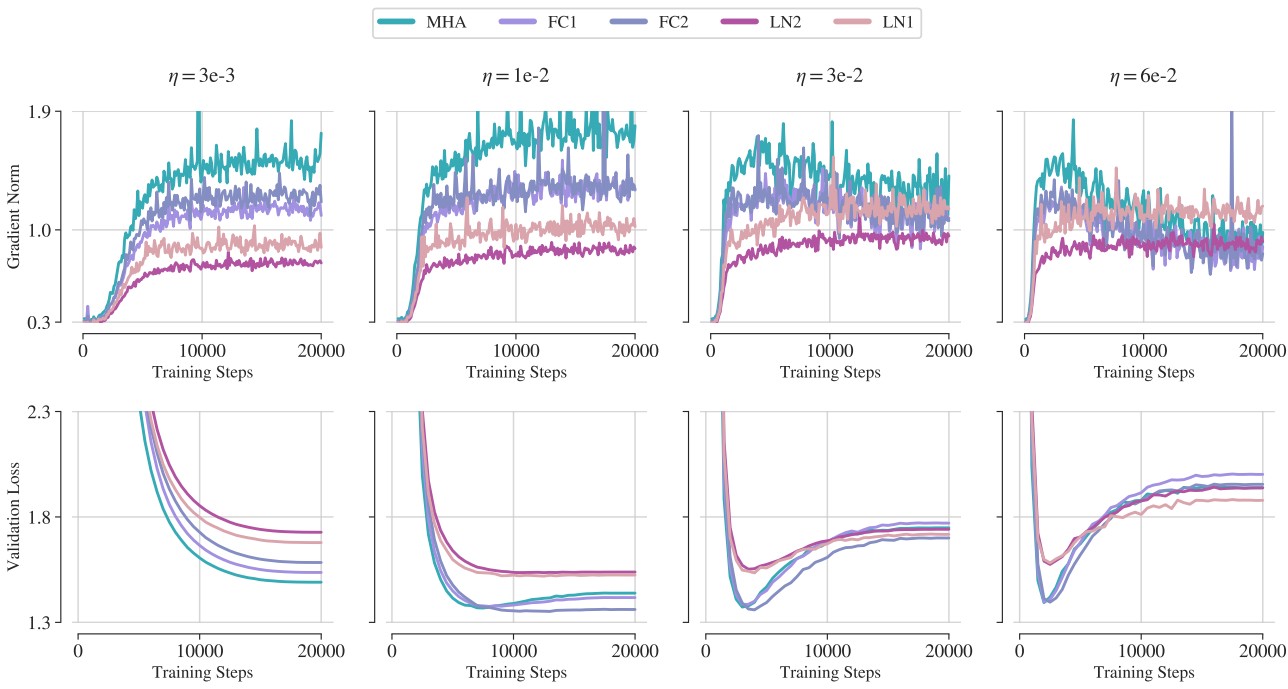

*Figure 47.* **Training dynamics on Sketch with seed** 3407**.** Akin to Fig. 45, we observe the same consistent pattern of faster and better convergence for components with high plasticity.

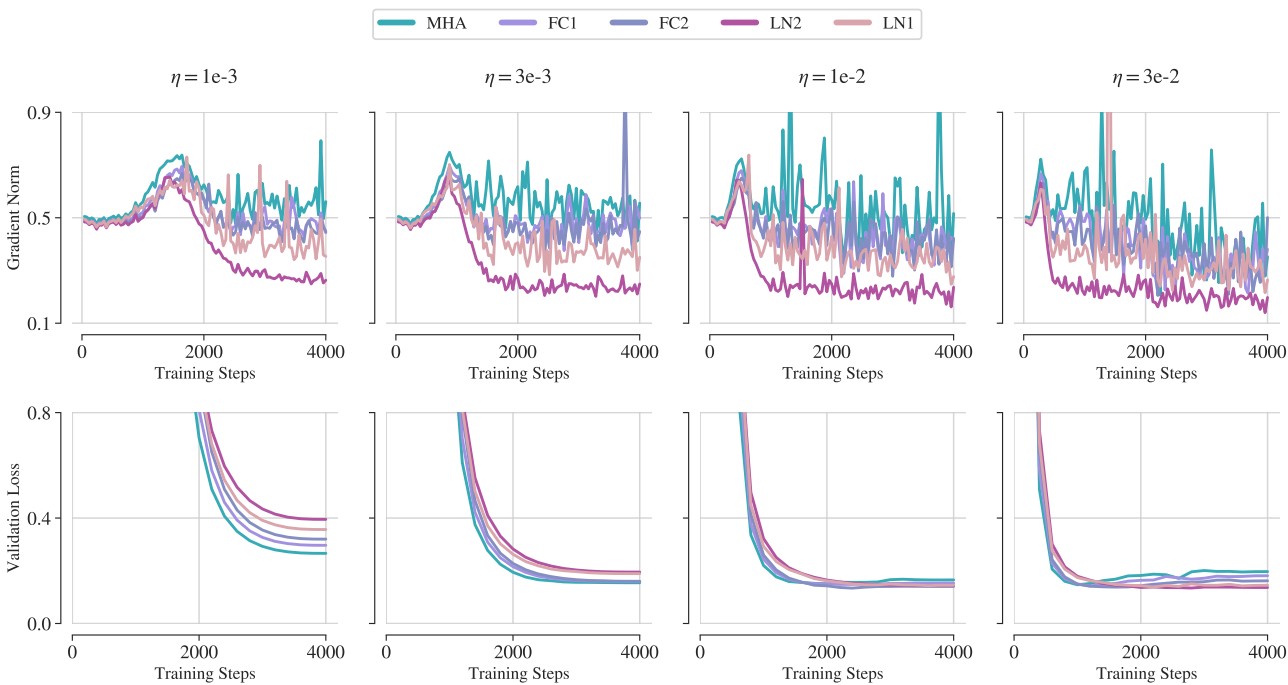

*Figure 48.* **Training dynamics on Pet with seed** 0. We display the evolution during training of the gradient norms (**top**) and the validation loss (**bottom**) of each finetuning configuration of Table 5, with increasing learning rate $\eta$ from **left to right**. Components are ordered in terms of decreasing plasticity in the legend. On Pet, the pretrained model already achieves good linear probing performance as shown in Table 8. This leads to lower gradient norms for all components compared to more challenging datasets. That being said, we observe for low learning rate that plastic components have higher gradient norms ans a steeper descent in the validation loss.

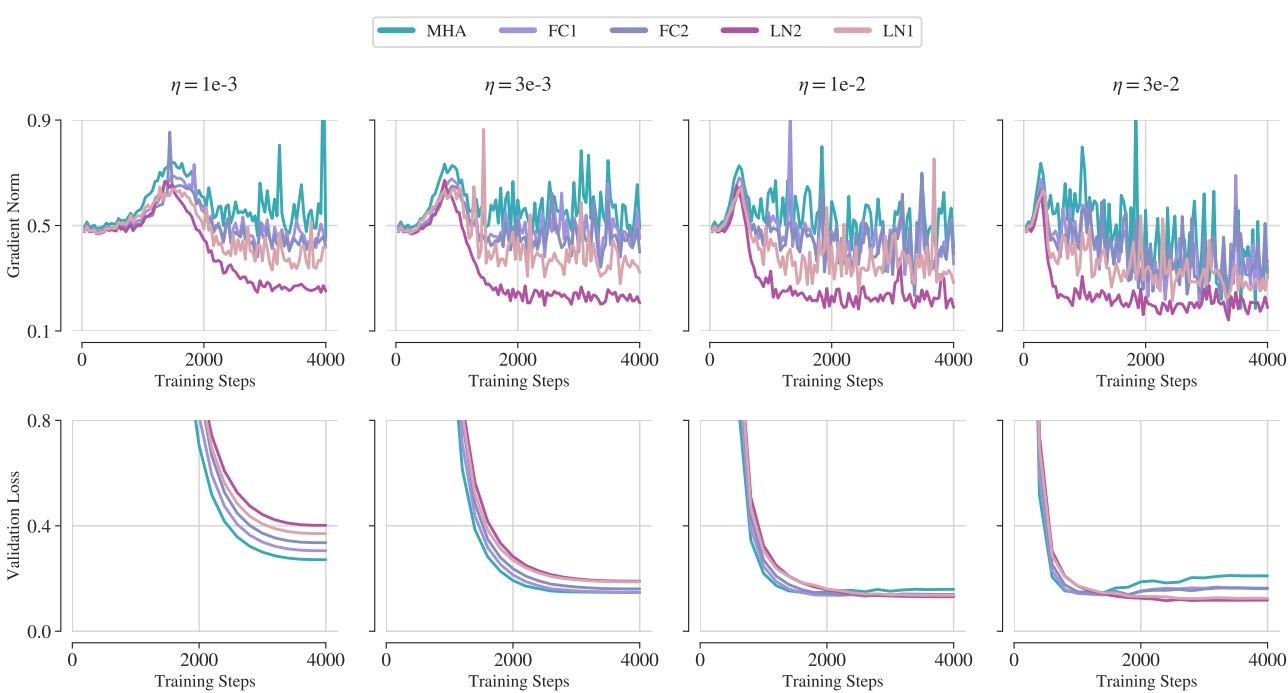

*Figure 49.* **Training dynamics on Pet with seed** 42. Akin to Fig. 48, we observe the same consistent patterns.

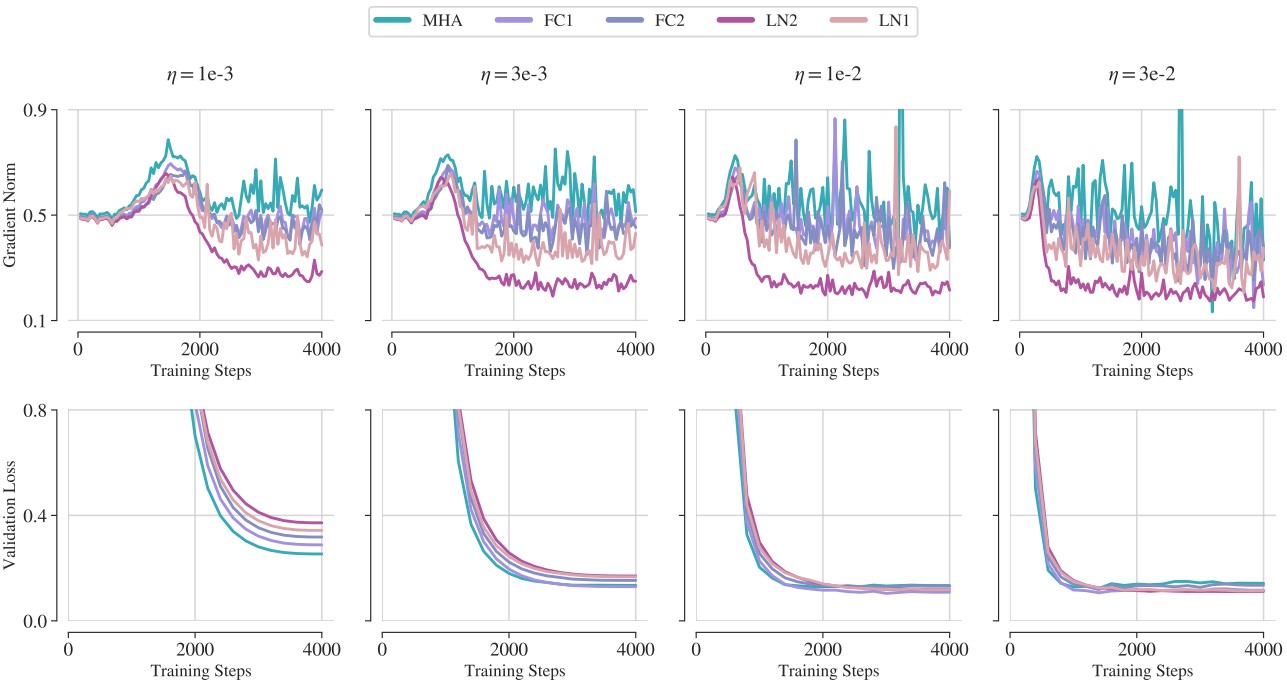

*Figure 50.* **Training dynamics on Pet with seed** 3407. Akin to Fig. 48, we observe the same consistent patterns.

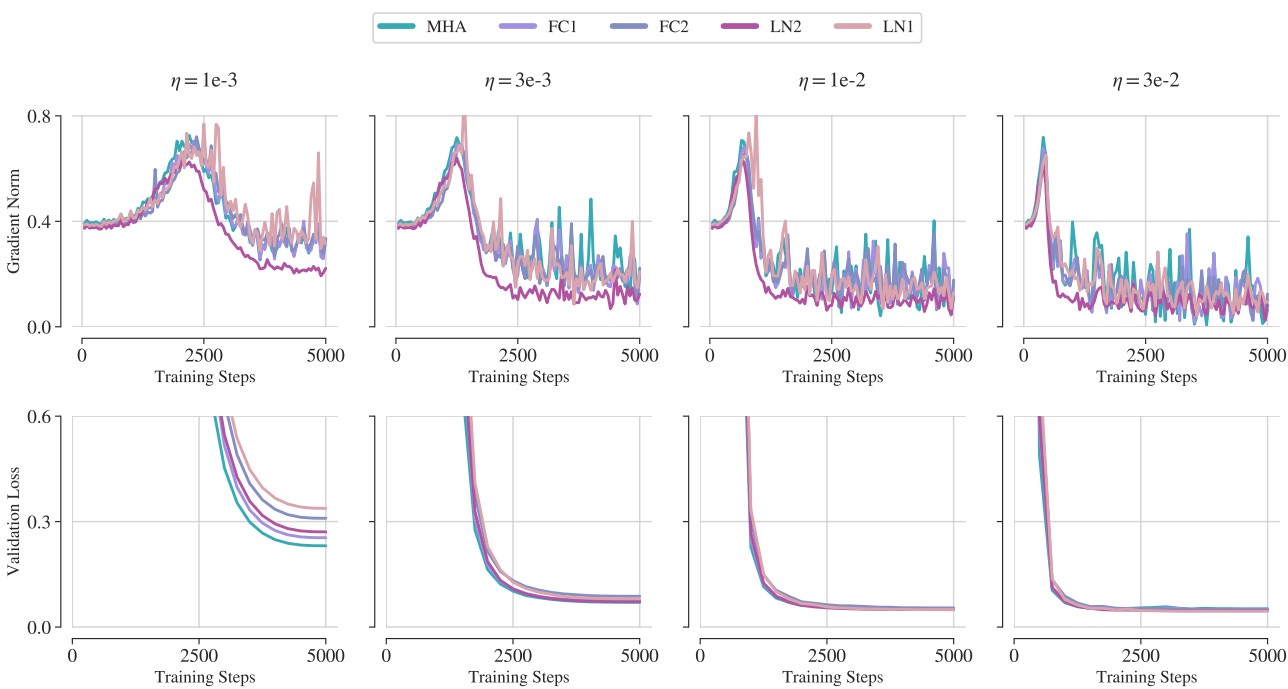

*Figure 51.* **Training dynamics on Flowers102 with seed** 0. We display the evolution during training of the gradient norms (**top**) and the validation loss (**bottom**) of each finetuning configuration of Table 5, with increasing learning rate $\eta$ from **left to right**. Components are ordered in terms of decreasing plasticity in the legend. On Flowers102, the pretrained model already achieves good linear probing performance as shown in Table 8. This leads to lower gradient norms for all components compared to more challenging datasets. That being said, we observe for low learning rate that plastic components have higher gradient norms ans a steeper descent in the validation loss.

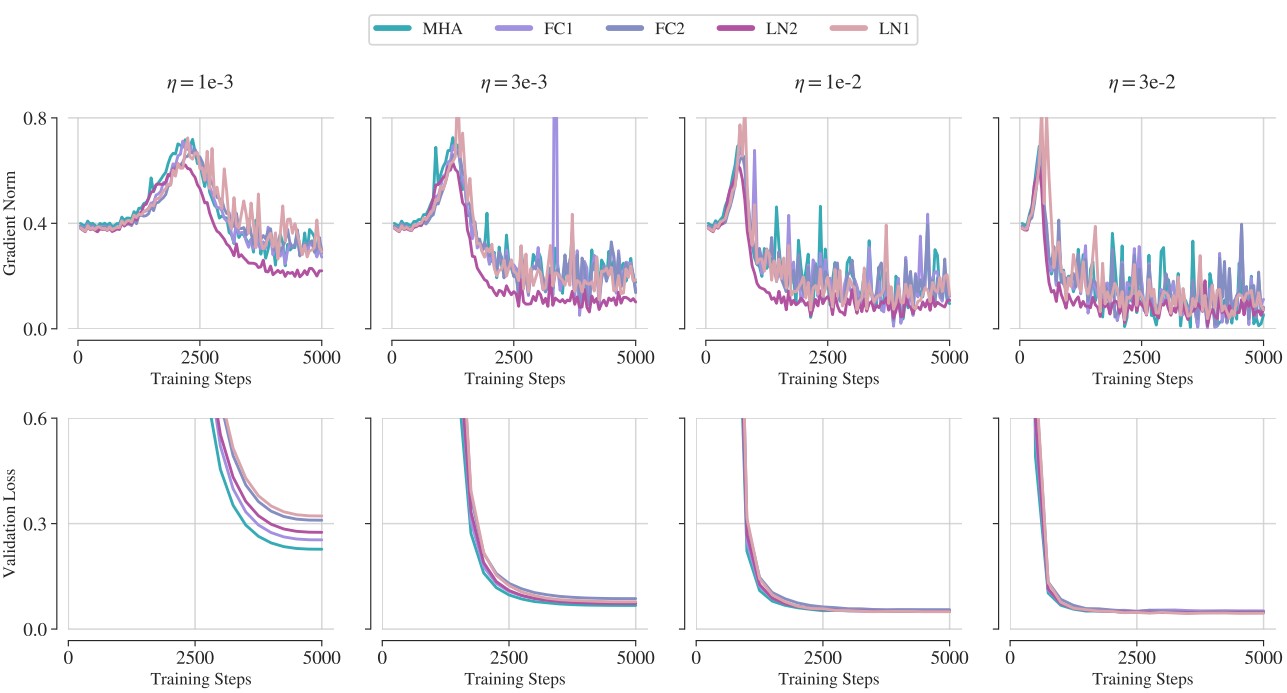

*Figure 52.* **Training dynamics on Flowers102 with seed** 42. Akin to Fig. 51, we observe the same consistent patterns.

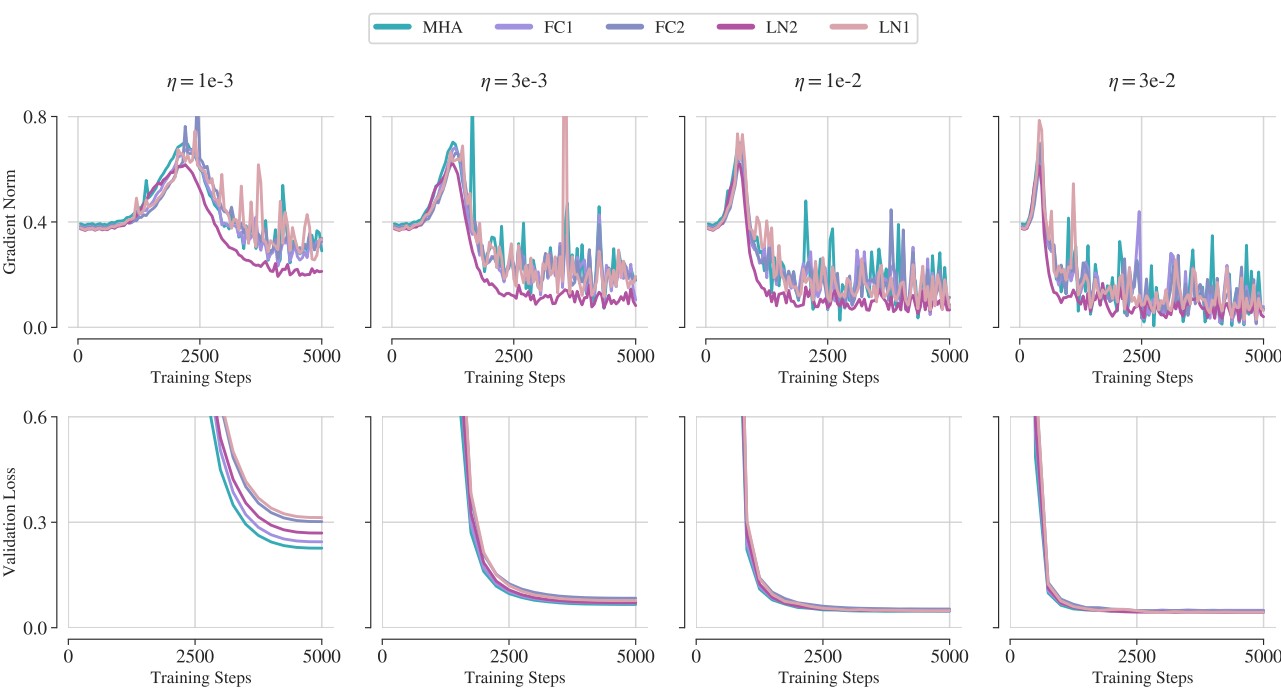

*Figure 53.* **Training dynamics on Flowers102 with seed** 42. Akin to Fig. 51, we observe the same consistent patterns.

### E.2.5. EXTENSION TO ADAM

In our work, we use the SGD optimizer (Bottou, 2010) following the original ViT paper (Dosovitskiy et al., 2021). The Adam optimizer (Kingma & Ba, 2014), more precisely its decoupled weight-decay version from Loshchilov & Hutter (2019), is also a prevalent choice, notably to pretrain large language models (Orvieto & Gower, 2026). As such, it is natural to wonder whether our component-wise analysis holds with Adam. We first note that full-finetuning leads to similar performance for both Adam and SGD (Touvron et al., 2021). We keep Adam's default parameters $\beta_1 = 0.9, \beta_2 = 0.999$ and, for a fair comparison, we choose smaller learning rates as is standard in the literature on adaptive methods (Dosovitskiy et al., 2021; Kumar et al., 2023; Touvron et al., 2021). More precisely, we follow Kumar et al. (2023) and scale the values with the formula $\mathrm{lr_{adam}} = 1e{-}2 \times \mathrm{lr_{sgd}}$, where the SGD learning rates of $\mathrm{lr_{sgd}}$ are given in Table 7. The full results comparison between Adam and SGD is displayed in Table 10. We observe that finetuning non-smooth components (in gray shade) consistently yields higher accuracy and superior stability across learning rates compared to smoother modules. This trend is even more salient with Adam. This showcases that the core thesis that non-smoothness is beneficial to finetuning generalizes across optimizers. Finally, the robust adaptation and interplay between plasticity and optimization are also similar between Adam and SGD as shown in **??**. The evolution of the gradient norms (middle) and the validation loss (right) is given for the finetuning run on Sketch that achieves the highest accuracy (this corresponds to learning rate $\eta = 1e{-}2$). This confirms our intuition: the ordering of gradient norms is aligned with the *plasticity ranking* established in Section 5.1 and the loss descent is steeper for components with high plasticity, such as the attention modules and the feedforward layers. These patterns are consistent with those of SGD displayed in Fig. 5 and hold across learning rates as shown in Figs. 55 to 58.

*Table 10.* **Adam vs SGD**. Average top-1 finetuning accuracy on the test set with standard deviation across learning rates. Transformer components are ordered in terms of decreasing plasticity. The best performance among components for each optimizer is in **bold** and non-smooth components are highlighted in gray. The takeaways are similar for SGD and Adam: non-smooth components lead to better and more stable finetuning performance. We note that, although the accuracy is slightly lower with Adam, the stability of non-smooth components is more salient.

| configuration | MHA | | FC1 | | FC2 | | LN2 | | LN1 | |
|---|---|---|---|---|---|---|---|---|---|---|
| | Adam | SGD | Adam | SGD | Adam | SGD | Adam | SGD | Adam | SGD |
| Cifar100 | 91.0 ±0.2 | **91.7** ±1.1 | **91.3** ±0.6 | 91.6 ±1.6 | 90.6 ±1.4 | 91.2 ±0.7 | 89.4 ±2.7 | 90.6 ±1.4 | 88.4 ±3.2 | 89.9 ±1.7 |
| Motion Blur | 92.4 ±0.2 | 94.1 ±0.2 | 91.8 ±0.4 | **94.4** ±0.2 | 90.7 ±0.9 | 93.7 ±0.3 | 91.7 ±2.4 | 93.7 ±0.6 | 90.7 ±2.4 | 92.6 ±0.4 |
| Clipart | 76.4 ±0.5 | **76.9** ±0.6 | 74.8 ±0.8 | 75.7 ±0.6 | 75.8 ±0.3 | 75.9 ±0.8 | 73.3 ±1.9 | 73.6 ±0.8 | 73.5 ±1.5 | 74.0 ±0.5 |
| Sketch | 69.0 ±0.5 | 68.5 ±1.1 | 67.9 ±0.6 | 68.1 ±1.6 | **69.2** ±0.4 | 68.1 ±2.0 | 63.1 ±2.9 | 63.9 ±1.6 | 64.0 ±2.6 | 64.5 ±1.0 |
| Average | **82.2** | **82.8** | 81.4 | 82.5 | 81.7 | 82.1 | 79.4 | 80.5 | 79.2 | 80.2 |

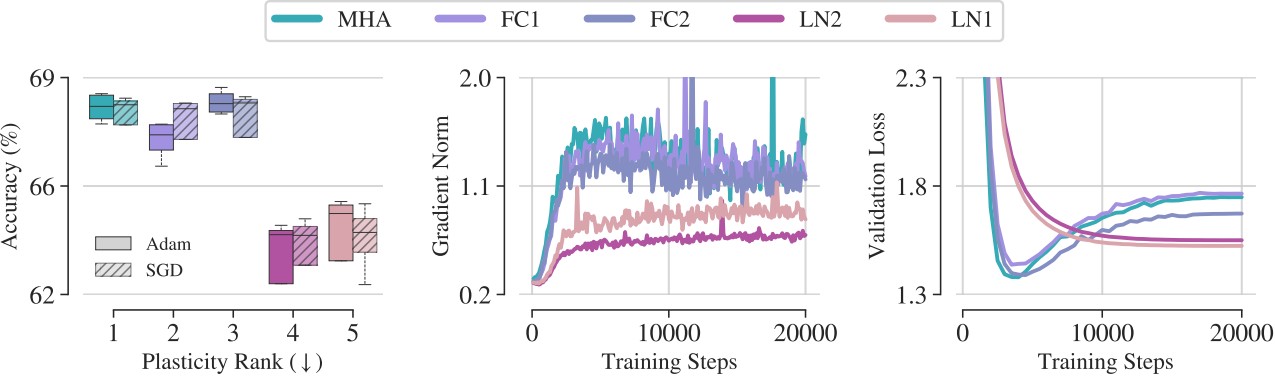

*Figure 54.* **Benefits of plasticity with Adam on Sketch.** Transformer components are ordered in terms of decreasing plasticity. We can see that the performance across learning rates (**left**) is better and more stable for plastic components, with similar trends for both Adam and SGD. The evolution of Adam's gradient norms (**middle**) and validation loss (**right**) throughout training is consistent with those of SGD shown in Fig. 5: the higher plasticity, the larger gradient norms, and the better the generalization.

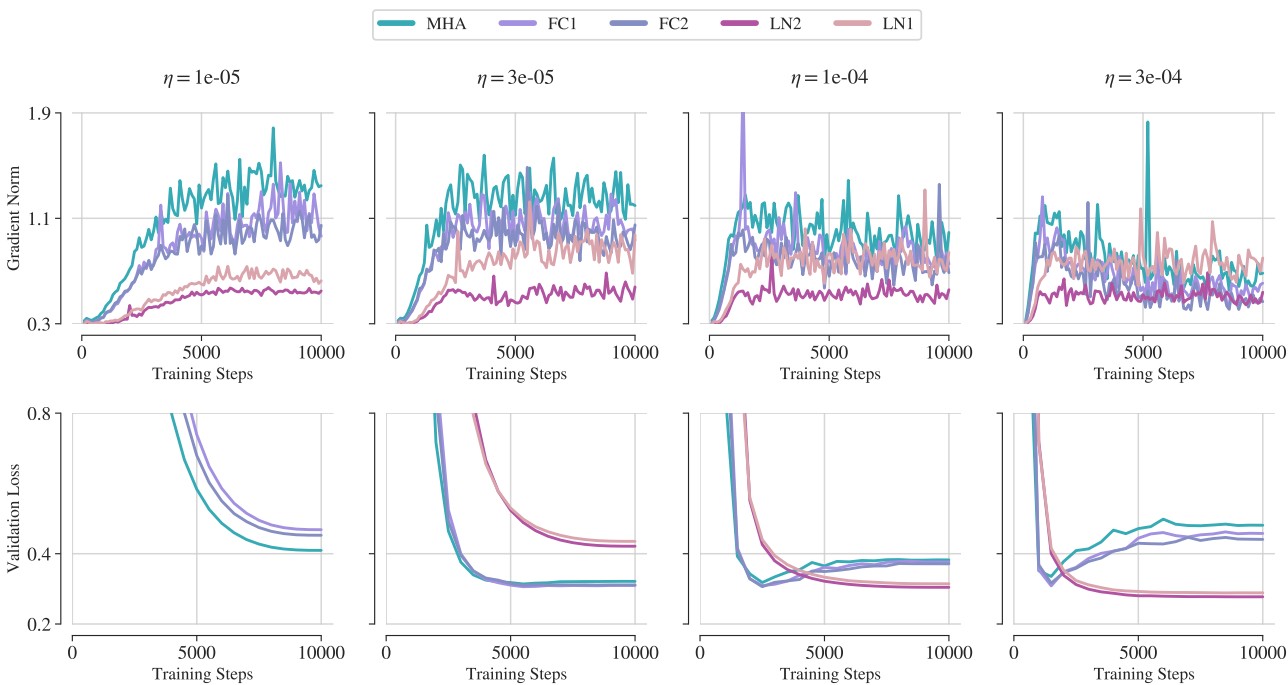

*Figure 55.* **Training dynamics on Cifar100 with Adam and seed** 0. We display the evolution during training of the gradient norms (**top**) and the validation loss (**bottom**) of each finetuning configuration of Table 5, with increasing learning rate $\eta$ from **left to right**. Components are ordered in terms of decreasing plasticity in the legend. Plastic components have higher gradient norms, which leads to a steeper descent in the validation loss and better downstream performance. The benefits of plasticity are even more salient with low learning rates. Overall, higher plasticity leads to better optimization and generalization.

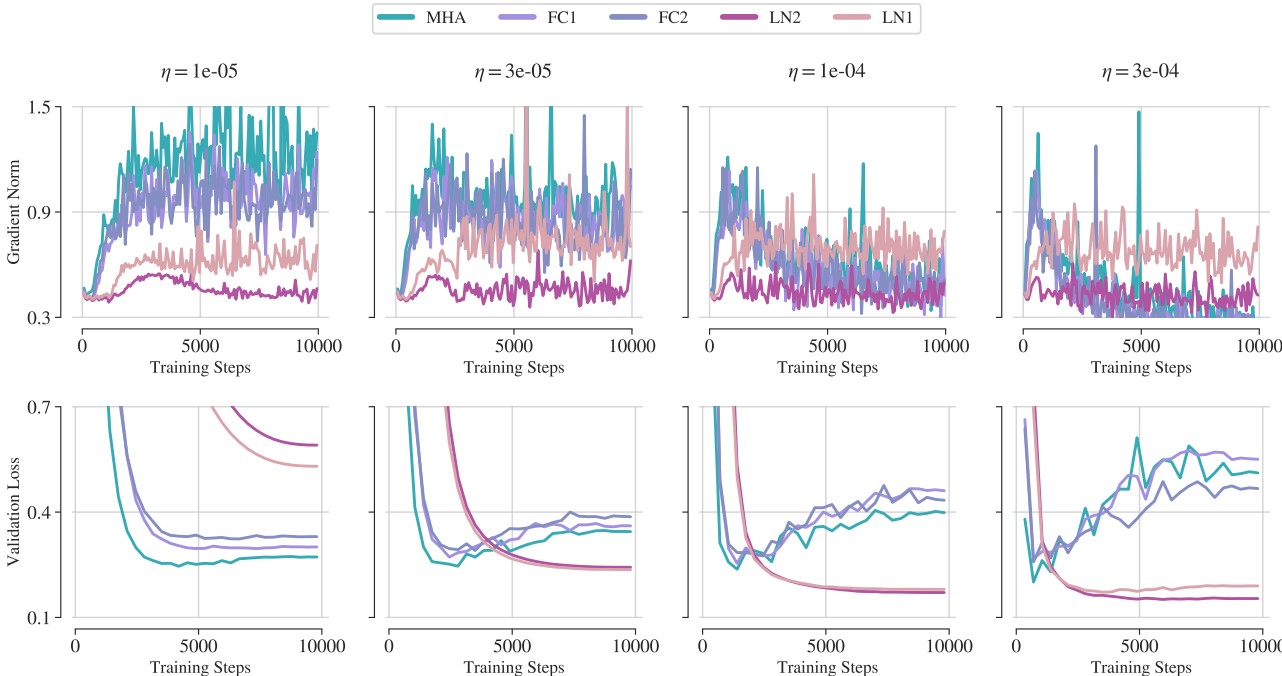

*Figure 56.* **Training dynamics on Motion Blur with Adam and seed** 0. We display the evolution during training of the gradient norms (**top**) and the validation loss (**bottom**) of each finetuning configuration of Table 5, with increasing learning rate $\eta$ from **left to right**. Components are ordered in terms of decreasing plasticity in the legend. Plastic components have higher gradient norms, which leads to a steeper descent in the validation loss and better downstream performance. The benefits of plasticity are even more salient with low learning rates. Overall, higher plasticity leads to better optimization and generalization.

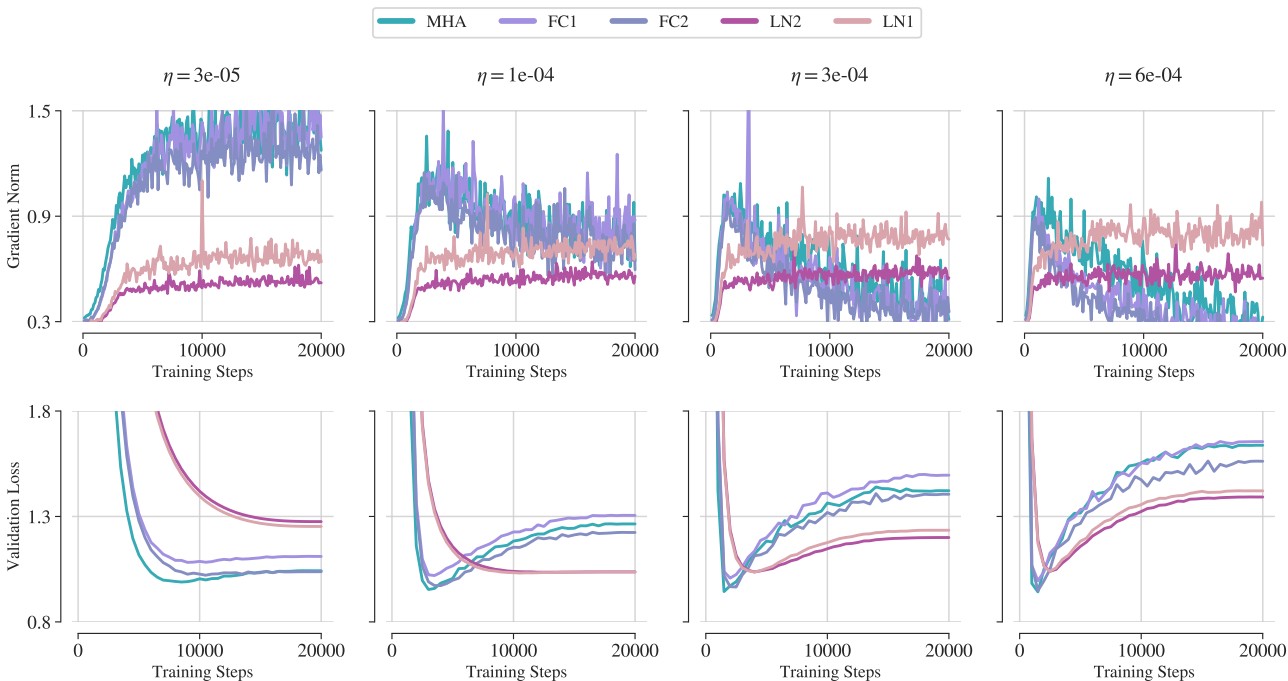

*Figure 57.* **Training dynamics on Clipart with Adam and seed** 0. We display the evolution during training of the gradient norms (**top**) and the validation loss (**bottom**) of each finetuning configuration of Table 5, with increasing learning rate $\eta$ from **left to right**. Components are ordered in terms of decreasing plasticity in the legend. Plastic components have higher gradient norms, which leads to a steeper descent in the validation loss and better downstream performance. The benefits of plasticity are even more salient with low learning rates. Overall, higher plasticity leads to better optimization and generalization.

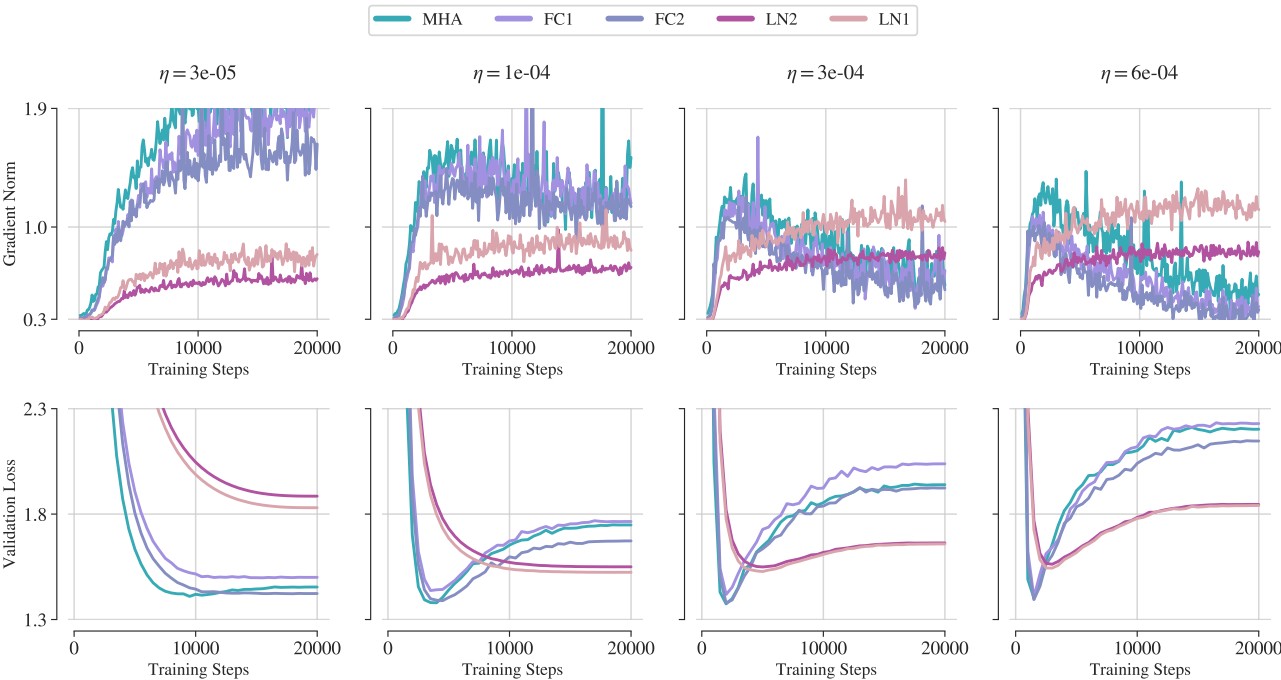

*Figure 58.* **Training dynamics on Sketch with Adam and seed** 0. We display the evolution during training of the gradient norms (**top**) and the validation loss (**bottom**) of each finetuning configuration of Table 5, with increasing learning rate $\eta$ from **left to right**. Components are ordered in terms of decreasing plasticity in the legend. Plastic components have higher gradient norms, which leads to a steeper descent in the validation loss and better downstream performance. The benefits of plasticity are even more salient with low learning rates. Overall, higher plasticity leads to better optimization and generalization.

