# OpenReview forum: "Vision Transformer Finetuning Benefits from Non-Smooth Components"
_ICML.cc/2026/Conference — ICML 2026 regular_

### Official Review · Reviewer_92jq · 2026-03-04

**Soundness:** 2
**Presentation:** 3
**Significance:** 2
**Originality:** 3
**Overall Recommendation:** 4
**Confidence:** 4

**Summary:**

This paper investigates which components of a Transformer should be prioritized during fine-tuning by analyzing their plasticity, defined as the average sensitivity of a module’s output to changes in its input. The key intuition is that modules with higher plasticity produce larger gradient norms during optimization, allowing them to adapt more quickly and effectively to downstream tasks. The authors derive theoretical bounds on the plasticity of different Transformer components and validate these predictions empirically using pretrained Vision Transformers (ViTs) on multiple image classification benchmarks. Both theory and experiments reveal the same ordering of module plasticity: MHA$\rightarrow$FC1$\rightarrow$FC2$\rightarrow$LN2$\rightarrow$LN1. Fine-tuning experiments further show that prioritizing high-plasticity modules, particularly MHA and FC1, yields higher and more stable downstream performance across diverse datasets, learning rates, and random seeds.

**Compliance With Llm Reviewing Policy:**

Affirmed.

**Final Justification:**

The authors have addressed my concerns. This work provides theoretical insight into the prioritization of components in transformer architectures, and the proposed ordering generally holds in both full- and low-data settings (though not consistently). Overall, I view the paper positively.

**Key Questions For Authors:**

Please see the weaknesses part.

**Limitations:**

yes

**Strengths And Weaknesses:**

**Strengths:**
1. The paper supports its claims with both theoretical analysis and empirical validation.

2. The authors conduct experiments across diverse settings to verify the proposed idea.

3. The paper is well written and easy to follow.

**Weaknesses:**
1. The proposed plasticity metric differs only marginally from the Lipschitz constant, as both are derived from the same ratio, with plasticity taking the expectation and the Lipschitz constant the supremum. It is unclear whether the Lipschitz constant would yield the same component ranking; if so, the motivation for introducing plasticity as a new measure is weak. A direct empirical comparison between the two would be straightforward but is missing.

2.  All experiments are conducted on full datasets, while real-world fine-tuning of foundation models often occurs in low-data regimes. High-plasticity modules, which introduce large gradient norms may overfit or cause catastrophic forgetting when trained on limited samples [1, 2, 3], whereas low-plasticity components like LayerNorm could better preserve pretrained representations. It would be more comprehensive to include low-data experiments to verify if prioritizing high-plasticity modules still holds on low-data settings.

3. Experiments are restricted to supervised ViTs pretrained on ImageNet-21k. It remains unclear whether the plasticity ranking holds for other architectures (e.g., Swin Transformer [4] ) or self-supervised pretraining methods (MAE [5] or Dino [6]).

4. In table 3, attention and feedforward layers have 28M parameters each, while LayerNorm has only 18K. This large gap makes it unclear whether the performance gains come from plasticity or simply more trainable parameters. A fair comparison should control for parameter count.


[1] Béthune et al, Scaling Laws for Forgetting during Finetuning with Pretraining Data Injection, ICML 2025.

[2] Ni et al, PACE: Marrying generalization in PArameter-efficient fine-tuning with Consistency rEgularization, NeurIPS 2024.

[3] Bafghi et al, Parameter Efficient Fine-tuning of Self-supervised ViTs without Catastrophic Forgetting,  CVPR Workshop 2024.

[4] Liu et al, Swin Transformer: Hierarchical Vision Transformer using Shifted Windows, ICCV 2021.

[5] He et al, Masked Autoencoders Are Scalable Vision Learners, CVPR 2022.

[6] Caron et al, Emerging Properties in Self-Supervised Vision Transformers, ICCV 2021.

---

> ### Author Rebuttal · Authors · 2026-03-31
>
> We thank the reviewer 92jq for their constructive feedback. We appreciate that the reviewer found our paper's claims well supported theoretically and experimentally across diverse settings. We address the reviewer's concerns point by point below.
>
> > **(1) The proposed plasticity metric differs only marginally from the Lipschitz [...]**
>
> Our plasticity measure must capture the typical response of a given component over the distribution of input sequences. The expectation is a natural choice to aggregate ratios as it provides an informative and representative summary, robust to outliers. On the contrary, Lipschitz constant can be pessimistic since a single outlier can push its value arbitrarily high. This issue is known in the literature: [1] showed that Lipschitz constant can be dramatically larger than the typical behavior, choosing instead the average smoothness to derive sharper generalization bounds. During the elaboration of the method and theoretical analysis (Sections 3 and 4), we also found the supremum to be too sensitive to extreme fluctuations. For instance, the derivative of LayerNorm is $\left|\frac{\partial \text{LN}(x)}{\partial x_i}\right| \propto \frac{\gamma}{\sigma(x)}$ and the Lipschitz constant is $\hat{L}(\text{LN}) = \sup_x \left|\frac{\partial \text{LN}(x)}{\partial x_i}\right|
> \propto \frac{\gamma}{\inf_x \sigma(x)}$. This explodes when $\sigma(x) \to 0$ for a token with low variance in $\mathbb{R}^d$ (e.g., a patch representing a uniform part of an image like some background). LayerNorm is by design robust to input changes (it limits the propagation of perturbations by rescaling features). Using Lipschitz constant as a plasticity measure could lead to arbitrarily large values regardless of how the component behaves on all other patches. This makes Lipschitz constant unreliable to rank transformer components by adaptability to input changes. That being said, on clean data with stable ratios, taking the expectation or the supremum could lead to the same ranking. Does this clarify the reviewer's concerns regarding the motivation behind using the expectation?
>
> > **(2) All experiments are conducted on full datasets, while real-world fine-tuning [...]**
>
> We follow the finetuning setup of the original ViT paper with training sizes from 1K to 50K, and as such, it includes low-data regimes. We thank the reviewer for the valuable references [2, 3], which consider the same datasets as us with similar training sizes (e.g., Pet, Flowers102, Cifar10, Cifar100). Our findings indicate that the benefits of high plasticity modules for finetuning hold both in low-data and medium-data regimes. Following the reviewer's suggestion, we will add a more detailed discussion on low-data regimes in the revised version.
>
> > **(3) Experiments are restricted to supervised ViTs pretrained on ImageNet-21k. It [...].**
>
> We thank the reviewer for the valuable suggestions. We add plasticity analysis on DINOv3 (7B) and GPT2 (using ImageNet and Cifar100 for DINOv3; WikiText and AG News for GPT2). As shown in the table below, the plasticity ranking (MHA > FC > LN) remains consistent across **architectures** (encoder vs. decoder), **pretraining objectives** (supervised, discriminative SSL, and next-token prediction), and **modalities** (vision and NLP). New figure consistent with Figure 3 at this anonymous link: https://osf.io/y3fhg/files/x3w25?view_only=452822d7e7ca4ab7ac03ab813cfa4cda. The table below displays the average plasticity of each transformer component for ViT, DINOv3, and GPT2, with the highest plasticity in **bold**.
>
> | Plasticity (architecture) | MHA | FC1 | FC2 | LN2 | LN1 |
> | :--- | :--- | :--- | :--- | :--- | :--- |
> | **ViT-Base** | **7.9** | 5.4 | 2.3 | 0.6 | 0.3 |
> | **DINOv3 (7B)** | **13.5** | 5.00 | 3.2 | 0.6 | 0.5 |
> | **GPT2** | **27.8** | 8.1 | 3.3 | 2.1 | 1.9 |
>
> We will add these new results to the revised version.
>
> > **(4) In table 3, attention and feedforward layers have 28M parameters each, while [...]**
>
> **Due to space constraints, see answer (4) to Reviewer TEqP (sorry for the inconvenience)**.
>
> Given the significant computational cost (**~800 finetuning runs** for a single optimizer across all modules and datasets), we focused on a rigorous analysis of a widely used setting (ViT-Base has **4M monthly downloads on HuggingFace**). The consistency of our findings across additional runs (spanning optimizers, architecture, and modality) confirms the generalizability of our observations. We believe it significantly strengthens our work. We remain available for further clarification and thank the reviewer for helping us improve our paper!
>
> [1] Ashlagi et al. Functions with average smoothness: structure, algorithms, and learning.  COLT 2021
>
> [2] Ni et al, PACE: Marrying generalization in PArameter-efficient fine-tuning with Consistency rEgularization, NeurIPS 2024.
>
> [3] Bafghi et al, Parameter Efficient Fine-tuning of Self-supervised ViTs without Catastrophic Forgetting, CVPR Workshop 2024.

---

> > ### Author Rebuttal · Reviewer_92jq · 2026-04-04
> >
> > Thank you to the authors for addressing my concerns. While most issues have been resolved, I would like to see experiments in low-data settings before considering a score increase.
> >
> > **Follow-up on the authors’ response**: Thank you for addressing my concerns. They have been fully resolved, and I will increase my score accordingly.

---

> > > ### Author Response · Authors · 2026-04-07
> > >
> > > We thank the reviewer for their constructive feedback.
> > >
> > > To address the reviewer’s final concern about performance in low-data settings, we have conducted new experiments following the VTAB protocol [1], the standard low-data benchmark in the literature, in which training is restricted to 1,000 samples per dataset.
> > >
> > > Our findings remain consistent with the full-dataset setting and indicate that **prioritizing high-plasticity modules still holds in low-data settings**.
> > >
> > > Following the original ViT paper setup [2], we finetuned models for 2,500 steps using only 1000 training samples. As shown in the table below (high-plasticity components in *italic*, best accuracy in **bold**), prioritizing high-plasticity components (*MHA*, *FC1*, *FC2*) yields overall superior performance compared to low-plasticity components (LN1, LN2). This follows our original observations in the full-dataset setting.
> > >
> > > | configuration | *MHA* | *FC1* | *FC2* | LN2 | LN1 |
> > > | :--- | :---: | :---: | :---: | :---: | :---: |
> > > | **Motion Blur** | **94.3** | 93.4 | 92.7 | 91.8 | 90.8 |
> > > | **Pets** | 92.5 | **92.8** | 92.4 | 92.5 | 92.4 |
> > > | **Snow** | 94.7 | **94.9** | 94.0 | 93.8 | 93.7 |
> > > | **Speckle Noise** | **88.5** | 86.9 | 85.2 | 85.5 | 85.5 |
> > > | **AVERAGE** | **92.5** | 92.0 | 91.1 | 90.9 | 90.6 |
> > >
> > > We will add these new results to the revised version. We agree with the reviewer that this provides a more comprehensive validation of our work.
> > >
> > > The consistency of our observations across different optimizers, data scales, architectures, and modalities showcases the generalizability of our claims. We remain available for further clarification and thank the reviewer for helping us improve our paper!
> > >
> > > [1] Zhai et al. A large-scale study of representation learning with the visual task adaptation benchmark. arXiv 2020
> > >
> > > [2] Dosovitskiy et al. An Image is Worth 16x16 Words: Transformers for Image Recognition at Scale. ICLR 2021

---

### Official Review · Reviewer_TEqP · 2026-03-10

**Soundness:** 3
**Presentation:** 3
**Significance:** 2
**Originality:** 2
**Overall Recommendation:** 4
**Confidence:** 3

**Summary:**

The authors address a critical gap in vision transformer (ViT) finetuning: while transformer smoothness has been widely studied for generalization, training stability, and adversarial robustness, its role in transfer learning remains poorly understood. This submission focuses on the core question of which ViT components to prioritize during finetuning, proposing "plasticity" (defined as the average rate of change of a component’s output in response to input perturbations) as a key metric. Through theoretical analysis and experiments, the authors demonstrate that non-smooth components with high plasticity—specifically multi-head attention (MHA) modules and feedforward (FFN) layers—consistently yield better and more stable finetuning performance compared to smooth components like LayerNorm. The work challenges the prevailing assumption that smoothness is universally desirable.

**Compliance With Llm Reviewing Policy:**

Affirmed.

**Final Justification:**

My concerns have been adequately addressed. I maintain my initial rating.

**Key Questions For Authors:**

1. The authors attribute the performance differences between ViT components during finetuning solely to their smoothness (via plasticity), but fail to adequately control for or rule out the impact of other confounding factors, such as the number of parameters per component. This raises questions about whether the observed performance gaps are driven by plasticity itself or by the greater parameter capacity of high-plasticity components.
2. The paper focuses on component-wise finetuning in isolation. How would combining high-plasticity components (e.g., MHA + FC1) perform compared to individual component finetuning?

**Limitations:**

Yes

**Strengths And Weaknesses:**

## Strengths:

1. The formalization of "plasticity" as a measurable metric (average rate of change) provides a principled theoretical foundation for selecting finetuning targets, bridging the gap between transformer component properties and practical adaptation performance.
2. The paper with clear and logical structure is easy to follow.

## Weaknesses:

1. While the paper compares component-wise finetuning, it lacks a detailed comparison with state-of-the-art parameter-efficient finetuning methods (e.g., LoRA) that combine multiple component adaptations or reparameterization techniques.
2. The practical tradeoffs of prioritizing non-smooth components are not fully explored. Whether high plasticity components are more prone to overfitting on small downstream datasets or require additional regularization.
3. While the paper references comprehensive experimental results across 11 benchmarks, key comparative data  are only available in the appendix. The main text lacks concise, accessible tables that summarize core experimental contrasts, making it difficult for readers to quickly grasp performance differences between components.

---

> ### Author Rebuttal · Authors · 2026-03-31
>
> We thank the reviewer for their positive feedback and for noting that we address a critical gap in ViT finetuning with a principled theoretical foundation bridging the gap between component properties and practical performance. We address the reviewer's concerns point by point below.
>
> > **(1) While the paper compares component-wise finetuning, it lacks a [...].**
>
> While LoRA is extensively used in LLMs, we surprisingly found few papers using it with vision transformers. We add new finetuning with the standard LoRA setup ( LoRA on attention with r=8 and Adam) on Cifar100, following [1] (Reviewer 92jq reference). LoRA results are consistent with [1] and lower than our single-component approach, even when compared to LayerNorms that have **~$15 \times$ fewer trainable parameters (400K for LoRA vs. 28K for LN)**. This shows that single-component finetuning is still a competitive approach and showcases that parameter count alone does not determine the performance (best accuracy in **bold**, worst in *italic*).
>
> | Configuration | *MHA* | *FC1* | *FC2* | LN2 | LN1 | LoRA |
> | :--- | :---: | :---: | :---: | :---: | :---: | :---: |
> | **CIFAR-100** | 92.7 | **92.9** | 92.3 | 91.9 | 91.4 | *88.1* |
>
> > **(2) The practical tradeoffs of prioritizing non-smooth components are not fully explored[ [...].**
>
> Our setting of selective tuning reduces the risk of overfitting, as shown in [2]. We follow the original ViT paper setup and do not need any additional regularization for high-plasticity components. The nb of training samples ranges from 1K to 50K. We observe similar trends over all datasets, including the small ones, with non-smooth components leading to better and more stable finetuning accuracy. The benefits of non-smooth components is even more salient on challenging datasets like Cifar100, Motion Blur, or DomainNet, where the shift between pretraining and downstream data is strong (compared to Pet / Flowers, which have overall higher accuracies). We will add more discussion on the matter in the revised version.
>
> > **(3) While the paper references comprehensive experimental results across 11 benchmarks [...].**
>
> We thank the reviewer for their valuable suggestion. We will improve the presentation in the revised version by adding summarizing tables and plots in the main paper using the additional page.
>
> > **(4)The authors attribute the performance differences between ViT components during [...].**
>
> Our findings show that the plasticity ranking (MHA > FC > LN) remains consistent *including* among components with the same number of parameters: MHA leads to better and more stable performance compared to FC2 (see Tab.  1 and Fig. 2) despite both having 28M params. Normalization layers can be very effective, despite their size, to mitigate distribution shifts [3] or adapt to new data [4, 5], which suggests that lower parameter count does not limit their expressivity. This explains why normalization layers are a common choice for adapter PEFT methods [5]. It aligns with recent evidence in the literature that lower nb of params does not necessarily imply worse performance, e.g., in surgical finetuning [2] or LoRA (see answer (1)), which maintains very high performance despite drastically reducing the parameter count. We will add this discussion to the revised version of the paper.
>
> > **(5) The paper focuses on component-wise finetuning in isolation. How would combining high-plasticity [...]**
>
> We thank the reviewer for the suggestion. Combining highly plastic components such as MHA and FC1 will further improve adaptation, as the optimization will benefit from both. Additional experiments on Clipart below show that MHA+FC1 outperforms both MHA and FC1.
>
> | Configuration | *MHA + FC1* | *MHA* | *FC1* |
> | :--- | :---: | :---: | :---: |
> | **ViT-Base** | **77.5 ± 0.5** | 76.9 ± 0.6 | 75.7 ± 0.6 |
>
> Given the significant computational cost (**~800 finetuning runs** for a single optimizer across all modules and datasets), we focused on a rigorous analysis of a widely used setting (ViT-Base has **4M monthly downloads on HuggingFace**). The consistency of our findings across additional runs spanning optimizers, architecture, and modality confirms the generalizability of our observations. We believe it significantly strengthens our work. We remain available for further clarification and thank the reviewer for helping us improve our paper!
>
> [1] Bafghi et al, Parameter Efficient Fine-tuning of Self-supervised ViTs without Catastrophic Forgetting, CVPR Workshop 2024.
>
> [2] Lee et al. Surgical finetuning improves adaptation to distribution shifts. ICLR 2023
>
> [3] Kim et al. Reversible Instance Normalization for Accurate Time-Series Forecasting against Distribution Shift. ICLR 2022
>
> [4] Zhao et al. Tuning LayerNorm in Attention: Towards Efficient Multi-Modal LLM Finetuning. ICLR 2024
>
> [5] Wang et al. Tent: Fully Test-time Adaptation by Entropy Minimization. ICLR 2021

---

> > ### Author Rebuttal · Reviewer_TEqP · 2026-04-03
> >
> > Most of my concerns are adequately addressed. As pointed out by the other reviewers, there are some reasonable questions. Accordingly, I prefer to maintain the initial rating (4) for now.

---

> > > ### Author Response · Authors · 2026-04-07
> > >
> > > We thank the reviewer for acknowledging that our responses addressed their initial concerns. Regarding the *"reasonable questions"* raised by other reviewers, we have since provided extensive additional empirical evidence and clarifications that resolve them.
> > >
> > > We detail the resolution of these points below:
> > >
> > > - **Reviewer BUse**: We have resolved all concerns through strong experimental validation with extensive additional runs across optimizers, architecture, and modality. These results further validate the generalizability and significance of our findings.
> > >
> > > - **Reviewer Fepr**: We have resolved all concerns through additional experiments and clarifications regarding the significance of our work:
> > >     - Our experimental contribution provides principled guidance for practitioners and researchers to use in their specific use cases.
> > >     - Our work enables direct knowledge transfer to the community without prohibitive  computational costs (reproducing our conclusions would require over 1,000 finetuning runs (~3700 GPU hours), costing up to $40,000 on cloud providers such as Azure or AWS.)
> > >     - Our work provides a novel perspective on the role of smoothness in finetuning, combining experimental and theoretical analysis, which is notoriously challenging in deep learning, notably across different architectures and modalities.
> > >
> > > - **Reviewer 92jq**: We have resolved all concerns with extensive additional experiments, including their final request with new results in the low-data regime. Our findings remain consistent with the full-dataset setting, confirming that prioritizing high-plasticity modules holds across data scales, optimizers, architecture, and modality.
> > >
> > > We believe these enhancements address the reviewers' concerns and significantly strengthen our work.
> > >
> > > If the reviewer believes our contributions deserve to be shared with the community, we would deeply appreciate it if they could reflect these improvements in their final justification and help advocate for the paper’s acceptance.
> > >
> > > We thank them again for helping us improve our work.
> > >
> > > Best regards,
> > >
> > > The authors

---

### Official Review · Reviewer_Fepr · 2026-03-12

**Soundness:** 3
**Presentation:** 3
**Significance:** 2
**Originality:** 3
**Overall Recommendation:** 4
**Confidence:** 2

**Summary:**

This paper investigates the plasticity of ViT components during the finetuning stage. Challenging the prevailing deep learning assumption that smoothness is always beneficial, the authors demonstrate through rigorous theoretical derivation and extensive empirical analysis that high-plasticity (non-smooth) components actually yield superior finetuning results in transfer learning.

**Compliance With Llm Reviewing Policy:**

Affirmed.

**Final Justification:**

I am satisfied with the authors' responses.

**Key Questions For Authors:**

1.The theoretical derivation and experiments in this article mainly focus on ViT and image classification tasks. Considering the widespread application of large language models based on Decoder-only architecture, do you think this plasticity theory is suitable for text generation tasks? Especially after introducing causal masks, the training dynamics of attention mechanisms will undergo significant changes. A discussion for LLMs would be better for this paper.

2.If multiple plasticity components (such as MHA+FC1) are jointly fine tuned, what synergistic effect will this combination produce?

3.While the authors argue that non-smooth components improve fine-tuning, such plasticity might negatively impact pre-training. Given that fine-tuning follows pre-training, how does this characteristic ultimately influence the final fine-tuning results?

**Limitations:**

yes

**Strengths And Weaknesses:**

Strengths

1.This paper challenges the traditional assumption that smoothness is always beneficial and proposes that high plasticity (non-smooth) components are more conducive to finetuning, which is insightful.

2.This paper is theoretically supported by rigorous mathematical theory deduction.

3.This paper provides sufficient experimental support and provides clear and theoretically based model selection guidance for PEFT.

Weaknesses

1.The paper is limited to visual architecture and has not yet been validated in broader scenarios such as Large Language Models(LLMs).

2.This paper only tests a single component separately and lacks exploration of the synergistic effects of jointly fine-tuning multiple high plasticity components.

3.The experiment is mainly based on SGD optimizer, and the impact of different optimizers such as AdamW on fine-tuning performance has not been fully evaluated.

---

> ### Author Rebuttal · Authors · 2026-03-31
>
> We thank the reviewer for their valuable feedback and for noting that rigorous theoretical results and sufficient experiments support our claims. We address the reviewer's concerns point by point below.
>
> > **(1) The paper is limited to visual architecture and has not yet been validated [...]**
>
> Theoretical results of Section 3 naturally extend to LLMs (the only change from the encoder is the causal attention, for which we can extend Proposition 3 using [1, Theorem 4.3]). We add plasticity analysis on DINOv3 (7B) and GPT2 (using ImageNet and Cifar100 for DINOv3; WikiText and AG News for GPT2). As shown in the table (**see answer (3) to Reviewer 92jq**), the plasticity ranking (MHA > FC > LN) remains consistent across **architectures** (encoder vs. decoder), **pretraining objectives** (supervised, discriminative SSL, next-token prediction), and **modalities** (vision and NLP). New figure consistent with Figure 3 at this anonymous link: https://osf.io/y3fhg/files/x3w25?view_only=452822d7e7ca4ab7ac03ab813cfa4cda.
>
> > **(2) This paper only tests a single component separately [...]**
>
> We thank the reviewer for the suggestion. Combining highly plastic components such as MHA and FC1 will further improve adaptation, as the optimization will benefit from both. Additional experiments on DomainNet-Clipart below showcase that MHA+FC1 outperforms both MHA and FC1.
>
> | Configuration | *MHA + FC1* | *MHA* | *FC1* |
> | :--- | :---: | :---: | :---: |
> | **ViT-Base** | **77.5 ± 0.5** | 76.9 ± 0.6 | 75.7 ± 0.6 |
>
> > **(3) The experiment is mainly based on SGD optimizer [...]**
>
> We conduct **~120 additional finetuning runs** on ViT-Base using Adam with learning rates scaled by $10^{-2}$ following [2,3]. Our core thesis (non-smoothness is beneficial to finetuning) generalizes across **optimizers**. Finetuning non-smooth components (*in italic*) consistently yields higher accuracy and superior stability across learning rates compared to smoother modules. This trend is even more salient with Adam. New figure with Adam consistent with Figure 4 at this anonymous link: https://osf.io/y3fhg/files/2wcth?view_only=452822d7e7ca4ab7ac03ab813cfa4cda. Tables below display average finetuning accuracy across learning rates with 1) Adam (**new**), 2) SGD (**original**). Components are ranked by decreasing order of plasticity, with the best accuracy in **bold**. Results are also consistent when scaling the model size (**see answer (3) to Reviewer BUse**)
>
> | Finetuning with Adam | *MHA* | *FC1* | *FC2* | LN2 | LN1 |
> | :--- | :---: | :---: | :---: | :---: | :---: |
> | **Cifar100** | 91.0 ± 0.2 | **91.3** ± 0.6 | 90.6 ± 1.4 | 89.4 ± 2.7 | 88.4 ± 3.2 |
> | **Motion Blur** | **92.4** ± 0.2 | 91.8 ± 0.4 | 90.7 ± 0.9 | 91.7 ± 2.4 | 90.7 ± 2.4 |
> | **Clipart** | **76.4** ± 0.5 | 74.8 ± 0.8 | 75.8 ± 0.3 | 73.3 ± 1.9 | 73.5 ± 1.5 |
> | **Sketch** | 69.0 ± 0.5 | 67.9 ± 0.6 | **69.2** ± 0.4 | 63.1 ± 2.9 | 64.0 ± 2.6 |
> | **AVERAGE** | **82.2 ± 0.4** | 81.4 ± 0.6 | 81.7 ± 0.6 | 79.4 ± 2.5 | 79.2 ± 2.4 |
>
> | Finetuning with SGD | *MHA* | *FC1* | *FC2* | LN2 | LN1 |
> | :--- | :---: | :---: | :---: | :---: | :---: |
> | **Cifar100** | **91.7 ± 1.1** | 91.6 ± 1.6 | 91.2 ± 0.7 | 90.6 ± 1.4 | 89.9 ± 1.7 |
> | **Motion Blur** | 94.1 ± 0.2 | **94.4 ± 0.2** | 93.7 ± 0.3 | 93.7 ± 0.6 | 92.6 ± 0.4 |
> | **Clipart** | **76.9 ± 0.6** | 75.7 ± 0.6 | 75.9 ± 0.8 | 73.6 ± 0.8 | 74.0 ± 0.5 |
> | **Sketch** | **68.5 ± 1.1** | 68.1 ± 1.6 | 68.1 ± 2.0 | 63.9 ± 1.6 | 64.5 ± 1.0 |
> | **AVERAGE** | **82.8 ± 0.8** | 82.5 ± 1.0 | 82.1 ± 1.2 | 80.5 ± 1.1 | 80.2 ± 0.9 |
>
> > **(4) The theoretical derivation and experiments in this article [...]**
>
> Answered in (1).
>
> > **(5) If multiple plasticity components (such as MHA+FC1) are jointly [...]**
>
> Answered in (2)
>
> > **(6) While the authors argue that non-smooth components improve [...]**
>
> The entire network is optimized during pretraining and, as such, the smoothness of the whole architecture is at play, not the smoothness of individual components. Interestingly, [4] showed that achieving strong performance with deep transformers required very high Lipschitz constants (non-smooth network).
>
> Given the significant computational cost (**~800 finetuning runs** for a single optimizer across all modules and datasets), we focused on a rigorous analysis of a widely used setting (ViT-Base has **4M monthly downloads on HuggingFace**). The consistency of our findings across **120+ additional runs** (spanning optimizers, architecture, and modality) confirms the generalizability of our observations. We believe it significantly strengthens our work. We remain available for further clarification and thank the reviewer for helping us improve our paper!
>
> [1] Castin et al. How Smooth is Attention? ICML 2024
>
> [2] Touvron et al. Training data-efficient image transformers & distillation through attention. ICML 2021
>
> [3] Kumar et al. How to fine-tune vision models with SGD. arXiv 2023.
>
> [4] Newhouse et al. Training Transformers with Enforced Lipschitz Bounds. arXiv 2025

---

> > ### Author Rebuttal · Reviewer_Fepr · 2026-04-03
> >
> > Thank you for your response. I have the following question.
> >
> > Without your theory, people might still discover through experimentation which component is better for fine-tuning. So how do you demonstrate the significance of your paper?

---

> > > ### Author Response · Authors · 2026-04-04
> > >
> > > We are pleased that our extensive additional experiments partially addressed the reviewer's concerns.
> > >
> > > We clarify the significance of our work below (we would appreciate further clarification should our understanding of the reviewer's question be incomplete).
> > >
> > > As acknowledged by the reviewer, our paper provides clear model selection guidance for PEFT methods, supported both **theoretically and experimentally**.
> > >
> > > - **Experimental contributions**: We demonstrate generalizable findings across optimizers, architectures, and modalities. Thanks to the reviewers' valuable suggestions, our contributions were strengthened during the rebuttal (extension to Adam, DINOv3, GPT2). Our study serves as **principled guidance** for practitioners and researchers to use in their specific use cases.
> > >
> > > - **Knowledge transfer**: Our analysis is available to the community (alongside an open-source implementation upon publication). This is critical for **resource-efficient research**: reproducing our conclusions requires over 1,000 finetuning runs (equivalent to **3700 GPU hours**), costing up to **$40,000** on cloud providers such as Azure or AWS. Our paper enables direct knowledge transfer without these computational and environmental costs.
> > >
> > > - **Scope of our contributions**: Our work provides a novel perspective on the role of smoothness in finetuning. All reviewers acknowledge that our claims are well-supported both theoretically and experimentally, which is notoriously challenging to achieve in deep learning, notably across different architectures and modalities.
> > >
> > > We thank the reviewer for their valuable suggestions, which helped us improve our work and remain at their disposal to answer any remaining questions.
> > >
> > > Best regards,
> > >
> > > The authors

---

### Official Review · Reviewer_BUse · 2026-03-13

**Soundness:** 3
**Presentation:** 4
**Significance:** 4
**Originality:** 4
**Overall Recommendation:** 4
**Confidence:** 3

**Summary:**

This paper argues that the best ViT components to finetune are not the smoothest ones, but the most plastic ones. It defines plasticity as the average input-output rate of change of a transformer component, proves that attention and feedforward layers should have higher plasticity than LayerNorms, and validates this on large-scale experiments. The empirical ranking is MHA > FC1 > FC2 > LN2 > LN1, and this same ordering predicts downstream finetuning performance. The main practical message is that attention modules, and then feedforward layers, should be prioritized for selective ViT finetuning because their non-smoothness makes them easier to optimize and better able to adapt to new tasks.

**Compliance With Llm Reviewing Policy:**

Affirmed.

**Final Justification:**

I appreciate the authors’ detailed responses to my concerns. Considering that the work demonstrates solid experimental rigor and holds potential for meaningful impact, I am adjusting my score in a more favorable direction. However, my confidence in this assessment remains unchanged.

**Key Questions For Authors:**

Please see the weaknesses

**Limitations:**

Yes

**Strengths And Weaknesses:**

Strengths:
- Conceptually, it gives a principled explanation for why some selective finetuning choices work better than others.
- Practically, it suggests a simple rule for ViT adaptation
- It also hints that future PEFT methods, such as LoRA could become stronger if applied preferentially to the high-plasticity modules rather than uniformly.

Weaknesses:
- The central claim of the paper—that non-smooth components are beneficial for finetuning—appears difficult to reconcile with a substantial body of literature suggesting that smoother optimization landscapes lead to improved training stability and generalization [ref_1][ref_2][ref_3]. While the authors focus on module-level plasticity rather than loss landscape curvature, the paper does not clearly explain how these notions relate to or differ from established smoothness-based theories of generalization. As a result, it remains unclear whether the observed benefits stem from non-smoothness itself or from other architectural or optimization factors.

- Given that the claim challenges commonly held assumptions about smoothness and generalization, stronger empirical validation would be necessary. However, the experiments are limited to classification benchmarks on pretrained ViTs, mostly ViT-Base and a single ViT-Huge analysis. The evaluation does not include other architectures, downstream tasks, or modern parameter-efficient finetuning methods. As a result, it is difficult to determine whether the observed phenomenon generalizes beyond the specific experimental setup considered in this paper.

- Preventing excessive smoothing may help preserve representational diversity, but this does not necessarily imply that non-smooth dynamics should be preferred. The paper does not clearly distinguish between these two mechanisms. As a result, it remains unclear whether the observed improvements arise from avoiding over-smoothing or from genuinely beneficial non-smooth dynamics.

If the claims are further validated both empirically and visually across a wider range of experimental setups, I would be willing to reconsider my score.

References:

[ref_1] Zhang, X., Wang, P., & Wang, W. Lost in the Non-convex Loss Landscape: How to Fine-tune the Large Time Series Model?. In The Fourteenth International Conference on Learning Representations.

[ref_2] Ly, A., & Gong, P. (2025). Optimization on multifractal loss landscapes explains a diverse range of geometrical and dynamical properties of deep learning. Nature Communications, 16(1), 3252.

[ref_3] Runkel, C., Meli, N. K., Lukasik, J., Biguri, A., Schönlieb, C. B., & Moeller, M. (2025). Smooth Model Compression without Fine-Tuning. arXiv preprint arXiv:2505.24469.

---

> ### Author Rebuttal · Authors · 2026-03-31
>
> We thank the reviewer for the helpful feedback and for recognizing the conceptual and practical value of our work. We address the reviewer's concerns point by point below.
>
> > **(1) The central claim of the paper—that non-smooth components are [...]**
>
> We use functional smoothness to identify components that best adapt the model to new data.  Loss landscape smoothness relates to the parameter space via $\nabla^2_\theta \mathcal{L}$. Our central claim does not contradict the consensus on loss landscape:
> - [1] shows that models converging to sharper minima often outperform flatter counterparts (Fig. 1). Similar findings are given in [2] via large-scale pretraining/finetuning of transformers, but show that finding the right sharpness measure that correlates with generalization is data-dependent and remains an open problem (abstract and Fig. 3).
> - Our fine-grained module-level study gives practical guidance for when non-smoothness can be beneficial. While we do not claim to solve the open problem in [2], our notion of smoothness seems to consistently correlate with performance for the specific case of vision transformers.
>
> In the revised version, we will include these references and position our work with respect to them as suggested by the reviewer.
>
> > **(2) As a result, it remains unclear whether the observed benefits stem from non-smoothness [...]**
>
> Additional experiments on Adam and ViT-Large (307M) show that observed benefits are consistent across optimizers and scales. Details below.
>
> > **(3) Given that the claim challenges commonly held assumptions [...]**
>
> New extensive experiments with **~120 additional finetuning runs** show that our core thesis (plasticity ranking of components inherent to the transformer architecture / non-smoothness is beneficial to finetuning) generalizes across **optimizers, model sizes, and architectures**.
>
> - **Optimizers**: new finetuning runs using Adam with learning rates scaled by $10^{-2}$ following [3,4] (**see answer (3) to Reviewer Fepr**). Our findings are invariant to the optimizer: finetuning non-smooth components (*in italic*) consistently yields higher accuracy and superior stability across learning rates compared to smoother modules. This trend is even more salient with Adam. New figure with Adam consistent with Figure 4 at this anonymous link: https://osf.io/y3fhg/files/2wcth?view_only=452822d7e7ca4ab7ac03ab813cfa4cda.
> - **Model size**: we add finetuning runs on ViT-Large, with consistent findings when scaling from 86M to 307M params:
> | configuration | **MHA** | *FC1* | *FC2* | LN2 | LN1 |
> | :--- | :---: | :---: | :---: | :---: | :---: |
> | **ViT-Base (86M)** | **76.9** ± 0.6 | 75.7 ± 0.6 | 75.9 ± 0.8 | 73.6 ± 0.8 | 74.0 ± 0.5 |
> | **ViT-Large (307M)** | **78.4** ± 0.2 | 77.3 ± 0.2 | 78.3 ± 0.3 | 76.1 ± 0.3 | 76.8 ± 0.3 |
> - **Architectures**: new plasticity analysis on DINOv3 (7B) and GPT2 (using ImageNet and Cifar100 for DINOv3; WikiText and AG News for GPT2) . As shown in the table (**see answer (3) to Reviewer 92jq**), the plasticity ranking (MHA > FC > LN) remains remarkably consistent across **architectures** (encoder vs. decoder), **pretraining objectives** (supervised, discriminative SSL, and next-token prediction), and **modalities** (vision and NLP). New figure consistent with Figure 3 at this anonymous link: https://osf.io/y3fhg/files/x3w25?view_only=452822d7e7ca4ab7ac03ab813cfa4cda.
> - **LoRA**: we add finetuning with LoRA (**see answer (1) to Reviewer TEqP**). It achieves lower performance, even compared to LayerNorms that have **~$15 \times$ fewer trainable parameters (400K for LoRA vs. 28K for LN)**. This shows that single-component finetuning is a competitive approach.
>
> > **(4) Preventing excessive smoothing may help preserve [...]**
>
> Our findings indicate that the plasticity ranking (MHA > FC > LN) consistently leads to better and more stable finetuning performance: while the feedforward layers are not over-smoothed, they exhibit lower plasticity than attention modules and accordingly lower finetuning accuracy.
>
> Given significant computational cost (**~800 finetuning runs** for a single optimizer across all modules and datasets), we focused on a rigorous analysis of a widely used setting (ViT-Base has **4M monthly downloads on HuggingFace**). The consistency of our findings across **120+ additional runs** (spanning optimizers, architecture, and modality) confirms the generalizability of our observations. We believe it significantly strengthens our work. We remain available for further clarification and thank the reviewer for helping us improve our paper!
>
> [1] Mason-Williams et al. A function-centric perspective on flat and sharp minima. arxiv 2025.
>
> [2] Andriushchenko et al. A Modern Look at the Relationship between Sharpness and Generalization. ICML 2023.
>
> [3] Touvron et al. Training data-efficient image transformers & distillation through attention. ICML 2021
>
> [4] Kumar et al. How to fine-tune vision models with SGD. arXiv 2023

---

> > ### Author Rebuttal · Reviewer_BUse · 2026-04-03
> >
> > I appreciate the authors’ thorough and thoughtful responses to my concerns. Given the strong experimental validation presented, which supports contributions that are both meaningful and potentially impactful, I am inclined to increase my score.

---

> > > ### Author Response · Authors · 2026-04-03
> > >
> > > We are pleased to read that the reviewer’s concerns have been fully addressed, thanks to our strong empirical validation. We thank the reviewer for their valuable suggestions, which helped us improve our work, and for the increased score. This kind of feedback feels encouraging.
> > >
> > > Best regards,
> > >
> > > The authors

---

### Decision · Program_Chairs · 2026-04-30

**Decision:**

Accept (regular)

**Comment:**

This paper theoretically and empirically explores the component-level plasticity of the transformer architecture and provides practically useful insights for fine tuning. The plasticity is formalized as adaptability of the output in response to input changes. The provided theory upper bounds the plasticity of the transformer component, showing the multi-head self-attention module has the highest upper bound (and then the feedforward linear layer). The experiments are mainly on Vision Transformer (ViT) architectures (with additional LLM experiments in the rebuttal), showing that the attention module has the highest empirical plasticity (and then the first and second feedforward layers). The experiments show that fine tuning only high plasticity components such the attention module and feedforward layers improves performance and stability.

This paper joins theory and practice to provide interesting foundational insights on analytical models and translating them to practical benefits on deep networks.

The authors addressed very well the initial reviews. Specifically, in the rebuttal, the authors provided additional experiments that supports their paper’s claims, including experiments for LLMs (GPT-2), ViT-L as a larger model than ViT-B, Adam as an additional optimizer beyond SGD, joint fine tuning of two high plasticity components, comparison to LoRA.

While the reviews reflect appreciation of the paper (and I also appreciate it), the overall positive evaluation is somewhat restrained (as reflected by no rating above Weak Accept, a question by Reviewer Fepr on the practical significance of the contribution, and a ``partially resolved’’ grade that Reviewer TEqP gave to the rebuttal).

Accordingly, this is a borderline paper that gets a Weak Accept recommendation.